# Max-Margin Token Selection in Attention Mechanism

**Davoud Ataee Tarzanagh**
University of Pennsylvania
`tarzanaq@upenn.edu`

**Yingcong Li**  **Xuechen Zhang**
University of California, Riverside
`{yli692,xzhan394}@ucr.edu`

**Samet Oymak**
University of Michigan
UC Riverside
`oymak@umich.edu`

## Abstract

Attention mechanism is a central component of the transformer architecture which led to the phenomenal success of large language models. However, the theoretical principles underlying the attention mechanism are poorly understood, especially its nonconvex optimization dynamics. In this work, we explore the seminal softmax-attention model $f(X) = \langle Xv, \texttt{softmax}(XWp)\rangle$, where $X$ is the token sequence and $(v, W, p)$ are trainable parameters. We prove that running gradient descent on $p$, or equivalently $W$, converges in direction to a max-margin solution that separates *locally-optimal* tokens from non-optimal ones. This clearly formalizes attention as an optimal token selection mechanism. Remarkably, our results are applicable to general data and precisely characterize *optimality* of tokens in terms of the value embeddings $Xv$ and problem geometry. We also provide a broader regularization path analysis that establishes the margin maximizing nature of attention even for nonlinear prediction heads. When optimizing $v$ and $p$ simultaneously with logistic loss, we identify conditions under which the regularization paths directionally converge to their respective hard-margin SVM solutions where $v$ separates the input features based on their labels. Interestingly, the SVM formulation of $p$ is influenced by the support vector geometry of $v$. Finally, we verify our theoretical findings via numerical experiments and provide insights.

## 1 Introduction

Since its introduction in the seminal work [1], attention mechanism has played an influential role in advancing natural language processing, and more recently, large language models [2, 3, 4, 5]. Initially introduced for encoder-decoder RNN architectures, attention allows the decoder to focus on the most relevant parts of the input sequence, instead of relying solely on a fixed-length hidden state. Attention mechanism has taken the center stage in the transformers [6], where the self-attention layer – which calculates softmax similarities between input tokens – serves as the backbone of the architecture. Since their inception, transformers have revolutionized natural language processing, from models like BERT [7] to ChatGPT [8], and have also become the architecture of choice for foundation models [9] addressing diverse challenges in generative modeling [3, 10], computer vision [11, 12], and reinforcement learning [13, 14, 15].

The prominence of the attention mechanism motivates a fundamental theoretical understanding of its role in optimization and learning. While it is well-known that attention enables the model to focus on the relevant parts of the input sequence, the precise mechanism by which this is achieved is far from clear. To this end, we ask

> **Q:** What are the optimization dynamics and inductive biases of the attention mechanism?

We study this question using the fundamental attention model $f(X) = \langle Xv, \mathbb{S}(XW^\top p)\rangle$. Here, $X$ is the sequence of input tokens, $v$ is the prediction head, $W$ is the trainable key-query weights, and $\mathbb{S}$

37th Conference on Neural Information Processing Systems (NeurIPS 2023).

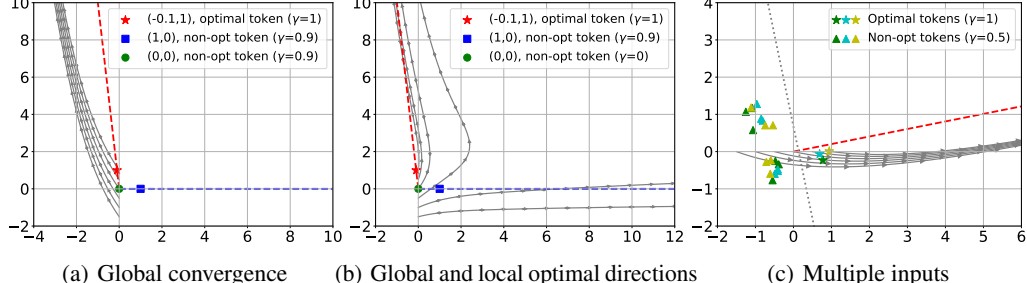

Figure 1: The convergence behavior of the gradient descent on the attention weights $p$ using the logistic loss in (ERM). The arrows (—>—) represent trajectories from different initializations. Here, (- - -) and (- - -) denote the **g**lobally- and **l**ocally-optimal **m**ax-**m**argin directions (GMM, LMM). $\gamma$ denotes the *score* of a token per Definition 1. Discussion is provided under Theorems 2 and 3.

denotes the softmax nonlinearity. For transformers, $p$ corresponds to the [CLS] token or tunable prompt [16, 17, 18], whereas for RNN architectures [1], $p$ corresponds to the hidden state. Given training data $(Y_i, X_i)_{i=1}^n$ with labels $Y_i \in \{-1, 1\}$ and inputs $X_i \in \mathbb{R}^{T \times d}$, we consider the empirical risk minimization with a decreasing loss function $\ell(\cdot) : \mathbb{R} \to \mathbb{R}$,

$$\mathcal{L}(v, p, W) = \frac{1}{n} \sum_{i=1}^n \ell(Y_i \cdot f(X_i)), \quad \text{where} \quad f(X_i) = v^\top X_i^\top \mathbb{S}(X_i W^\top p). \tag{1}$$

At a high-level, this work establishes fundamental equivalences between the optimization trajectories of (1) and hard-margin SVM problems. Our main contributions are as follows:

• **Optimization geometry of attention (Sec 2):** We first show that gradient iterations of $p$ and $W$ admit a one-to-one mapping, thus we focus on optimizing $p$ without losing generality. In Theorem 3, we prove that, under proper initialization:

> Gradient descent on $p$ converges in direction to a max-margin solution – namely (ATT-SVM) – that separates locally-optimal tokens from non-optimal ones.

We call these **L**ocally-optimal **M**ax-**M**argin (LMM) directions and show that these thoroughly characterize the viable convergence directions of attention when the norm of its weights grows to infinity. We also identify conditions under which (algorithm-independent) regularization path and gradient descent path converge to **G**lobally-optimal **M**ax-**M**argin (GMM) direction in Theorems 1 and 2, respectively. A central feature of our results is precisely quantifying *optimality* in terms of *token scores* $\gamma_t = Y \cdot v^\top x_t$ where $x_t$ is the $t^{\text{th}}$ token of the input sequence $X$. *Locally-optimal* tokens are those with higher scores than their nearest neighbors determined by the SVM solution. These are illustrated in Figure 1.

• **Optimize attention $p$ and prediction-head $v$ jointly (Sec 3):** We study the joint problem under logistic loss function. We use regularization path analysis where (ERM) is solved under ridge constraints and we study the solution trajectory as the constraints are relaxed. Since the problem is linear in $v$, if the attention features $x_i^{att} = X_i^\top \mathbb{S}(X_i W^\top p)$ are separable based on their labels $Y_i$, $v$ would implement a max-margin classifier. Building on this, we prove that $p$ and $v$ converges to their respective max-margin solutions under proper geometric conditions (Theorem 5). Relaxing these conditions, we obtain a more general solution where margin constraints on $p$ are relaxed on the inputs whose attention features are not support vectors of $v$ (Theorem 6). Figure 3 illustrates these outcomes.

The next section introduces the preliminary concepts, Section 4 presents numerical experiments[1], Section 5 discusses related literature, and Section 6 highlights limitations and future work.

## 1.1 Preliminaries

**Notations.** For any integer $N \geq 1$, let $[N] := \{1, \ldots, N\}$. We use lower-case and upper-case bold letters (e.g. $a$ and $A$) to represent vectors and matrices, respectively. The entries of $a$ are denoted as $a_i$. We use $\bar{\sigma}(A)$ to denote the maximum singular value of $A$. We denote the minimum of two numbers $a, b$ as $a \wedge b$, and the maximum as $a \vee b$. Big-O notation $O(\cdot)$ hides the universal constants. Throughout, we will use $\mathcal{L}(p)$ and $\mathcal{L}(v, p)$ to denote Objective (1) with fixed $(v, W)$ and $W$, respectively.

**Optimization.** Given an objective function $\mathcal{L} : \mathbb{R}^d \to \mathbb{R}$ and an $\ell_2$-norm bound $R$, define the regularized solution as

---

[1]The code for experiments can be found at https://github.com/ucr-optml/max_margin_attention.

$$\bar{\boldsymbol{p}}(R) := \arg\min_{\|\boldsymbol{p}\|\leq R} \mathcal{L}(\boldsymbol{p}). \tag{2}$$

Regularization path – the evolution of $\bar{\boldsymbol{p}}(R)$ as $R$ grows – is known to capture the spirit of gradient descent as the ridge constraint $R$ provides a proxy for the number of gradient descent iterations. For instance, [19, 20, 21] study the implicit bias of logistic regression and rigorously connect the directional convergence of regularization path (i.e. $\lim_{R\to\infty} \bar{\boldsymbol{p}}(R)/R$) and gradient descent. For gradient descent, we assume the objective $\mathcal{L}(\boldsymbol{p})$ is smooth and describe the gradient descent process as

$$\boldsymbol{p}(t+1) = \boldsymbol{p}(t) - \eta(t)\nabla\mathcal{L}(\boldsymbol{p}(t)), \tag{3}$$

where $\eta(t)$ is the stepsize at time $t$ and $\nabla\mathcal{L}(\boldsymbol{p}(t))$ is the gradient of $\mathcal{L}$ at $\boldsymbol{p}(t)$.

**Attention in Transformers.** Next, we will discuss the connection between our model and the attention mechanism used in transformers. Our exposition borrows from [17], where the authors analyze the same attention model using gradient-based techniques on specific contextual datasets.

• **Self-attention** is the core building block of transformers [6]. Given an input consisting of $T$ tokens $\boldsymbol{X} = [\boldsymbol{x}_1, \ldots, \boldsymbol{x}_T]^\top \in \mathbb{R}^{T\times d}$, self-attention with key-query matrix $\boldsymbol{W} \in \mathbb{R}^{d\times d}$, and value matrix $\boldsymbol{V} \in \mathbb{R}^{d\times v}$, the self-attention model is defined as follows:

$$f_{sa}(\boldsymbol{X}) = \mathbb{S}(\boldsymbol{X}\boldsymbol{W}\boldsymbol{X}^\top)\boldsymbol{X}\boldsymbol{V}. \tag{4}$$

Here, $\mathbb{S}(\cdot)$ is the softmax nonlinearity that applies row-wise on the similarity matrix $\boldsymbol{X}\boldsymbol{W}\boldsymbol{X}^\top$.

• **Tunable tokens: [CLS] and prompt-tuning.** In practice, we append additional tokens to the raw input features $\boldsymbol{X}$: For instance, a [CLS] token is used for classification purposes [7] and prompt vectors can be appended for adapting a pretrained model to new tasks [16, 18]. Let $\boldsymbol{p} \in \mathbb{R}^d$ be the tunable token ([CLS] or prompt vector) and concatenate it to $\boldsymbol{X}$ to obtain $\boldsymbol{X_p} := [\boldsymbol{p}\ \boldsymbol{X}^\top]^\top \in \mathbb{R}^{(T+1)\times d}$. Consider the cross-attention features obtained from $\boldsymbol{X_p}$ and $\boldsymbol{X}$ given by

$$\begin{bmatrix} f_{cls}^\top(\boldsymbol{X}) \\ f_{sa}(\boldsymbol{X}) \end{bmatrix} = \mathbb{S}(\boldsymbol{X_p}\boldsymbol{W}\boldsymbol{X}^\top)\boldsymbol{X}\boldsymbol{V} = \begin{bmatrix} \mathbb{S}(\boldsymbol{p}^\top\boldsymbol{W}\boldsymbol{X}^\top) \\ \mathbb{S}(\boldsymbol{X}\boldsymbol{W}\boldsymbol{X}^\top) \end{bmatrix}\boldsymbol{X}\boldsymbol{V}.$$

The beauty of cross-attention is that it isolates the contribution of $\boldsymbol{p}$ under the upper term $f_{cls}(\boldsymbol{X}) = \boldsymbol{V}^\top\boldsymbol{X}^\top\mathbb{S}(\boldsymbol{X}\boldsymbol{W}^\top\boldsymbol{p}) \in \mathbb{R}^v$. In this work, we use the value weights for classification, thus we set $v = 1$, and denote $\boldsymbol{v} = \boldsymbol{V} \in \mathbb{R}^d$. This brings us to our attention model of interest:

$$f(\boldsymbol{X}) = \boldsymbol{v}^\top\boldsymbol{X}^\top\mathbb{S}(\boldsymbol{K}\boldsymbol{p}), \quad \text{where} \quad \boldsymbol{K} = \boldsymbol{X}\boldsymbol{W}^\top. \tag{5}$$

Here, $(\boldsymbol{v}, \boldsymbol{W}, \boldsymbol{p})$ are the tunable model parameters and $\boldsymbol{K}$ is the key embeddings. Note that $\boldsymbol{W}$ and $\boldsymbol{p}$ are playing the same role within softmax, thus, it is intuitive that they exhibit similar optimization dynamics. Confirming this, the next lemma shows that gradient iterations of $\boldsymbol{p}$ (after setting $\boldsymbol{W} \leftarrow$ Identity) and $\boldsymbol{W}$ admit a one-to-one mapping.

**Lemma 1** *Fix* $\boldsymbol{u} \in \mathbb{R}^d \setminus \{\boldsymbol{0}\}$ *. Let* $\psi : \mathbb{R}^d \to \mathbb{R}$ *and* $\ell : \mathbb{R} \to \mathbb{R}$ *be differentiable functions. On the same training data* $(Y_i, \boldsymbol{X}_i)_{i=1}^n$, *define* $\tilde{\mathcal{L}}(\boldsymbol{p}) := 1/n\sum_{i=1}^n \ell(Y_i \cdot \psi(\boldsymbol{X}_i^\top\mathbb{S}(\boldsymbol{X}_i\boldsymbol{p})))$ *and* $\mathcal{L}(\boldsymbol{W}) := 1/n\sum_{i=1}^n \ell(Y_i \cdot \psi(\boldsymbol{X}_i^\top\mathbb{S}(\boldsymbol{X}_i\boldsymbol{W}^\top\boldsymbol{u})))$. *Consider the gradient descent iterations on* $\boldsymbol{p}$ *and* $\boldsymbol{W}$ *with initial values* $\boldsymbol{p}(0)$ *and* $\boldsymbol{W}(0) = \boldsymbol{u}\boldsymbol{p}(0)^\top/\|\boldsymbol{u}\|^2$ *and stepsizes* $\eta$ *and* $\eta/\|\boldsymbol{u}\|^2$, *respectively:*

$$\boldsymbol{p}(t+1) = \boldsymbol{p}(t) - \eta\nabla\tilde{\mathcal{L}}(\boldsymbol{p}(t)),$$

$$\boldsymbol{W}(t+1) = \boldsymbol{W}(t) - \frac{\eta}{\|\boldsymbol{u}\|^2}\nabla\mathcal{L}(\boldsymbol{W}(t)).$$

*We have that* $\boldsymbol{W}(t) = \boldsymbol{u}\boldsymbol{p}(t)^\top/\|\boldsymbol{u}\|^2$ *for all* $t \geq 0$.

This lemma directly characterizes the optimization dynamics of $\boldsymbol{W}$ through the dynamics of $\boldsymbol{p}$, allowing us to reconstruct $\boldsymbol{W}$ from $\boldsymbol{p}$ using their gradient iterations. Therefore, we will fix $\boldsymbol{W}$ and concentrate on optimizing $\boldsymbol{p}$ in Section 2 and the joint optimization of $(\boldsymbol{v}, \boldsymbol{p})$ in Section 3.

---

**Problem definition:** Throughout, $(Y_i, \boldsymbol{X}_i)_{i=1}^n$ denotes training dataset where $Y_i \in \{-1, 1\}$ and $\boldsymbol{X}_i \in \mathbb{R}^{T\times d}$. We denote the key embeddings of $\boldsymbol{X}_i$ via $\boldsymbol{K}_i = \boldsymbol{X}_i\boldsymbol{W}^\top$ and explore the training risk

$$\mathcal{L}(\boldsymbol{v}, \boldsymbol{p}) = \frac{1}{n}\sum_{i=1}^n \ell\left(Y_i \cdot \boldsymbol{v}^\top\boldsymbol{X}_i^\top\mathbb{S}(\boldsymbol{K}_i\boldsymbol{p})\right). \tag{ERM}$$

Importantly, our results apply to general tuples $(Y_i, \boldsymbol{X}_i, \boldsymbol{K}_i)$ and do not assume that $(\boldsymbol{X}_i, \boldsymbol{K}_i)$ are tied via $\boldsymbol{W}$. Finally, the $t^{th}$ tokens of $\boldsymbol{X}_i, \boldsymbol{K}_i$ are denoted by $\boldsymbol{x}_{it}, \boldsymbol{k}_{it} \in \mathbb{R}^d$, respectively, for $t \in [T]$.

The highly nonlinear and nonconvex nature of the softmax operation makes the training problem in (ERM) a challenging nonconvex optimization problem for $\boldsymbol{p}$, even with a fixed $\boldsymbol{v}$. In the next section, we will introduce a set of assumptions to demonstrate the global and local convergence of gradient descent for margin maximization in the attention mechanism.

## 2  Global and Local Margin Maximization with Attention

In this section, we present the main results of this paper (Theorems 2 and 3) by examining the implicit bias of gradient descent on learning $\boldsymbol{p} \in \mathbb{R}^d$ given a fixed choice of $\boldsymbol{v} \in \mathbb{R}^d$. Notably, our results apply to general *decreasing loss functions without requiring convexity*. This generality is attributed to margin maximization arising from the exponentially-tailed nature of softmax within attention, rather than $\ell$. We maintain the following assumption on the loss function throughout this section.

**Assumption A (Well-behaved Loss)**  *Over any bounded interval: (1) $\ell : \mathbb{R} \to \mathbb{R}$ is strictly decreasing. (2) $\ell'$ is $M_0$-Lipschitz continuous and $|\ell'(u)| \le M_1$.*

Assumption A includes many common loss functions, including the logistic loss $\ell(u) = \log(1 + e^{-u})$, exponential loss $\ell(u) = e^{-u}$, and correlation loss $\ell(u) = -u$. Assumption A implies that $\mathcal{L}(\boldsymbol{p})$ is $L_p$–smooth (see Lemma 6 in Supplementary), where

$$L_p := \frac{1}{n} \sum_{i=1}^n \left( M_0 \|\boldsymbol{v}\|^2 \|\boldsymbol{W}\|^2 \|\boldsymbol{X}_i\|^4 + 3M_1 \|\boldsymbol{v}\| \, \|\boldsymbol{W}\|^2 \|\boldsymbol{X}_i\|^3 \right). \tag{6}$$

We now introduce a convex hard-margin SVM problem that separates one token of the input sequence from the rest, jointly solved over all inputs. We will show that this problem captures the optimization properties of softmax-attention. Fix indices $\boldsymbol{\alpha} = (\alpha_i)_{i=1}^n$ and consider

$$\boxed{\boldsymbol{p}^{mm}(\boldsymbol{\alpha}) = \arg\min_{\boldsymbol{p}} \|\boldsymbol{p}\| \quad \text{subject to} \quad \min_{t \ne \alpha_i} \boldsymbol{p}^\top(\boldsymbol{k}_{i\alpha_i} - \boldsymbol{k}_{it}) \ge 1, \ \text{ for all } \ 1 \le i \le n. \ \text{(ATT-SVM)}}$$

Note that existence of $\boldsymbol{p}^{mm}(\boldsymbol{\alpha})$ implies the separability of tokens $\boldsymbol{\alpha}$ from the others. Specifically, choosing direction $\boldsymbol{p}^{mm}(\boldsymbol{\alpha})$ will exactly select tokens $(\boldsymbol{x}_{i\alpha_i})_{i=1}^n$ at the attention output for each input sequence, that is, $\lim_{R \to \infty} \boldsymbol{X}_i^\top \mathbb{S}(R \cdot \boldsymbol{K}_i \boldsymbol{p}^{mm}(\boldsymbol{\alpha})) = \boldsymbol{x}_{i\alpha_i}$. We are now ready to introduce our main results that characterize the global and local convergence of the attention weights $\boldsymbol{p}$ via (ATT-SVM).

### 2.1  Global convergence of the attention weights *p*

We first identify the conditions that guarantee the global convergence of gradient descent for $\boldsymbol{p}$. The intuition is that, in order for attention to exhibit implicit bias, the softmax nonlinearity should be forced to select the *optimal token within each input sequence*. Fortunately, the optimal tokens that achieve the smallest training objective under decreasing loss function $\ell(\cdot)$ have a clear definition.

**Definition 1 (Token Scores, Optimality & GMM)**  *The score of token $\boldsymbol{x}_{it}$ of input $\boldsymbol{X}_i$ is defined as $\gamma_{it} := Y_i \cdot \boldsymbol{v}^\top \boldsymbol{x}_{it}$. The optimal tokens for input $\boldsymbol{X}_i$ are those tokens with highest scores given by*

$$opt_i \in \arg\max_{t \in [T]} \gamma_{it}.$$

*Globally-optimal max-margin (GMM) direction is defined as the solution of* (ATT-SVM) *with optimal indices $(opt_i)_{i=1}^n$ by $\boldsymbol{p}^{mm\star}$.*

It is worth noting that score definition simply uses the *value embeddings* $\boldsymbol{v}^\top \boldsymbol{x}_{it}$ of the tokens. Note that multiple tokens within an input might attain the same score, thus $opt_i$ or $\boldsymbol{p}^{mm\star}$ may not be unique. The theorem below provides our regularization path guarantee on the global convergence of attention.

**Theorem 1 (Regularization Path)**  *Suppose Assumption A on the loss function holds, and for all $i \in [n]$ and $t \ne opt_i$, the scores obey $\gamma_{it} < \gamma_{i\,opt_i}$. Then, the regularization path $\bar{\boldsymbol{p}}(R) = \arg\min_{\|\boldsymbol{p}\| \le R} \mathcal{L}(\boldsymbol{p})$ converges to the GMM direction i.e. $\lim_{R \to \infty} \bar{\boldsymbol{p}}(R)/R = \boldsymbol{p}^{mm\star}/\|\boldsymbol{p}^{mm\star}\|$.*

Theorem 1 shows that as the regularization strength $R$ increases towards the ridgeless problem $\min_{\boldsymbol{p}} \mathcal{L}(\boldsymbol{p})$, the optimal direction $\bar{\boldsymbol{p}}(R)$ aligns more closely with the max-margin solution $\boldsymbol{p}^{mm\star}$. Since this theorem allows for arbitrary token scores, it demonstrates that *max-margin token separation is an essential feature of the attention mechanism*. In fact, it is a corollary of Theorem 8, which applies to

the generalized model $f(X) = \psi(X^\top \mathbb{S}(XW^\top p))$ and accommodates multiple optimal tokens per input. However, while regularization path analysis captures the global behavior, gradient descent lacks general global convergence guarantees. In Section 2.2, we show that due to the nonconvex landscape and softmax nonlinearity, gradient descent often converges to local optima. We first establish that when (ERM) is trained with gradient descent, the norm of the parameters will diverge. For the restrictive setting of $n = 1$, gradient descent also exhibits a global convergence guarantee.

**Assumption B** *For all $i \in [n]$ and $t, \tau \neq \mathtt{opt}_i$, the scores per Definition 1 obey $\gamma_{it} = \gamma_{i\tau} < \gamma_{i\mathtt{opt}_i}$.*

**Theorem 2 (Global Convergence of Gradient Descent)** *Suppose Assumption A on the loss function $\ell$ and Assumption B on the tokens' score hold. Then, the gradient descent iterates $p(t + 1) = p(t) - \eta \nabla \mathcal{L}(p(t))$ on (ERM), with the stepsize $\eta \leq 1/L_p$ and any starting point $p(0)$ satisfy $\lim_{t\to\infty} \|p(t)\| = \infty$. If $n = 1$, we also have $\lim_{t\to\infty} p(t)/\|p(t)\| = p^{mm\star}/\|p^{mm\star}\|$.*

Theorem 2 shows that gradient descent will diverge in norm, and when $n = 1$, the normalized predictor $p(t)/\|p(t)\|$ converges towards $p^{mm\star}$, the separator of the globally optimal token. While $n = 1$ is a stringent condition, this requirement is in fact tight as discussed in Appendix E. To illustrate this theorem, we have conducted synthetic experiments. Let us first explain the setup used in Figure 1. We set $d = 3$ as the dimension, with each token having three entries $x = [x_1, x_2, x_3]$. We reserve the first two coordinates as key embeddings $k = [x_1, x_2, 0]$ by setting $W = \mathrm{diag}([1, 1, 0])$. This is what we display in our figures as token positions. Finally, in order to assign scores to the tokens we use the last coordinate by setting $v = [0, 0, 1]$. This way score becomes $Y \cdot v^\top x = Y \cdot x_3$, allowing us to assign any score (regardless of key embedding).

In Figure 1(a), the gray paths represent gradient descent trajectories from different initializations. The points $(0, 0)$ and $(1, 0)$ correspond to non-optimal tokens, while the point $(-0.1, 1)$ represents the optimal token. Notably, gradient descent iterates with various starting points converge towards the direction of the max-margin solution $p^{mm\star}$ (depicted by - - -). Moreover, as the iteration count $t$ increases, the inner product $\langle p(t)/\|p(t)\|, p^{mm\star}/\|p^{mm\star}\| \rangle$ consistently increases. Figure 1(c) also depicts the directional convergence of gradient descent from various initializations on multiple inputs, with the gray dotted line representing the separating hyperplane. These emphasize the gradual alignment between the evolving predictor and the max-margin solution throughout the optimization.

**Lemma 2** *Suppose for all $i \in [n]$ and $t \neq \mathtt{opt}_i$, $Y_i = 1$ and $\gamma_{it} < \gamma_{i\mathtt{opt}_i}$. Also assume $W \in \mathbb{R}^{d \times d}$ is full-rank. Then $p^{mm\star}$ exists – i.e. (ATT-SVM) is feasible for optimal indices $\alpha_i \leftarrow \mathtt{opt}_i$.*

## 2.2 Local convergence of the attention weights $p$

Theorem 2 on the global convergence of gradient descent serves as a prelude to the general behavior of the optimization. Once we relax Assumption B by allowing for arbitrary token scores, we will show that $p$ can converge (in direction) to a locally-optimal solution. However, this locally-optimal solution is still characterized in terms of (ATT-SVM) which separates *locally-optimal* tokens from the rest. Our theory builds on two new concepts: locally-optimal tokens and neighbors of these tokens.

**Definition 2 (SVM-Neighbor and Locally-Optimal Tokens)** *Fix token indices $\alpha = (\alpha_i)_{i=1}^n$ for which (ATT-SVM) is feasible to obtain $p^{mm} = p^{mm}(\alpha)$. Consider tokens $\mathcal{T}_i \subset [T]$ such that $(k_{i\alpha_i} - k_{it})^\top p^{mm} = 1$ for all $t \in \mathcal{T}_i$. We refer to $\mathcal{T}_i$ as SVM-neighbors of $k_{i\alpha_i}$. Additionally, tokens with indices $\alpha = (\alpha_i)_{i=1}^n$ are called locally-optimal if for all $i \in [n]$ and $t \in \mathcal{T}_i$ scores per Definition 1 obey $\gamma_{i\alpha_i} > \gamma_{it}$. Associated $p^{mm}$ is called a locally-optimal max-margin (LMM) direction.*

To provide a basis for discussing local convergence, we provide some preliminary definitions regarding cones. For a given $q$ and a scalar $\mu > 0$, we define $\mathrm{cone}_\mu(q)$ as the set of vectors $p \in \mathbb{R}^d$ such that the correlation coefficient between $p$ and $q$ is at least $1 - \mu$:

$$\mathrm{cone}_\mu(q) := \left\{ p \in \mathbb{R}^d \,\middle|\, \left\langle \frac{p}{\|p\|}, \frac{q}{\|q\|} \right\rangle \geq 1 - \mu \right\}. \quad (7)$$

Given $R > 0$, the intersection of $\mathrm{cone}_\mu(q)$ and the set $\{p \in \mathbb{R}^d | \|p\| \geq R\}$ is denoted as $C_{\mu,R}(q)$:

$$C_{\mu,R}(q) := \mathrm{cone}_\mu(q) \cap \left\{ p \in \mathbb{R}^d \,\middle|\, \|p\| \geq R \right\}. \quad (8)$$

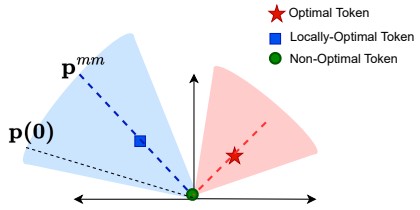

Figure 2: Gradient descent initialization $p(0)$ inside the cone containing the locally-optimal solution $p^{mm}$.

Next, we demonstrate the existence of parameters $\mu = \mu(\alpha) > 0$ and $R > 0$ such that when $R$ is sufficiently large, there are no stationary points within $C_{\mu,R}(\boldsymbol{p}^{mm})$. Further, the gradient descent initialized within $C_{\mu,R}(\boldsymbol{p}^{mm})$ converges in direction to $\boldsymbol{p}^{mm}/\|\boldsymbol{p}^{mm}\|$; refer to Figure 2 for a visualization.

**Theorem 3 (Local Convergence of Gradient Descent)** *Suppose Assumption A on the loss function $\ell$ holds and assume $\alpha = (\alpha_i)_{i=1}^n$ are indices of locally-optimal tokens per Definition 2. Then, there is a constant $\mu = \mu(\alpha) \in (0, 1)$ and $R > 0$ such that $C_{\mu,R}(\boldsymbol{p}^{mm})$ does not contain any stationary points. Further, for any starting point $\boldsymbol{p}(0) \in C_{\mu,R}(\boldsymbol{p}^{mm})$, gradient descent iterates $\boldsymbol{p}(t+1) = \boldsymbol{p}(t) - \eta\nabla\mathcal{L}(\boldsymbol{p}(t))$ on (ERM) with stepsize $\eta \leq 1/L_p$ satisfies $\lim_{t\to\infty} \|\boldsymbol{p}(t)\| = \infty$ and $\lim_{t\to\infty} \boldsymbol{p}(t)/\|\boldsymbol{p}(t)\| = \boldsymbol{p}^{mm}/\|\boldsymbol{p}^{mm}\|$.*

To further illustrate Theorem 3, we can consider Figure 1(b) where $n = 1$ and $T = 3$. In this figure, the point $(0, 0)$ represents the non-optimal tokens, while $(1, 0)$ represents the locally optimal token. Additionally, the gray paths represent the trajectories of gradient descent initiated from different points. By observing the figure, we can see that gradient descent, when properly initialized, converges towards the direction of $\boldsymbol{p}^{mm}$ (depicted by - - -). This direction of convergence effectively separates the locally optimal tokens $(1, 0)$ from the non-optimal token $(0, 0)$.

## 2.3 Regularization paths can only converge to locally-optimal max-margin directions

An important question arises regarding whether our definition of LMM (Definition 2) encompasses all possible convergence paths of the attention mechanism when $\|\boldsymbol{p}\| \to \infty$. To address this, we introduce the set of LMM directions as follows:

$$\mathcal{P}^{mm} := \left\{ \frac{\boldsymbol{p}^{mm}(\alpha)}{\|\boldsymbol{p}^{mm}(\alpha)\|} \,\middle|\, \alpha \text{ is locally-optimal per Definition 2} \right\}.$$

The following theorem establishes the tightness of these directions: It demonstrates that for any candidate $\boldsymbol{q} \notin \mathcal{P}^{mm}$, its local regularization path within an arbitrarily small neighborhood will provably not converge in the direction of $\boldsymbol{q}$.

**Theorem 4** *Fix $\boldsymbol{q} \notin \mathcal{P}^{mm}$ with unit $\ell_2$ norm. Assume that token scores are distinct (namely $\gamma_{it} \neq \gamma_{i\tau}$ for $t \neq \tau$) and key embeddings $\boldsymbol{k}_{it}$ are in general position (see Theorem 7). Fix arbitrary $\epsilon > 0, R_0 > 0$. Define the local regularization path of $\boldsymbol{q}$ as its $(\epsilon, R_0)$-conic neighborhood:*

$$\bar{\boldsymbol{p}}(R) = \underset{\boldsymbol{p}\in C_{\epsilon,R_0}(\boldsymbol{q}),\|\boldsymbol{p}\|\leq R}{\arg\min} \mathcal{L}(\boldsymbol{p}), \quad where \quad C_{\epsilon,R_0}(\boldsymbol{q}) = \mathsf{cone}_\epsilon(\boldsymbol{q}) \cap \left\{ \boldsymbol{p} \in \mathbb{R}^d \,\middle|\, \|\boldsymbol{p}\| \geq R_0 \right\}. \tag{9}$$

*Then, either $\lim_{R\to\infty} \|\bar{\boldsymbol{p}}(R)\| < \infty$ or $\lim_{R\to\infty} \bar{\boldsymbol{p}}(R)/\|\bar{\boldsymbol{p}}(R)\| \neq \boldsymbol{q}$. In both scenarios $\lim_{R\to\infty} \bar{\boldsymbol{p}}(R)/R \neq \boldsymbol{q}$.*

The result above nicely complements Theorem 3, which states that when gradient descent is initialized above a threshold ($\|\boldsymbol{p}(0)\| \geq R_0$) in an LMM direction, $\|\boldsymbol{p}(t)\|$ diverges but the direction converges to LMM. In contrast, Theorem 4 shows that regardless of how small the cone is (in terms of angle and norm lower bound $\|\boldsymbol{p}\| \geq R_0$), the optimal solution path will not converge along $\boldsymbol{q} \notin \mathcal{P}^{mm}$.

# 3 Joint Convergence of Head $v$ and Attention Weights $p$

In this section, we extend the preceding results to the general case of joint optimization of head $\boldsymbol{v}$ and attention weights $\boldsymbol{p}$ using a logistic loss function. To this aim, we focus on regularization path analysis, which involves solving (ERM) under ridge constraints and examining the solution trajectory as the constraints are relaxed.

**High-level intuition.** Since the prediction is linear as a function of $\boldsymbol{v}$, logistic regression in $\boldsymbol{v}$ can exhibit its own implicit bias to a max-margin solution. Concretely, define the attention features $\boldsymbol{x}_i^p = X_i^\top \mathbb{S}(K_i\boldsymbol{p})$ and define the dataset $\mathcal{D}^p = (Y_i, \boldsymbol{x}_i^p)_{i=1}^n$. If this dataset $\mathcal{D}^p$ is linearly separable, then fixing $\boldsymbol{p}$ and optimizing only $\boldsymbol{v}$ will converge in the direction of the standard max-margin classifier

$$\boldsymbol{v}^{mm} = \arg\min_{\boldsymbol{v}\in\mathbb{R}^d} \|\boldsymbol{v}\| \quad \text{subject to} \quad Y_i \cdot \boldsymbol{v}^\top \boldsymbol{r}_i \geq 1, \quad \text{for all} \quad 1 \leq i \leq n, \tag{SVM}$$

after setting inputs to the attention features $\boldsymbol{r}_i \leftarrow \boldsymbol{x}_i^p$ [22]. This motivates a clear question:

*Under what conditions, optimizing $\boldsymbol{v}, \boldsymbol{p}$ jointly will converge to their respective max-margin solutions?*

We study this question in two steps. Loosely speaking: **(1)** We will first assume that, at the optimal

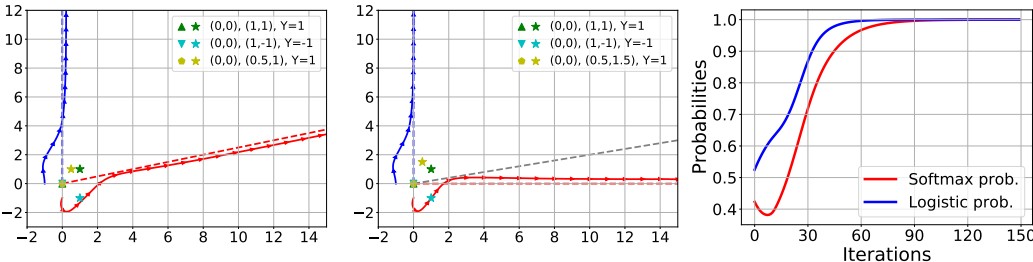

| (a) All inputs are support vectors | (b) (0.5,1.5) is not a support vector | (c) Probability evolutions in **(a)** |

Figure 3: **(a)** and **(b)** Joint convergence of attention weights $p$ (—>—) and classifier head $v$ (—>—) to max-margin directions. **(c)** Averaged softmax probability evolution of optimal tokens and logistic probability evolution of output in **(a)**.

tokens $x_{i\alpha_i}, i \in [n]$ selected by $p$, when solving (SVM) with $r_i \leftarrow x_{i\alpha_i}$, all of these tokens become support vectors of (SVM). **(2)** We will then relax this condition to uncover a more general implicit bias for $p$ that distinguish support vs non-support vectors. Throughout, we assume that the joint problem is separable and there exists $(v, p)$ asymptotically achieving zero training loss.

### 3.1 When all attention features are support vectors

In (SVM), define *label margin* to be $1/\|v^{mm}\|$. Our first insight in quantifying the joint implicit bias is that, **optimal tokens** admit a natural definition: *Those that maximize the downstream label margin when selected.* This is formalized below where we assume that: (1) Selecting the token indices $\alpha = (\alpha_i)_{i=1}^n$ from each input data achieves the largest label margin. (2) The optimality of the $\alpha$ choice is strict in the sense that mixing other tokens will shrink the label margin in (SVM).

**Assumption C (Optimal Tokens)** *Let $\Gamma > 0$ be the label margin when solving (SVM) with $r_i \leftarrow x_{i\alpha_i}$. There exists $\nu > 0$ such that for all $p$, solving (SVM) with $r_i \leftarrow x_i^p$ results in a label margin of at most $\Gamma - \nu \cdot \max_{i \in [n]}(1 - s_{i\alpha_i})$ where $s_i = \mathbb{S}(K_i p)$.*

**Example:** To gain intuition, let us fix $a \in \mathbb{R}^d$ and consider the dataset obeying $x_{i1} = Y_i \cdot a$ and $\|x_{it}\| < \|a\|$ for all $t \geq 2$ and all $i \in [n]$. For this dataset, we can choose $\alpha_i = 1$, $v^{mm} = a/\|a\|^2$, $\Gamma = 1/\|v^{mm}\| = \|a\|$ and $\nu = \|a\| - \sup_{i \in [n], t \geq 2} \|x_{it}\|$.

**Theorem 5** *Consider the ridge-constrained solutions $(v_r, p_R)$ of (ERM) defined as*

$$(v_r, p_R) = \arg\min_{\|v\| \leq r, \|p\| \leq R} \mathcal{L}(v, p).$$

*Suppose there are token indices $\alpha = (\alpha_i)_{i=1}^n$ for which $\|p^{mm}(\alpha)\|$ exists (ATT-SVM is feasible) and Assumption C holds for some $\Gamma, \nu > 0$. Then, $\lim_{R \to \infty} p_R/R = p^{mm}/\|p^{mm}\|$, where $p^{mm}$ is the solution of (ATT-SVM); and $\lim_{r \to \infty} v_r/r = v^{mm}/\|v^{mm}\|$, where $v^{mm}$ is the solution of (SVM) with $r_i = x_{i\alpha_i}$.*

As further discussion, consider Figure 3(a) where we set $n = 3, T = d = 2$ and $W =$ Identity. All three inputs share the point $(0, 0)$ which corresponds to their non-optimal tokens. The optimal tokens (denoted by $\star$) are all support vectors of the (SVM) since $v^{mm} = [0, 1]$ is the optimal classifier direction (depicted by - - -). Because of this, $p^{mm}$ will separate optimal $\star$ tokens from tokens at the $(0, 0)$ coordinate via (ATT-SVM) and its direction is dictated by yellow and teal colored $\star$s which are the support vectors.

### 3.2 General solution when selecting one token per input

Can we relax Assumption C, and if so, what is the resulting behavior? Consider the scenario where the optimal $p$ diverges to $\infty$ and ends up selecting one token per input. Suppose this $p$ selects some coordinates $\alpha = (\alpha_i)_{i=1}^n$. Let $\mathcal{S} \subset [n]$ be the set of indices where the associated token $x_{i\alpha_i}$ is a support vector when solving (SVM). Set $\bar{\mathcal{S}} = [n] - \mathcal{S}$. Our intuition is as follows: Even if we slightly perturb this $p$ choice and mix other tokens $t \neq \alpha_i$ over the input set $\bar{\mathcal{S}} \subset [n]$, since $\bar{\mathcal{S}}$ is not support vector for (SVM), we can preserve the label margin (by only preserving the support vectors $\mathcal{S}$). This means that $p$ may not have to enforce *max-margin* constraint over inputs $i \in \mathcal{S}$, instead, it suffices to just select

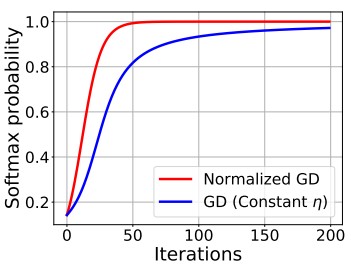
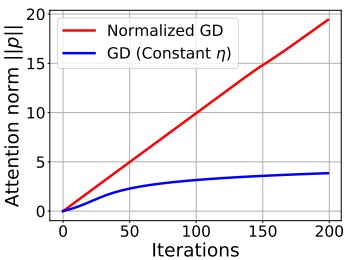
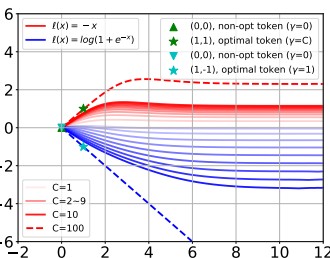

(a) Evolution of softmax probability     (b) Evolution of attention weights

Figure 4: Evolution of softmax probability and attention weights when training with normalized gradient descent or constant step size $\eta$ respectively.

Figure 5: Trajectories of $\boldsymbol{p}$ with different loss functions and scores in Theorem 2.

these tokens (asymptotically). This results in the following **relaxed SVM** problem:

$$\boldsymbol{p}^{relax} = \arg\min_{\boldsymbol{p}} \|\boldsymbol{p}\| \quad \text{such that} \quad \boldsymbol{p}^\top(\boldsymbol{k}_{i\alpha_i} - \boldsymbol{k}_{it}) \geq \begin{cases} 1 & \text{for all} \quad t \neq \alpha_i, \ i \in \mathcal{S} \\ 0 & \text{for all} \quad t \neq \alpha_i, \ i \in \bar{\mathcal{S}} \end{cases}. \tag{10}$$

Here, $\boldsymbol{p}^\top(\boldsymbol{k}_{i\alpha_i} - \boldsymbol{k}_{it}) \geq 0$ corresponds to the *selection* idea. Building on this intuition, the following theorem captures the generalized behavior of the joint regularization path.

**Theorem 6** *Consider the same* (ERM) *problem as discussed in Theorem 5. Suppose* $\mathbb{S}(\boldsymbol{K}_i\boldsymbol{p}_R)_{\alpha_i} \to 1$, *i.e., the tokens* $(\alpha_i)_{i=1}^n$ *are asymptotically selected. Let* $\boldsymbol{v}^{mm}$ *be the solution of* (SVM) *with* $\boldsymbol{r}_i = \boldsymbol{x}_{i\alpha_i}$ *and* $\mathcal{S}$ *be its set of support vector indices. Suppose Assumption C holds over* $\mathcal{S}$ *i.e. having* $s_{i\alpha_i} < 1$ *shrinks the margin when* (SVM) *is only solved over* $\mathcal{S} \subset [n]$. *Then,* $\lim_{r\to\infty} \boldsymbol{v}_r/r = \boldsymbol{v}^{mm}/\|\boldsymbol{v}^{mm}\|$ *and* $\lim_{R\to\infty} \boldsymbol{p}_R/R = \boldsymbol{p}^{relax}/\|\boldsymbol{p}^{relax}\|$, *where* $\boldsymbol{p}^{relax}$ *is the solution of* (10) *with* $(\alpha_i)_{i=1}^n$ *choices.*

To illustrate this numerically, consider Figure 3(b) which modifies Figure 3(a) by pushing the yellow ★ to the northern position (0.5, 1.5). We still have $\boldsymbol{v}^{mm} = [0, 1]$ however the yellow ★ is no longer a support vector of (SVM). Thus, $\boldsymbol{p}$ solves the relaxed problem (10) which separates green and teal ★'s by enforcing the max-margin constraint on $\boldsymbol{p}$ (which is the red direction). Instead, yellow ★ only needs to achieve positive correlation with $\boldsymbol{p}$ (unlike Figure 3(a) where it dictates the direction). We also display the direction of $\boldsymbol{p}^{mm}$ using a gray dashed line.

We further investigate the evolution of softmax and logistic output probabilities throughout the training process of Figure 3(a), and the results are illustrated in Figure 3(c). The averaged softmax probability of optimal tokens is represented by the red curve and is calculated as $\frac{1}{n}\sum_{i=1}^n \max_{t\in[T]} \mathbb{S}(\boldsymbol{K}_i\boldsymbol{p})_t$. An achievement of 1 for this probability indicates that the attention mechanism successfully selects the optimal tokens. On the other hand, the logistic probability of the output is represented by the blue curve and is determined by $1/n\sum_{i=1}^n 1/(1 + e^{-Y_i \cdot f(X_i)})$. This probability also reaches a value of 1, suggesting that the inputs are correctly classified.

## 4  Experiments

**Sparsity of softmax and evolution of attention weights.** It is well known that, in practice, attention maps often exhibit sparsity and highlight salient tokens that aid inference. Our results provide a formal explanation of this when tokens are separable: Since attention selects a locally-optimal token within the input sequence and suppresses the rest, the associated attention map $\mathbb{S}(\boldsymbol{X}\boldsymbol{p})$ will (eventually) be a sparse vector. Additionally, the sparsity should arise in tandem with the increasing norm of attention weights. We provide empirical evidence to support these findings.

*Synthetic experiments.* Figures 4(a) and 4(b) show the evolution of the largest softmax probability and attention weights over time when using either normalized gradient or a fixed stepsize $\eta$ for training. The dataset model follows Figure 1(c). The softmax probability shown in Figure 4(a) is defined as $\frac{1}{n}\sum_{i=1}^n \max_{t\in[T]} \mathbb{S}(\boldsymbol{K}_i\boldsymbol{p})_t$. When this average probability reaches the value of 1, it means attention selects only a single token per input. The attention norm in Figure 4(b), is simply equal to $\|\boldsymbol{p}\|$.

The red curves in both figures represent the normalized gradient method, which updates the model parameters $\boldsymbol{p}$ using $\boldsymbol{p}(t + 1) = \boldsymbol{p}(t) - \eta\nabla\mathcal{L}(\boldsymbol{p}(t))/\|\nabla\mathcal{L}(\boldsymbol{p}(t))\|$ with $\eta = 0.1$. The blue curves correspond

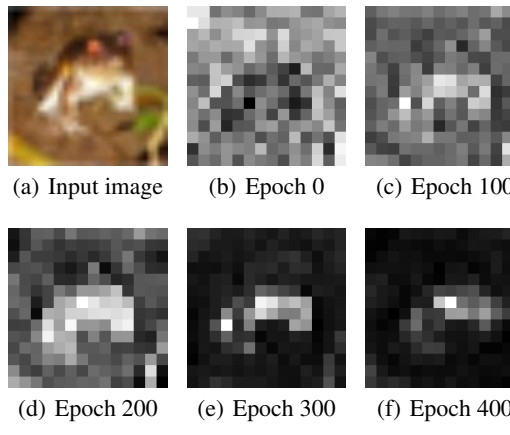

(a) Input image      (b) Epoch 0      (c) Epoch 100

(d) Epoch 200      (e) Epoch 300      (f) Epoch 400

Figure 6: Illustration of the progressive change in attention weights of the [CLS] token during training in the transformer model, using a specific input image shown in Figure 6(a).

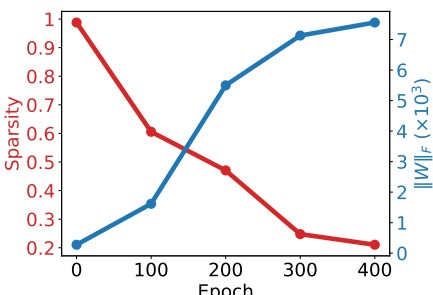

Figure 7: Red curve is the sparsity level $\widehat{\mathrm{nnz}}(s)/T$ of the average attention map which takes values on [0,1]. A sparser vector implies that few key tokens receive significantly higher attention, while the majority of the tokens receive minimal attention. Blue curve is the Frobenius norm of attention weights $\|W\|_F$ of the final layer. We display their evolutions over epochs.

to gradient descent with constant learning rate given by $p(t + 1) = p(t) - \eta \nabla \mathcal{L}(p(t))$ with $\eta = 1$. Observe that the normalized gradient method achieves a softmax probability of 1 quicker as vanilla GD suffers from vanishing gradients. This is visible in Figure 4(b) where blue norm curve levels off.

*Real experiments.* To study softmax sparsity and the evolution of attention weights throughout training, we train a vision transformer (ViT-base) model [23] from scratch, utilizing the CIFAR-10 dataset [24] for 400 epochs with fixed learning rate $3 \times 10^{-3}$. ViT tokenizes an image into $16 \times 16$ patches, thus, its softmax attention maps can be easily visualized. We examine the average attention map – associated with the [CLS] token – computed from all 12 attention heads within the model. Figure 6 provides a visual representation of the resulting attention weights ($16 \times 16$ grids) corresponding to the original patch locations within the image. During the initial epochs of training, the attention weights are randomly distributed and exhibit a dense pattern. However, as the training progresses, the attention map gradually becomes sparser and the attention mechanism begins to concentrate on fewer salient patches within the image that possess distinct features that aid classification. This illustrates the evolution of attention from a random initial state to a more focused and sparse representation. These salient patches highlighted by attention conceptually corresponds to the optimal tokens within our theory.

We quantify the sparsity of the attention map via a soft-sparsity measure, denoted by $\widehat{\mathrm{nnz}}(s)$ where $s$ is the softmax probability vector. The soft-sparsity is computed as the ratio of the $\ell_1$–norm to the squared $\ell_2$–norm, defined as $\widehat{\mathrm{nnz}}(s) = \|s\|_1/\|s\|^2$. $\widehat{\mathrm{nnz}}(s)$ takes values between 1 to $T = 256$ and a smaller value indicates a sparser vector. Also note that $\|s\|_1 = \sum_{t=1}^{T} s_t = 1$. Together with sparsity, Figure 7 also displays the Frobenius norm of the combined key-query matrix $W$ of the last attention layer over epochs. The theory suggests that the increase in sparsity is associated with the growth of attention weights – which converge directionally. The results in Figure 7 align with the theory, demonstrating the progressive sparsification of the attention map as $\|W\|_F$ grows.

**Transient optimization dynamics and the influence of the loss function.** Theorem 2 shows that the asymptotic direction of gradient descent is determined by $p^{mm\star}$. However, it is worth noting that transient dynamics can exhibit bias towards certain input examples and their associated optimal tokens. We illustrate this idea in Fig 5(a), which displays the trajectories of the gradients for different scores and loss functions. We consider two optimal tokens ($\star$) with scores $\gamma_1 = 1$ and $\gamma_2 = C$, where $C$ varies. For our analysis, we examine the correlation loss $\ell(x) = -x$ and the logistic loss $\ell(x) = \log(1 + e^{-x})$.

In essence, as $C$ increases, we can observe that the correlation loss $\ell(x) = -x$ exhibits a bias towards the token with a high score, while the logistic loss is biased towards the token with a low score. The underlying reason for this behavior can be observed from the gradients of individual inputs: $\nabla \mathcal{L}_i(p) = \ell_i' \cdot K_i^\top \mathbb{S}'(Xp)Xv$, where $\mathbb{S}'(\cdot)$ represents the derivative of the softmax function and $\ell_i' := \ell'(Y_i \cdot v^\top X_i^\top \mathbb{S}(X_i p))$. Assuming that $p$ (approximately) selects the optimal tokens, this

simplifies to $\ell_i' \approx \ell'(\gamma_i)$ and $\|\nabla \mathcal{L}_i(\boldsymbol{p})\| \propto |\ell'(\gamma_i)| \cdot \gamma_i$. With the correlation loss, $|\ell'| = 1$, resulting in $\|\nabla \mathcal{L}_i(\boldsymbol{p})\| \propto \gamma_i$, meaning that a larger score induces a larger gradient. On the other hand, the logistic loss behaves similarly to the exponential loss under separable data, i.e., $|\ell'| = e^{-x}/(1 + e^{-x}) \approx e^{-x}$. Consequently, $\|\nabla \mathcal{L}_i(\boldsymbol{p})\| \propto \gamma_i e^{-\gamma_i} \approx e^{-\gamma_i}$, indicating that a smaller score leads to a larger gradient. These observations explain the empirical behavior we observe.

## 5 Related Work

**Implicit Regularization.** The implicit bias of gradient descent in classification tasks involving separable data has been extensively examined by [22, 25, 26, 27, 28, 29]. These works typically use logistic loss or, more generally, exponentially-tailed losses to make connections to margin maximization. These results are also extended to non-separable data by [30, 31, 21]. Furthermore, there have been notable investigations into the implicit bias in regression problems/losses utilizing techniques such as mirror descent [32, 25, 33, 34, 35, 36]. In addition, several papers have explored the implicit bias of stochastic gradient descent [37, 38, 39, 40, 41, 42], as well as adaptive and momentum-based methods [43, 44, 45, 46]. Although there are similarities between our optimization approach for $\boldsymbol{v}$ and existing works, the optimization of $\boldsymbol{p}$ stands out as significantly different. Firstly, our optimization problem is nonconvex, introducing new challenges and complexities. Secondly, it necessitates the introduction of novel concepts such as locally-optimal tokens and requires a fresh analysis specifically tailored to the cones surrounding them.

**Attention Mechanism.** Transformers, introduced by [6], revolutionized the field of NLP and machine translation, with earlier works on self-attention by [47, 48, 49, 50]. Self-attention differs from traditional models like MLPs and CNNs by leveraging global interactions for feature representations, showing exceptional empirical performance. However, the underlying mechanisms and learning processes of the attention layer remain unknown. Recent studies such as [51, 52, 53, 54, 23] have focused on specific aspects like representing sparse functions, convex-relaxations, and expressive power. In contrast to our nonconvex (ERM), [52] studies self-attention with linear activation instead of softmax, while [53] approximates softmax using a linear operation with unit simplex constraints. Their main objective is to derive convex reformulations for ERM-based training problem. [55, 56] have developed initial results to characterize the optimization and generalization dynamics of attention. [17] is another closely related work where the authors analyze the same attention model (ERM) as us. Specifically, they jointly optimize $\boldsymbol{v}, \boldsymbol{p}$ for three gradient iterations for a contextual dataset model. However, all of these works make stringent assumptions on the data, namely, tokens are tightly clusterable or can be clearly split into clear relevant and irrelevant sets. Additionally [56] requires assumptions on initialization and [55] considers a simplified attention structure where the attention matrix is not directly parameterized with respect to the input. Our work links attention models to hard-margin SVM problems and pioneers the study of gradient descent's implicit bias in these models.

## 6 Discussion

We have provided a thorough optimization-theoretic characterization of the fundamental attention model $f(\boldsymbol{X}) = \boldsymbol{v}^\top \boldsymbol{X}^\top \mathbb{S}(\boldsymbol{X}\boldsymbol{W}\boldsymbol{p})$ by formally connecting it to max-margin problems. We first established the convergence of gradient descent on $\boldsymbol{p}$ (or equivalently $\boldsymbol{W}$) in isolation. We also explored joint convergence of $(\boldsymbol{v}, \boldsymbol{p})$ via regularization path which revealed surprising implicit biases such as (10). These findings motivate several exciting avenues for future research. An immediate open problem is characterizing the (local) convergence of gradient descent for joint optimization of $(\boldsymbol{v}, \boldsymbol{p})$. Another major direction is to extend similar analysis to study self-attention layer (4) or to allow for multiple tunable tokens (where $\boldsymbol{p}$ becomes a matrix). Either setting will enrich the problem by allowing the attention to discover multiple hyperplanes to separate tokens. While our convergence guarantees apply when tokens are separable, it would be interesting to characterize the non-separable geometry by leveraging results developed for logistic regression analysis [31, 22]. Ideas from such earlier results can also be useful for characterizing the non-asymptotic/transient dynamics of how gradient descent aligns with the max-margin direction. Overall, we believe that max-margin token selection is a fundamental characteristic of attention mechanism and the theory developed in this work lays the groundwork of these future extensions.

## Acknowledgements

This work was supported by the NSF grants CCF-2046816 and CCF-2212426, Google Research Scholar award, and Army Research Office grant W911NF2110312.

The authors express their gratitude for the valuable feedback provided by the anonymous reviewers and Christos Thrampoulidis, which has significantly improved this paper.

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

**Roadmap.** The appendix is organized as follows: Section A provides basic facts about the training risk. Section B presents the proof of local and global gradient descent and regularized path for learning $\boldsymbol{p} \in \mathbb{R}^d$ with a fixed $\boldsymbol{v} \in \mathbb{R}^d$ choice. Section C provides the proof of regularized path applied to the general case of joint optimization of head $\boldsymbol{v}$ and attention weights $\boldsymbol{p}$ using a logistic loss function. Section D presents the regularized path applied to a more general model $f(\boldsymbol{X}) = \psi(\boldsymbol{X}^\top \mathbb{S}(\boldsymbol{X}\boldsymbol{W}^\top \boldsymbol{p}))$ with a nonlinear head $\psi$. Section E provides implementation details. Finally, Section F discusses additional related work on implicit bias and self-attention.

# Table of Contents

# A  Addendum to Section 1

## A.1  Preliminaries on the Training Risk

By our assumption $\psi : \mathbb{R}^d \to \mathbb{R}$ and $\ell : \mathbb{R} \to \mathbb{R}$ are differentiable functions. Recall the objective

$$\mathcal{L}(\boldsymbol{p}, \boldsymbol{W}) = \frac{1}{n} \sum_{i=1}^{n} \ell\left(Y_i \cdot \psi(\boldsymbol{X}_i^\top \mathbb{S}(\boldsymbol{K}_i \boldsymbol{p}))\right) \tag{11}$$

with the generic prediction model $\psi(\boldsymbol{X}^\top \mathbb{S}(\boldsymbol{K}\boldsymbol{p}))$ and $\boldsymbol{K} = \boldsymbol{X}\boldsymbol{W}^\top$.

Here, we write down the gradients of $W$ and $p$ in (11) to highlight the connection. Set $q := W^\top p$, $z\{X\} := X^\top \mathbb{S}(Kp)$, and $a\{X\} := Kp$. Given $X$ and using $K = XW^\top$, we have that

$$\nabla_q \psi(p, W) = X^\top \mathbb{S}'(a\{X\})X \cdot \psi'(z\{X\}), \tag{12a}$$

$$\nabla_p \psi(p, W) = W \nabla_q \psi(p, W), \tag{12b}$$

$$\nabla_W \psi(p, W) = p \nabla_q^\top \psi(p, W), \tag{12c}$$

where

$$\mathbb{S}'(a\{X\}) = \mathrm{diag}(\mathbb{S}(a\{X\})) - \mathbb{S}(a\{X\})\mathbb{S}(a\{X\})^\top \in \mathbb{R}^{T \times T}.$$

Setting $\psi(z) = v^\top z$ for linear head, we obtain

$$\nabla_q \psi(p, W) = X^\top \mathbb{S}'(a\{X\})\gamma, \tag{13a}$$

$$\nabla_p \psi(p, W) = W \nabla_q \psi(p, W) = K^\top \mathbb{S}'(a\{X\})\gamma, \tag{13b}$$

$$\nabla_W \psi(p, W) = p \nabla_q^\top \psi(p, W) = p v^\top X^\top \mathbb{S}'(a\{X\})X. \tag{13c}$$

Recalling (12b) and (12c), and defining $\ell_i' := \ell'(Y_i \cdot \psi(z\{X_i\})) \in \mathbb{R}$, we have that

$$\nabla_p \mathcal{L}(p, W) = \frac{1}{n} \sum_{i=1}^n \ell_i' \cdot Y_i \cdot W \nabla_q \psi(p, W), \tag{14a}$$

$$\nabla_W \mathcal{L}(p, W) = \frac{1}{n} \sum_{i=1}^n \ell_i' \cdot Y_i \cdot p \nabla_q^\top \psi(p, W). \tag{14b}$$

Setting $\psi(z) = v^\top z$ for linear head and $\gamma_i = Y_i \cdot X_i v$, we obtain

$$\nabla_p \mathcal{L}(p, W) = \frac{1}{n} \sum_{i=1}^n \ell_i' \cdot K_i^\top \mathbb{S}'(a\{X_i\})\gamma_i, \tag{15a}$$

$$\nabla_W \mathcal{L}(p, W) = p \left( \frac{1}{n} \sum_{i=1}^n \ell_i' \cdot \gamma_i^\top \mathbb{S}'(a\{X_i\})X_i \right). \tag{15b}$$

**Lemma 3 (Key Lemma)** *For any $p, q \in \mathbb{R}^d$, let $a = Kq$, $s = \mathbb{S}(Kp)$, and $\gamma = Xv$. Set*

$$\Gamma = \sup_{t,\tau \in [T]} |\gamma_t - \gamma_\tau| \quad \text{and} \quad A = \sup_{t \in [T]} \|k_t\| \cdot \|q\|.$$

*We have that*

$$\left| a^\top \mathrm{diag}(s)\gamma - a^\top ss^\top \gamma - \sum_{t \geq 2}^T (a_1 - a_t)s_t(\gamma_1 - \gamma_t) \right| \leq 2\Gamma A(1 - s_1)^2.$$

**Proof.** Set $\bar{\gamma} = \sum_{t=1}^T \gamma_t s_t$. We have

$$\gamma_1 - \bar{\gamma} = \sum_{t \geq 2}^T (\gamma_1 - \gamma_t)s_t, \quad \text{and} \quad |\gamma_1 - \bar{\gamma}| \leq \Gamma(1 - s_1).$$

Then,

$$a^\top \mathrm{diag}(s)\gamma - a^\top ss^\top \gamma = \sum_{t=1}^T a_t \gamma_t s_t - \sum_{t=1}^T a_t s_t \sum_{t=1}^T \gamma_t s_t$$

$$= a_1 s_1(\gamma_1 - \bar{\gamma}) - \sum_{t \geq 2}^T a_t s_t(\bar{\gamma} - \gamma_t). \tag{16}$$

Since

$$\left| \sum_{t \geq 2}^T a_t s_t(\bar{\gamma} - \gamma_t) - \sum_{t \geq 2}^T a_t s_t(\gamma_1 - \gamma_t) \right| \leq A\Gamma(1 - s_1)^2,$$

we obtain[2]

$$a^\top \text{diag}(s)\gamma - a^\top ss^\top \gamma = a_1 s_1(\gamma_1 - \bar{\gamma}) - \sum_{t \geq 2}^{T} a_t s_t(\gamma_1 - \gamma_t) \pm A\Gamma(1 - s_1)^2$$

$$= a_1 s_1 \sum_{t \geq 2}^{T}(\gamma_1 - \gamma_t)s_t - \sum_{t \geq 2}^{T} a_t s_t(\gamma_1 - \gamma_t) \pm A\Gamma(1 - s_1)^2$$

$$= \sum_{t \geq 2}^{T}(a_1 s_1 - a_t)s_t(\gamma_1 - \gamma_t) \pm A\Gamma(1 - s_1)^2$$

$$= \sum_{t \geq 2}^{T}(a_1 - a_t)s_t(\gamma_1 - \gamma_t) \pm 2A\Gamma(1 - s_1)^2.$$

Here, $\pm$ on the right handside uses the fact that

$$\left| \sum_{t \geq 2}^{T}(a_1 s_1 - a_1)s_t(\gamma_1 - \gamma_t) \right| \leq (1 - s_1)A\Gamma \sum_{t \geq 2}^{T} s_t = (1 - s_1)^2 A\Gamma.$$

∎

## A.2 Proof of Lemma 1

**Proof.** Let us prove the result for a general step size sequence $(\eta_t)_{t \geq 0}$. On the same training data $(Y_i, X_i)_{i=1}^n$, recall the objectives $\tilde{\mathcal{L}}(p) = \frac{1}{n} \sum_{i=1}^n \ell(Y_i \cdot \psi(X_i^\top \mathbb{S}(X_i p)))$ and $\mathcal{L}(W) = \frac{1}{n} \sum_{i=1}^n \ell(Y_i \cdot \psi(X_i^\top \mathbb{S}(X_i W^\top u)))$. Suppose claim is true till iteration $t$. For iteration $t + 1$, using $W(t)^\top u = p(t)$, define and observe that

$$S_i = \mathbb{S}'(X_i W(t)^\top u) = \mathbb{S}'(X_i p(t)),$$

$$s_i = \mathbb{S}(X_i W(t)^\top u) = \mathbb{S}(X_i p(t)),$$

$$z\{X_i\} = X_i^\top \mathbb{S}(X_i p(t)) = X_i^\top \mathbb{S}(X_i W(t)^\top u),$$

for all $i \in [n]$.

Thus, using (14), we have that

$$\nabla \tilde{\mathcal{L}}(p(t)) = \frac{1}{n} \sum_{i=1}^n \ell_i' \cdot Y_i \cdot X_i^\top S_i X_i \cdot \psi'(z\{X_i\}),$$

$$\nabla \mathcal{L}(W(t)) = u \left( \frac{1}{n} \sum_{i=1}^n \ell_i' \cdot Y_i \cdot X_i^\top S_i X_i \cdot \psi'(z\{X_i\}) \right)^\top.$$

Consequently, we found that gradient is rank-1 with left singular space equal to $u$, i.e.,

$$\nabla \mathcal{L}(W(t)) = u \nabla^\top \tilde{\mathcal{L}}(p(t)).$$

Since $W(t)$'s left singular space is guaranteed to be in $u$ (including $W(0)$ by initialization), we only need to study the right singular vector. Using the induction till $t$, this yields

$$W(t + 1)^\top u = W(t)^\top u - \eta_t \|u\|^{-2} \nabla^\top \mathcal{L}(W(t))u$$

$$= p(t) - \eta_t \|u\|^{-2} u^\top u \nabla \tilde{\mathcal{L}}(p(t))$$

$$= p(t + 1).$$

This concludes the induction. ∎

---

[2]For simplicity, we use $\pm$ on the right hand side to denote the upper and lower bounds.

# B  Addendum to Section 2

## B.1  Descent and Gradient Correlation Conditions

The lemma below identifies conditions under which $\boldsymbol{p}^{mm\star}$ is a global descent direction for $\mathcal{L}(\boldsymbol{p})$.

**Lemma 4** *Suppose $\ell(\cdot)$ is a strictly decreasing differentiate loss function and Assumption B holds. Then, for all $\boldsymbol{p} \in \mathbb{R}^d$, the training loss (ERM) obeys $\langle \nabla \mathcal{L}(\boldsymbol{p}), \boldsymbol{p}^{mm\star} \rangle < 0$.*

**Proof.**  Set

$$\boldsymbol{\gamma}_i = Y_i \cdot \boldsymbol{X}_i \boldsymbol{v}, \quad \boldsymbol{a}_i = \boldsymbol{K}_i \boldsymbol{p}, \quad \bar{\boldsymbol{a}}_i = \boldsymbol{K}_i \boldsymbol{p}^{mm\star}, \quad \text{and} \quad \ell'_i = \ell'\left(\boldsymbol{\gamma}_i^\top \mathbb{S}(\boldsymbol{K}_i \boldsymbol{p})\right). \tag{17}$$

Let us recall the gradient evaluated at $\boldsymbol{p}$ which is given by

$$\nabla \mathcal{L}(\boldsymbol{p}) = \frac{1}{n} \sum_{i=1}^{n} \ell'_i \cdot \boldsymbol{K}_i^\top \mathbb{S}'(\boldsymbol{a}_i) \boldsymbol{\gamma}_i. \tag{18}$$

This implies that

$$\left\langle \nabla \mathcal{L}(\boldsymbol{p}), \boldsymbol{p}^{mm\star} \right\rangle = \frac{1}{n} \sum_{i=1}^{n} \ell'_i \cdot \langle \bar{\boldsymbol{a}}_i, \mathbb{S}'(\boldsymbol{a}_i) \boldsymbol{\gamma}_i \rangle. \tag{19}$$

To proceed, we will prove that individual summands are all strictly negative. To show that, without losing generality, let us focus on the first input and drop the subscript $i$ for cleaner notation. This yields

$$\langle \bar{\boldsymbol{a}}, \mathbb{S}'(\boldsymbol{a}) \boldsymbol{\gamma} \rangle = \bar{\boldsymbol{a}}^\top \operatorname{diag}(\mathbb{S}(\boldsymbol{a})) \boldsymbol{\gamma} - \bar{\boldsymbol{a}}^\top \mathbb{S}(\boldsymbol{a}) \mathbb{S}(\boldsymbol{a})^\top \boldsymbol{\gamma}. \tag{20}$$

Without losing generality, assume optimal token is the first one and $\boldsymbol{\gamma}_t$ is a constant for all $t \geq 2$.

To proceed, we will prove the following: Suppose $\gamma = \boldsymbol{\gamma}_{t \geq 2}$ is constant, $\boldsymbol{\gamma}_1, \bar{\boldsymbol{a}}_1$ are the largest indices of $\boldsymbol{\gamma}, \bar{\boldsymbol{a}}$. Then, for any $\boldsymbol{s}$ obeying $\sum_{t \in [T]} s_t = 1, s_t \geq 0$, we have that $\bar{\boldsymbol{a}}^\top \operatorname{diag}(\boldsymbol{s}) \boldsymbol{\gamma} - \bar{\boldsymbol{a}}^\top \boldsymbol{s} \boldsymbol{s}^\top \boldsymbol{\gamma} > 0$. To see this, we write

$$
\begin{aligned}
\bar{\boldsymbol{a}}^\top \operatorname{diag}(\boldsymbol{s}) \boldsymbol{\gamma} - \bar{\boldsymbol{a}}^\top \boldsymbol{s} \boldsymbol{s}^\top \boldsymbol{\gamma} &= \sum_{t=1}^{T} \bar{\boldsymbol{a}}_t \boldsymbol{\gamma}_t s_t - \sum_{t=1}^{T} \bar{\boldsymbol{a}}_t s_t \sum_{t=1}^{T} \boldsymbol{\gamma}_t s_t \\
&= \left( \bar{\boldsymbol{a}}_1 \boldsymbol{\gamma}_1 s_1 + \gamma \sum_{t \geq 2} \bar{\boldsymbol{a}}_t s_t \right) - \left( \boldsymbol{\gamma}_1 s_1 + \gamma(1 - s_1) \right) \left( \bar{\boldsymbol{a}}_1 s_1 + \sum_{t \geq 2} \bar{\boldsymbol{a}}_t s_t \right) \\
&= \bar{\boldsymbol{a}}_1 (\boldsymbol{\gamma}_1 - \gamma) s_1 (1 - s_1) + \left( \gamma - (\boldsymbol{\gamma}_1 s_1 + \gamma(1 - s_1)) \right) \sum_{t \geq 2} \bar{\boldsymbol{a}}_t s_t \\
&= \bar{\boldsymbol{a}}_1 (\boldsymbol{\gamma}_1 - \gamma) s_1 (1 - s_1) - (\boldsymbol{\gamma}_1 - \gamma) s_1 \sum_{t \geq 2} \bar{\boldsymbol{a}}_t s_t \\
&= (\boldsymbol{\gamma}_1 - \gamma)(1 - s_1) s_1 \left[ \bar{\boldsymbol{a}}_1 - \frac{\sum_{t \geq 2}^{T} \bar{\boldsymbol{a}}_t s_t}{\sum_{t \geq 2}^{T} s_t} \right].
\end{aligned}
\tag{21}
$$

To proceed, let $\boldsymbol{\gamma}_{gap} = \boldsymbol{\gamma}_1 - \gamma$ and $\boldsymbol{a}_{gap} = \bar{\boldsymbol{a}}_1 - \max_{t \geq 2} \boldsymbol{a}_t$. With these, we obtain

$$\bar{\boldsymbol{a}}^\top \operatorname{diag}(\boldsymbol{s}) \boldsymbol{\gamma} - \bar{\boldsymbol{a}}^\top \boldsymbol{s} \boldsymbol{s}^\top \boldsymbol{\gamma} \geq \boldsymbol{a}_{gap} \boldsymbol{\gamma}_{gap} s_1 (1 - s_1). \tag{22}$$

Note that

$$a_{gap}^i \geq \inf_{t \neq \mathrm{opt}_i} (\boldsymbol{k}_{i\mathrm{opt}_i} - \boldsymbol{k}_{it})^\top \boldsymbol{p}^{mm\star} \geq 1,$$

$$\gamma_{gap}^i = \inf_{t \neq \mathrm{opt}_i} \boldsymbol{\gamma}_{i\mathrm{opt}_i} - \boldsymbol{\gamma}_{it} > 0,$$

$$s_{i1}(1 - s_{i1}) > 0.$$

On the other hand, by our assumption $\ell'_i < 0$. Hence, infimum'ing (22) over all inputs, multiplying by $\ell'_i$ and using (19) give the desired result. ∎

**Lemma 5 (Gradient Correlation Conditions)** *Consider $n = 1$ and let $\boldsymbol{p}^{mm} = \boldsymbol{p}^{mm\star}$ be* (ATT-SVM) *solution separating $\alpha = \mathrm{opt}$ from remaining tokens of input $\boldsymbol{X}$. Suppose $\ell(\cdot)$ is a strictly decreasing differentiate loss function and Assumption B holds. For any choice of $\pi > 0$, there exists $R := R_\pi$ such that, for any $\boldsymbol{p}$ with $\|\boldsymbol{p}\| \geq R$, we have*

$$\left\langle \nabla \mathcal{L}(\boldsymbol{p}), \frac{\boldsymbol{p}}{\|\boldsymbol{p}\|} \right\rangle \geq (1 + \pi) \left\langle \nabla \mathcal{L}(\boldsymbol{p}), \frac{\boldsymbol{p}^{mm}}{\|\boldsymbol{p}^{mm}\|} \right\rangle.$$

*Above, observe that as $R \to \infty$, we eventually get to set $\pi = 0$.*

**Proof.** The proof is similar to Lemma 4 at a high-level. However, we also need to account for the impact of $\boldsymbol{p}$ besides $\boldsymbol{p}^{mm}$ in the gradient correlation. The main goal is showing that $\boldsymbol{p}^{mm}$ is the near-optimal descent direction, thus, $\boldsymbol{p}$ cannot significantly outperform it.

To proceed, let $\bar{\boldsymbol{p}} = \|\boldsymbol{p}^{mm}\| \boldsymbol{p}/\|\boldsymbol{p}\|$, $M = \sup_t \|\boldsymbol{k}_t\|$, $\Theta = 1/\|\boldsymbol{p}^{mm}\|$, $\boldsymbol{s} = \mathbb{S}(\boldsymbol{K}\boldsymbol{p})$, $\boldsymbol{a} = \boldsymbol{K}\bar{\boldsymbol{p}}$, $\bar{\boldsymbol{a}} = \boldsymbol{K}\boldsymbol{p}^{mm}$. Without losing generality assume $\mathrm{opt} = 1$. Set $\gamma = \gamma_{t\geq 2}$. Repeating the proof of Lemma 4 yields

$$\langle \nabla \mathcal{L}(\boldsymbol{p}), \boldsymbol{p}^{mm} \rangle = \ell' \cdot (\gamma_1 - \gamma)(1 - s_1)s_1 \left[ \bar{a}_1 - \frac{\sum_{t\geq 2}^T \bar{a}_t s_t}{\sum_{t\geq 2}^T s_t} \right],$$

$$\langle \nabla \mathcal{L}(\boldsymbol{p}), \bar{\boldsymbol{p}} \rangle = \ell' \cdot (\gamma_1 - \gamma)(1 - s_1)s_1 \left[ a_1 - \frac{\sum_{t\geq 2}^T a_t s_t}{\sum_{t\geq 2}^T s_t} \right].$$

Given $\pi$, for sufficiently large $R$, we wish to show that

$$a_1 - \frac{\sum_{t\geq 2}^T a_t s_t}{\sum_{t\geq 2}^T s_t} \leq (1 + \pi) \cdot \left[ \bar{a}_1 - \frac{\sum_{t\geq 2}^T \bar{a}_t s_t}{\sum_{t\geq 2}^T s_t} \right]. \tag{23}$$

We consider two scenarios.

**Scenario 1:** $\|\bar{\boldsymbol{p}} - \boldsymbol{p}^{mm}\| \leq \epsilon := \pi/(2M)$. In this scenario, for any token, we find that
$$|a_t - \bar{a}_t| = |\boldsymbol{k}_t^\top (\bar{\boldsymbol{p}} - \boldsymbol{p}^{mm})| \leq M\|\bar{\boldsymbol{p}} - \boldsymbol{p}^{mm}\| \leq M\epsilon.$$

Consequently, we obtain

$$\bar{a}_1 - \frac{\sum_{t\geq 2}^T \bar{a}_t s_t}{\sum_{t\geq 2}^T s_t} \geq a_1 - \frac{\sum_{t\geq 2}^T a_t s_t}{\sum_{t\geq 2}^T s_t} - 2M\epsilon = a_1 - \frac{\sum_{t\geq 2}^T a_t s_t}{\sum_{t\geq 2}^T s_t} - \pi.$$

Also noticing $\bar{a}_1 - \frac{\sum_{t\geq 2}^T \bar{a}_t s_t}{\sum_{t\geq 2}^T s_t} \geq 1$ (thanks to $\boldsymbol{p}^{mm}$ satisfying $\geq 1$ margin), this implies (23).

**Scenario 2:** $\|\bar{\boldsymbol{p}} - \boldsymbol{p}^{mm}\| \geq \epsilon := \pi/(2M)$. In this scenario, for some $\nu = \nu(\epsilon)$ and $\tau \geq 2$, we have that
$$\bar{\boldsymbol{p}}^\top (\boldsymbol{k}_1 - \boldsymbol{k}_\tau) = a_1 - a_\tau \leq 1 - 2\nu.$$

Here $\tau = \arg\max_{t\geq 2} \bar{\boldsymbol{p}}^\top \boldsymbol{k}_t$ denotes the nearest point to $\boldsymbol{k}_1$. Recall that $\boldsymbol{s} = \mathbb{S}(\bar{R}\boldsymbol{a})$ where $\bar{R} = \|\boldsymbol{p}\|/\|\boldsymbol{p}^{mm}\|$. To proceed, split the tokens into two groups: Let $\mathcal{N}$ be the group of tokens obeying $\bar{\boldsymbol{p}}^\top (\boldsymbol{k}_1 - \boldsymbol{k}_t) \leq 1 - \nu$ for $t \in \mathcal{N}$ and $[T] - \mathcal{N}$ be the rest. Observe that

$$\frac{\sum_{t\in[T]-\mathcal{N}} s_t}{\sum_{t\geq 2}^T s_t} \leq \frac{\sum_{t\in[T]-\mathcal{N}} s_t}{s_\tau} \leq T \frac{e^{\nu\bar{R}}}{e^{2\nu\bar{R}}} = Te^{-\bar{R}\nu}.$$

Set $\bar{M} = M/\Theta$ and note that $\|a_t\| \leq \|\boldsymbol{p}^{mm}\| \cdot \|\boldsymbol{k}_t\| \leq \bar{M}$. Using $\bar{\boldsymbol{p}}^\top (\boldsymbol{k}_1 - \boldsymbol{k}_t) \leq 1 - \nu$ over $t \in \mathcal{N}$ and plugging in the above bound, we obtain

$$\frac{\sum_{t\geq 2}^T (a_1 - a_t) s_t}{\sum_{t\geq 2}^T s_t} = \frac{\sum_{t\in\mathcal{N}} (a_1 - a_t) s_t}{\sum_{t\geq 2}^T s_t} + \frac{\sum_{t\in[T]-\mathcal{N}} (a_1 - a_t) s_t}{\sum_{t\geq 2}^T s_t}$$

$$\leq 1 - \nu + 2\bar{M}Te^{-\bar{R}\nu}.$$

Using the fact that $\bar{a}_1 - \frac{\sum_{t\geq 2}^T \bar{a}_t s_t}{\sum_{t\geq 2}^T s_t} \geq 1$, the above implies (23) with $\pi' = 2\bar{M}Te^{-\bar{R}\nu} - \nu$. To proceed, choose $R_\pi = \nu^{-1}\Theta^{-1}\log(2\bar{M}T/\pi)$ to ensure $\pi' \leq \pi$. ∎

The following lemma states the descent property of gradient descent for $\mathcal{L}(\boldsymbol{p})$ under Assumption A. It is important to note that although the infimum of the optimization problem is $\mathcal{L}^\star$, it is not achieved at any finite $\boldsymbol{p}$. Additionally, there are no finite critical points $\boldsymbol{p}$.

**Lemma 6** *Under Assumption A, the function $\mathcal{L}(\boldsymbol{p})$ is $L_p$-smooth, where*

$$L_p := \frac{1}{n} \sum_{i=1}^{n} \left( M_0 \|\boldsymbol{v}\|^2 \|\boldsymbol{W}\|^2 \|\boldsymbol{X}_i\|^4 + 3M_1 \|\boldsymbol{v}\| \, \|\boldsymbol{W}\|^2 \|\boldsymbol{X}_i\|^3 \right). \tag{24}$$

*Furthermore, if $\eta \leq 1/L_p$, then, for any initialization $\boldsymbol{p}(0)$, with the GD sequence $\boldsymbol{p}(t+1) = \boldsymbol{p}(t) - \eta \nabla \mathcal{L}(\boldsymbol{p}(t))$, we have*

$$\mathcal{L}(\boldsymbol{p}(t+1)) - \mathcal{L}(\boldsymbol{p}(t)) \leq -\frac{\eta}{2} \|\nabla \mathcal{L}(\boldsymbol{p}(t))\|^2, \tag{25}$$

*for all $t \geq 0$. This implies that*

$$\sum_{t=0}^{\infty} \|\nabla \mathcal{L}(\boldsymbol{p}(t))\|^2 < \infty, \quad \text{and} \quad \lim_{t \to \infty} \|\nabla \mathcal{L}(\boldsymbol{p}(t))\|^2 = 0. \tag{26}$$

**Proof.** Recall that we defined $\boldsymbol{\gamma}_i = Y_i \cdot X_i \boldsymbol{v}$ and $\boldsymbol{a}_i = \boldsymbol{K}_i \boldsymbol{p}$. The gradient of $\mathcal{L}(\boldsymbol{p})$ is given by

$$\nabla \mathcal{L}(\boldsymbol{p}) = \frac{1}{n} \sum_{i=1}^{n} \ell' \left( \boldsymbol{\gamma}_i^\top \mathbb{S}(\boldsymbol{K}_i \boldsymbol{p}) \right) \cdot \boldsymbol{K}_i^\top \mathbb{S}'(\boldsymbol{a}_i) \boldsymbol{\gamma}_i.$$

Note that for any $\boldsymbol{p} \in \mathbb{R}^d$, the Jacobian of $\mathbb{S}(\boldsymbol{K}_i \boldsymbol{p})$ is given by

$$\frac{\partial \mathbb{S}(\boldsymbol{K}_i \boldsymbol{p})}{\partial \boldsymbol{p}} = \mathbb{S}'(\boldsymbol{K}_i \boldsymbol{p}) \boldsymbol{K}_i = \left( \text{diag}(\mathbb{S}(\boldsymbol{K}_i \boldsymbol{p})) - \mathbb{S}(\boldsymbol{K}_i \boldsymbol{p}) \mathbb{S}(\boldsymbol{K}_i \boldsymbol{p})^\top \right) \boldsymbol{K}_i. \tag{27}$$

The Jacobian (27) together with the definition of the softmax function $\mathbb{S}(\cdot)$ implies that $\|\partial \mathbb{S}(\boldsymbol{K}_i \boldsymbol{p})/\partial \boldsymbol{p}\| \leq \|\boldsymbol{K}_i\|$. Hence, for any $\boldsymbol{p}, \dot{\boldsymbol{p}} \in \mathbb{R}^d$, we have

$$\|\mathbb{S}(\boldsymbol{K}_i \boldsymbol{p}) - \mathbb{S}(\boldsymbol{K}_i \dot{\boldsymbol{p}})\| \leq \|\boldsymbol{K}_i\| \, \|\boldsymbol{p} - \dot{\boldsymbol{p}}\|, \tag{28a}$$

and

$$\begin{aligned}
\left\| \mathbb{S}'(\boldsymbol{K}_i \boldsymbol{p}) - \mathbb{S}'(\boldsymbol{K}_i \dot{\boldsymbol{p}}) \right\| &\leq \left\| \text{diag}(\mathbb{S}(\boldsymbol{K}_i \boldsymbol{p})) - \text{diag}(\mathbb{S}(\boldsymbol{K}_i \dot{\boldsymbol{p}})) \right\| \\
&\quad + \left\| \mathbb{S}(\boldsymbol{K}_i \boldsymbol{p}) \mathbb{S}(\boldsymbol{K}_i \boldsymbol{p})^\top - \mathbb{S}(\boldsymbol{K}_i \dot{\boldsymbol{p}}) \mathbb{S}(\boldsymbol{K}_i \dot{\boldsymbol{p}})^\top \right\| \\
&\leq 3 \|\boldsymbol{K}_i\| \, \|\boldsymbol{p} - \dot{\boldsymbol{p}}\|.
\end{aligned} \tag{28b}$$

Here, the last inequality uses the fact that $|ab - cd| \leq |d||a - c| + |a||b - d|$.

Next, for any $\boldsymbol{p}, \dot{\boldsymbol{p}} \in \mathbb{R}^d$, we have

$$\begin{aligned}
\|\nabla \mathcal{L}(\boldsymbol{p}) - \nabla \mathcal{L}(\dot{\boldsymbol{p}})\| &\leq \frac{1}{n} \sum_{i=1}^{n} \left\| \ell' \left( \boldsymbol{\gamma}_i^\top \mathbb{S}(\boldsymbol{K}_i \boldsymbol{p}) \right) \cdot \boldsymbol{K}_i^\top \mathbb{S}'(\boldsymbol{K}_i \boldsymbol{p}) \boldsymbol{\gamma}_i - \ell' \left( \boldsymbol{\gamma}_i^\top \mathbb{S}(\boldsymbol{K}_i \dot{\boldsymbol{p}}) \right) \cdot \boldsymbol{K}_i^\top \mathbb{S}'(\boldsymbol{K}_i \dot{\boldsymbol{p}}) \boldsymbol{\gamma}_i \right\| \\
&\leq \frac{1}{n} \sum_{i=1}^{n} \left\| \boldsymbol{K}_i^\top \mathbb{S}'(\boldsymbol{K}_i \dot{\boldsymbol{p}}) \boldsymbol{\gamma}_i \right\| \, \left| \ell' \left( \boldsymbol{\gamma}_i^\top \mathbb{S}(\boldsymbol{K}_i \boldsymbol{p}) \right) - \ell' \left( \boldsymbol{\gamma}_i^\top \mathbb{S}(\boldsymbol{K}_i \dot{\boldsymbol{p}}) \right) \right| \\
&\quad + \frac{1}{n} \sum_{i=1}^{n} \left| \ell'(\boldsymbol{\gamma}_i^\top \mathbb{S}(\boldsymbol{K}_i \boldsymbol{p})) \right| \, \left\| \boldsymbol{K}_i^\top \mathbb{S}'(\boldsymbol{K}_i \boldsymbol{p}) \boldsymbol{\gamma}_i - \boldsymbol{K}_i^\top \mathbb{S}'(\boldsymbol{K}_i \dot{\boldsymbol{p}}) \boldsymbol{\gamma}_i \right\| \\
&\leq \frac{1}{n} \sum_{i=1}^{n} M_0 \|\boldsymbol{\gamma}_i\|^2 \, \|\boldsymbol{K}_i\| \, \|\mathbb{S}(\boldsymbol{K}_i \boldsymbol{p}) - \mathbb{S}(\boldsymbol{K}_i \dot{\boldsymbol{p}})\| \\
&\quad + \frac{1}{n} \sum_{i=1}^{n} M_1 \|\boldsymbol{\gamma}_i\| \, \|\boldsymbol{K}_i\| \, \left\| \mathbb{S}'(\boldsymbol{K}_i \boldsymbol{p}) - \mathbb{S}'(\boldsymbol{K}_i \dot{\boldsymbol{p}}) \right\|,
\end{aligned} \tag{29}$$

where the second inequality follows from the fact that $|ab - cd| \leq |d||a - c| + |a||b - d|$ and the third inequality uses Assumption A.

Substituting (28a) and (28b) into (29), we get

$$\begin{aligned}
\|\nabla \mathcal{L}(\boldsymbol{p}) - \nabla \mathcal{L}(\dot{\boldsymbol{p}})\| &\leq \frac{1}{n} \sum_{i=1}^{n} \left( M_0 \|\boldsymbol{\gamma}_i\|^2 \|\boldsymbol{K}_i\|^2 + 3M_1 \|\boldsymbol{K}_i\|^2 \|\boldsymbol{\gamma}_i\| \right) \|\boldsymbol{p} - \dot{\boldsymbol{p}}\| \\
&\leq \frac{1}{n} \sum_{i=1}^{n} \left( M_0 \|\boldsymbol{v}\|^2 \|\boldsymbol{W}\|^2 \|\boldsymbol{X}_i\|^4 + 3M_1 \|\boldsymbol{v}\| \|\boldsymbol{W}\|^2 \|\boldsymbol{X}_i\|^3 \right) \|\boldsymbol{p} - \dot{\boldsymbol{p}}\| \\
&\leq L_p \|\boldsymbol{p} - \dot{\boldsymbol{p}}\|,
\end{aligned}$$

where $L_p$ is defined in (24).

The remaining proof follows standard gradient descent analysis (see e.g. [22, Lemma 10]). Since $\mathcal{L}(\boldsymbol{p})$ is $L_p$-smooth, we get

$$
\begin{aligned}
\mathcal{L}(\boldsymbol{p}(t+1)) &\leq \mathcal{L}(\boldsymbol{p}(t)) + \nabla\mathcal{L}(\boldsymbol{p}(t))^\top (\boldsymbol{p}(t+1) - \boldsymbol{p}(t)) + \frac{L_p}{2}\|\boldsymbol{p}(t+1) - \boldsymbol{p}(t)\|^2 \\
&= \mathcal{L}(\boldsymbol{p}(t)) - \eta\|\nabla\mathcal{L}(\boldsymbol{p}(t))\|^2 + \frac{L_p\eta^2}{2}\|\nabla\mathcal{L}(\boldsymbol{p}(t))\|^2 \\
&= \mathcal{L}(\boldsymbol{p}(t)) - \eta\left(1 - \frac{L_p\eta}{2}\right)\|\nabla\mathcal{L}(\boldsymbol{p}(t))\|^2 \\
&\leq \mathcal{L}(\boldsymbol{p}(t)) - \frac{\eta}{2}\|\nabla\mathcal{L}(\boldsymbol{p}(t))\|^2,
\end{aligned}
$$

where the last inequality follows from our assumption on the stepsize.

The above inequality implies that

$$
\sum_{t=0}^{\infty}\|\nabla\mathcal{L}(\boldsymbol{p}(t))\|^2 \leq \frac{2}{\eta}\left(\mathcal{L}(\boldsymbol{p}(0)) - \mathcal{L}^*\right), \tag{30}
$$

where the right hand side is upper bounded by a finite constant. This is because, by Assumption A, $\mathcal{L}(\boldsymbol{p}(0)) < \infty$ and $\mathcal{L}^* \leq \mathcal{L}(\boldsymbol{p}(t))$, where $\mathcal{L}^*$ denotes the minimum objective.

Finally, (30) yields the expression (26). ∎

In the following lemma, we demonstrate the existence of parameters $\mu = \mu(\alpha) > 0$ and $R_\mu > 0$ such that when $R_\mu$ is sufficiently large, there are no stationary points within $C_{\mu, R_\mu}(\boldsymbol{p}^{mm})$. Additionally, we provide the local gradient correlation condition.

**Lemma 7 (Local Gradient Condition)** *Suppose Assumption A on the loss function $\ell$ holds. Let $\boldsymbol{\alpha} = (\alpha_i)_{i=1}^n$ be indices of locally-optimal tokens per Definition 2.*

**L1.** *There exists a positive scalar $\mu = \mu(\alpha) > 0$ such that for sufficiently large $\bar{R}_\mu$, no stationary point exists within $C_{\mu, \bar{R}_\mu}(\boldsymbol{p}^{mm})$, where $C_{\mu, \bar{R}_\mu}$ is defined in (8).*

**L2.** *For all $\boldsymbol{q}, \boldsymbol{p} \in \text{cone}_\mu(\boldsymbol{p}^{mm})$ with $\|\boldsymbol{q}\| = \|\boldsymbol{p}^{mm}\|$ and $\|\boldsymbol{p}\| \geq \bar{R}_\mu$ with same $\bar{R}_\mu$ choice as (**L1.**), there exist dataset dependent constants $C, c > 0$ such that*

$$
C \cdot \frac{1}{n}\sum_{i\in[n]}\{1 - \mathbb{S}(\boldsymbol{K}_i\boldsymbol{p})_{\alpha_i}\} \geq -\langle\nabla\mathcal{L}(\boldsymbol{p}), \boldsymbol{q}\rangle \geq c \cdot \frac{1}{n}\sum_{i\in[n]}\{1 - \mathbb{S}(\boldsymbol{K}_i\boldsymbol{p})_{\alpha_i}\} > 0, \tag{31a}
$$

$$
\|\nabla\mathcal{L}(\boldsymbol{p})\| \leq \bar{A}C \cdot \frac{1}{n}\sum_{i\in[n]}\{1 - \mathbb{S}(\boldsymbol{K}_i\boldsymbol{p})_{\alpha_i}\} \leq \bar{A}CTe^{-\bar{R}_\mu\Theta/2}. \tag{31b}
$$

$$
-\left\langle\frac{\boldsymbol{q}}{\|\boldsymbol{q}\|}, \frac{\nabla\mathcal{L}(\boldsymbol{p})}{\|\nabla\mathcal{L}(\boldsymbol{p})\|}\right\rangle \geq \frac{c\Theta}{C\bar{A}} > 0, \tag{31c}
$$

*Here, $\bar{A} = \max_{i\in[n], t, \tau\in[T]}\|\boldsymbol{k}_{it} - \boldsymbol{k}_{i\tau}\|$ and $\Theta = 1/\|\boldsymbol{p}^{mm}\|$.*

**L3.** *For any $\pi > 0$, there exists $R_\pi$ such that $R_\pi \geq \bar{R}_\mu$ and all $\boldsymbol{p} \in C_{\mu, R_\pi}(\boldsymbol{p}^{mm})$ obeys*

$$
\left\langle\nabla\mathcal{L}(\boldsymbol{p}), \frac{\boldsymbol{p}}{\|\boldsymbol{p}\|}\right\rangle \geq (1 + \pi)\left\langle\nabla\mathcal{L}(\boldsymbol{p}), \frac{\boldsymbol{p}^{mm}}{\|\boldsymbol{p}^{mm}\|}\right\rangle.
$$

**Proof.** Let $\boldsymbol{p}^{mm} = \boldsymbol{p}^{mm}(\alpha)$ be the solution of (ATT-SVM). Recall

$$
C_{\mu, \bar{R}_\mu}(\boldsymbol{p}^{mm}) = \text{cone}_\mu(\boldsymbol{p}^{mm})\bigcap\{\boldsymbol{p}\,|\,\|\boldsymbol{p}\| \geq \bar{R}_\mu\}.
$$

Let $(\mathcal{T}_i)_{i=1}^n$ be the sets of all SVM-neighbors per Definition 2. Let $\bar{\mathcal{T}}_i = [T] - \mathcal{T}_i - \{\alpha_i\}$ be the set of non-SVM-neighbor tokens, $i \in [n]$. Let

$$\Theta = 1/\|p^{mm}\|,$$

$$\delta = 0.5 \min_{i \in [n]} \min_{t \in \mathcal{T}_i, \tau \in \bar{\mathcal{T}}_i} (k_{it} - k_{i\tau})^\top p^{mm},$$

$$A = \max_{i \in [n], t \in [T]} \|k_{it}\|/\Theta, \tag{32}$$

$$\mu \le \mu(\delta) = \frac{1}{8} \left( \frac{\min(0.5, \delta)}{A} \right)^2.$$

When $\bar{\mathcal{T}}_i = \emptyset$ for all $i \in [n]$ (i.e. globally-optimal indices), we set $\delta = \infty$ as all non-neighbor related terms will disappear. Since $p^{mm}$ is the max-margin model ensuring $(k_{i\alpha_i} - k_{it})^\top p^{mm} \ge 1$ for all $i \in [n]$, the following inequalities hold for all $q \in \text{cone}_\mu(p^{mm})$, $\|q\| = \|p^{mm}\|$ and all $i \in [n], t \in \mathcal{T}_i, \tau \in \bar{\mathcal{T}}_i$:

$$(k_{it} - k_{i\tau})^\top q \ge \delta > 0,$$

$$(k_{i\alpha_i} - k_{i\tau})^\top q \ge 1 + \delta, \tag{33}$$

$$3/2 \ge (k_{i\alpha_i} - k_{it})^\top q \ge 1/2.$$

Here, we used $\|q - p^{mm}\|^2/\|p^{mm}\|^2 \le 2\mu$ which implies $\|q - p^{mm}\| \le \sqrt{2\mu}/\Theta$.

**L1.** and **L2.**. Now that the choice of local cone is determined, we need to prove the main claims. We will lower bound $-q^\top \nabla \mathcal{L}(p)$ and establish its strict positivity for $\|p\| \ge R$, where $R = \bar{R}_\mu$. This will show that there is no stationary point as a by product.

Consider any $q \in \mathbb{R}^d$ satisfying $\|q\| = \|p^{mm}\|$. To proceed, we write the gradient correlation following (18) and (21)

$$\langle \nabla \mathcal{L}(p), q \rangle = \frac{1}{n} \sum_{i=1}^n \ell_i' \cdot \langle a_i, \mathbb{S}'(a_i')\gamma_i \rangle, \tag{34}$$

where we denoted $\ell_i' = \ell'(Y_i \cdot v^\top X_i^\top \mathbb{S}(K_i p))$, $a_i = K_i q$, $a_i' = K_i p$, $s_i = \mathbb{S}(K_i p)$.

Using (33), for all $t \in \mathcal{T}_i, \tau \in \bar{\mathcal{T}}_i$, for all $p \in C_{\mu,R}(p^{mm})$, we have that

$$a_{i\alpha_i}' - a_{i\tau}' \ge R\Theta(1 + \delta), \quad \text{and} \quad a_{it}' - a_{i\tau}' \ge R\Theta\delta.$$

Consequently, we can bound the softmax probabilities $s_i = \mathbb{S}(K_i p)$ as follows: For all $i \in [n]$,

$$S_i := \sum_{\tau \in \bar{\mathcal{T}}_i} s_{i\tau} \le 1 - s_{i\alpha_i} = \sum_{\tau \ne \alpha_i} s_{i\tau} \le Te^{-R\Theta/2} s_{i\alpha_i} \le Te^{-R\Theta/2},$$

$$Q_i := \sum_{\tau \in \bar{\mathcal{T}}_i} s_{i\tau} \le Te^{-R\Theta\delta} s_{it_i} \le Te^{-R\Theta\delta} S_i, \quad \forall t_i \in \mathcal{T}_i. \tag{35}$$

Recall scores $\gamma_{it} = Y_i \cdot v^\top x_{it}$. Define the score gaps over neighbors:

$$\gamma_i^{gap} = \gamma_{i\alpha_i} - \max_{t \in \mathcal{T}_i} \gamma_{it}, \quad \text{and} \quad \bar{\gamma}_i^{gap} = \gamma_{i\alpha_i} - \min_{t \in \mathcal{T}_i} \gamma_{it}.$$

It follows from (32) that

$$A = \max_{i \in [n], t \in [T]} \|k_{it}\|/\Theta \ge \max_{i \in [n], t \in [T]} \|a_{it}\| = \|k_{it} q\|.$$

Define the $\alpha$-dependent global scalar $\Gamma = \sup_{i \in [n], t, \tau \in [T]} |\gamma_{it} - \gamma_{i\tau}|$. Let us focus on a fixed datapoint $i \in [n]$, assume (without losing generality) $\alpha_i = 1$, and drop subscripts $i$, that is, $\alpha := \alpha_i = 1$, $X := X_i$, $Y := Y_i$, $K := K_i$, $a' = Kp$, $a = Kq$, $s = \mathbb{S}(Kp)$, $\gamma = Y \cdot Xv$, $\gamma^{gap} := \gamma_i^{gap}$, $\bar{\gamma}^{gap} := \bar{\gamma}_i^{gap}$, $Q := Q_i$, and $S := S_i$. Directly applying Lemma 3, we obtain

$$\left| a^\top \text{diag}(s)\gamma - a^\top ss^\top \gamma - \sum_{t \ge 2}^T (a_1 - a_t)s_t(\gamma_1 - \gamma_t) \right| \le 2\Gamma A(1 - s_1)^2.$$

To proceed, let us decouple the non-neighbors within $\sum_{t \ge 2}^T (a_1 - a_t)s_t(\gamma_1 - \gamma_t)$ via

$$\left| \sum_{t \in \bar{\mathcal{T}}} (a_1 - a_t)s_t(\gamma_1 - \gamma_t) \right| \le 2Q\Gamma A.$$

Aggregating these, we found

$$\left| \boldsymbol{a}^\top \text{diag}(\boldsymbol{s})\boldsymbol{\gamma} - \boldsymbol{a}^\top \boldsymbol{s}\boldsymbol{s}^\top \boldsymbol{\gamma} - \sum_{t \in \mathcal{T}_i} (\boldsymbol{a}_1 - \boldsymbol{a}_t) s_t (\boldsymbol{\gamma}_1 - \boldsymbol{\gamma}_t) \right| \le 2\Gamma A ((1 - s_1)^2 + Q). \tag{36}$$

To proceed, let us upper/lower bound the gradient correlation. We use two bounds depending on $\boldsymbol{q} \in \text{cone}_\mu(\boldsymbol{p}^{mm})$ (Case 1) or general $\boldsymbol{q} \in \mathbb{R}^d$ (Case 2).

• Case 1: $\boldsymbol{q} \in \text{cone}_\mu(\boldsymbol{p}^{mm})$. Since $1.5 \ge \boldsymbol{a}_1 - \boldsymbol{a}_t \ge 0.5$ following (33), we find

$$1.5 \cdot S \cdot \bar{\gamma}^{gap} \ge \sum_{t \in \mathcal{T}_i} (\boldsymbol{a}_1 - \boldsymbol{a}_t) s_t (\boldsymbol{\gamma}_1 - \boldsymbol{\gamma}_t) \ge 0.5 \cdot S \cdot \gamma^{gap}.$$

Next we claim that $S$ dominates $((1 - s_1)^2 + Q)$ for large $R$. Specifically, we wish for

$$S \cdot \gamma^{gap}/4 \ge 4\Gamma A \max((1 - s_1)^2, Q) \iff S \ge 16\frac{\Gamma A}{\gamma^{gap}} \max((1 - s_1)^2, Q). \tag{37}$$

Now choose $R \ge \delta^{-1} \log(T)/\Theta$ to ensure $Q \le S$ since $Q \le T e^{-R\Theta\delta} S$. Consequently

$$(1 - s_1)^2 = (Q + S)^2 \le 4S^2 \le 4ST e^{-R\Theta/2}.$$

Combining these, what we wish is ensured by guaranteeing

$$S \ge 16\frac{\Gamma A}{\gamma^{gap}} \max(4ST e^{-R\Theta/2}, T e^{-R\Theta\delta} S). \tag{38}$$

This in turn is ensured for all inputs $i \in [n]$ by choosing

$$R = \frac{\max(2, \delta^{-1})}{\Theta} \log\left(\frac{64T\Gamma A}{\gamma_{\min}^{gap}}\right), \tag{39}$$

where $\gamma_{\min}^{gap} = \min_{i \in [n]} \gamma_i^{gap}$ is the global scalar which is the worst case score gap over all inputs. With the above choice of $R$ we guaranteed

$$2(1 - s_1) \cdot \bar{\gamma}^{gap} \ge 2 \cdot S \cdot \bar{\gamma}^{gap} \ge \boldsymbol{a}^\top \text{diag}(\boldsymbol{s})\boldsymbol{\gamma} - \boldsymbol{a}^\top \boldsymbol{s}\boldsymbol{s}^\top \boldsymbol{\gamma} \ge \frac{S \cdot \gamma^{gap}}{4} \ge \frac{(1 - s_1)\gamma^{gap}}{8}.$$

Since this holds over all inputs, going back to the gradient correlation (34) and averaging above over all inputs $i \in [n]$ and plugging back the indices $i$, we obtain the advertised bound by setting $q_i = 1 - s_{i\alpha_i}$ (where we set $\alpha_i = 1$ above without losing generality)

$$\frac{2}{n} \sum_{i \in [n]} -\ell_i' \cdot q_i \cdot \bar{\gamma}_i^{gap} \ge -\langle \nabla \mathcal{L}(\boldsymbol{p}), \boldsymbol{q} \rangle \ge \frac{1}{8n} \sum_{i \in [n]} -\ell_i' \cdot q_i \cdot \gamma_i^{gap}. \tag{40}$$

Let $-\ell_{\min/\max}'$ be the min/max values negative loss derivative admits over the ball $[-B, B]$ for $B = \|\boldsymbol{v}\| \cdot \max_{i,t} \|\boldsymbol{x}_{it}\|$ and note that $\max_{i \in [n]} \bar{\gamma}_i^{gap} > 0$ and $\min_{i \in [n]} \gamma_i^{gap} > 0$ are dataset dependent constants. Then, we declare the constants $C = -2\ell_{\max}' \cdot \max_{i \in [n]} \bar{\gamma}_i^{gap} > 0, c = -(1/8)\ell_{\min}' \cdot \min_{i \in [n]} \gamma_i^{gap} > 0$ to obtain the bound

$$\frac{C}{n} \sum_{i \in [n]} q_i \ge -\langle \nabla \mathcal{L}(\boldsymbol{p}), \boldsymbol{q} \rangle \ge \frac{c}{n} \sum_{i \in [n]} q_i, \tag{41}$$

which is the desired statement in (31a).

• Case 2: $\boldsymbol{q} \in \mathbb{R}^d$ and $\|\boldsymbol{q}\| = \|\boldsymbol{p}^{mm}\|$. Define $\bar{A} = \max_{i \in [n], t, \tau \in [T]} \|\boldsymbol{k}_{it} - \boldsymbol{k}_{i\tau}\|$. For any $\|\boldsymbol{q}\| = \|\boldsymbol{p}^{mm}\|$, we use the fact that

$$\|\boldsymbol{a}_1 - \boldsymbol{a}_t\| \le \|\boldsymbol{k}_1 - \boldsymbol{k}_t\| \cdot \|\boldsymbol{q}\| \le \frac{\bar{A}}{\Theta}.$$

Note that by definition $\frac{\bar{A}}{\Theta} \ge 1$. To proceed, we can upper bound

$$\frac{\bar{A}}{\Theta} \cdot S \cdot \bar{\gamma}^{gap} \ge \sum_{t \in \mathcal{T}} (\boldsymbol{a}_1 - \boldsymbol{a}_t) s_t (\boldsymbol{\gamma}_1 - \boldsymbol{\gamma}_t). \tag{42}$$

By choosing the same $R$ as in (39) to ensure $S$ dominates $((1 - s_1)^2 + Q)$ and since $\frac{\bar{A}}{\Theta} \geq 1$, we guaranteed

$$\frac{2\bar{A}}{\Theta} \cdot S \cdot \bar{\gamma}^{gap} \geq \boldsymbol{a}^\top \text{diag}(\boldsymbol{s})\boldsymbol{\gamma} - \boldsymbol{a}^\top \boldsymbol{s}\boldsymbol{s}^\top \boldsymbol{\gamma}.$$

Going back to the gradient correlation (34) and averaging above over all inputs $i \in [n]$, with the same definition of $C > 0$, we obtain

$$\frac{\bar{A}C}{\Theta n} \sum_{i \in [n]} q_i \geq -\langle \nabla \mathcal{L}(\boldsymbol{p}), \boldsymbol{q} \rangle. \tag{43}$$

To proceed, since (43) holds for any $\boldsymbol{q} \in \mathbb{R}^d$ and $\|\boldsymbol{q}\| = \|\boldsymbol{p}^{mm}\|$, we observe that when choosing $\boldsymbol{q} = \frac{\|\boldsymbol{p}^{mm}\|}{\|\nabla \mathcal{L}(\boldsymbol{p})\|} \cdot \nabla \mathcal{L}(\boldsymbol{p})$, this implies that

$$\langle \nabla \mathcal{L}(\boldsymbol{p}), \boldsymbol{q} \rangle = \|\nabla \mathcal{L}(\boldsymbol{p})\| \cdot \|\boldsymbol{p}^{mm}\| \leq \frac{\bar{A}C}{\Theta n} \sum_{i \in [n]} q_i.$$

Simplifying $\Theta = 1/\|\boldsymbol{p}^{mm}\|$ on both sides yields (31b). Incorporating (35) in the bound above provides the exponential upper bound that decay with $R$.

Combining this with (41), we obtain that for all $\boldsymbol{q}, \boldsymbol{p} \in \text{cone}_\mu(\boldsymbol{p}^{mm})$ and $\|\boldsymbol{q}\| \geq \bar{R}_\mu$

$$-\left\langle \frac{\boldsymbol{q}}{\|\boldsymbol{q}\|}, \frac{\nabla \mathcal{L}(\boldsymbol{p})}{\|\nabla \mathcal{L}(\boldsymbol{p})\|} \right\rangle \geq \frac{c\Theta}{C\bar{A}}.$$

This gives the desired result in (31c).

### L3.: Establishing gradient correlation.

Our final goal is establishing gradient comparison between $\boldsymbol{p}, \boldsymbol{p}^{mm}$ for the same choice of $\mu > 0$ provided in (32). Define $\bar{\boldsymbol{p}} = \|\boldsymbol{p}^{mm}\|\boldsymbol{p}/\|\boldsymbol{p}\|$ to be the normalized vector. Set notations $\boldsymbol{a}_i = \boldsymbol{K}_i\bar{\boldsymbol{p}}$, $\bar{\boldsymbol{a}}_i = \boldsymbol{K}_i\boldsymbol{p}^{mm}$, and $\boldsymbol{\gamma}_i = Y_i \cdot \boldsymbol{X}_i \boldsymbol{v}$.

To establish the result, using (34), we will prove that, for any $\pi > 0$, there is sufficiently large $R = R_\pi$ such that for any $\boldsymbol{p} \in C_{\mu,R}(\boldsymbol{p}^{mm})$:

$$\left\langle -\nabla \mathcal{L}(\boldsymbol{p}), \frac{\boldsymbol{p}}{\|\boldsymbol{p}\|} \right\rangle = -\frac{1}{n} \sum_{i=1}^n \ell_i' \cdot \langle \boldsymbol{a}_i, \mathbb{S}'(\boldsymbol{K}_i\boldsymbol{p})\boldsymbol{\gamma}_i \rangle$$

$$\leq -\frac{1 + \pi}{n} \sum_{i=1}^n \ell_i' \cdot \langle \bar{\boldsymbol{a}}_i, \mathbb{S}'(\boldsymbol{K}_i\boldsymbol{p})\boldsymbol{\gamma}_i \rangle = (1 + \pi) \left\langle -\nabla \mathcal{L}(\boldsymbol{p}), \frac{\boldsymbol{p}^{mm}}{\|\boldsymbol{p}^{mm}\|} \right\rangle. \tag{44}$$

Following (36), for all $i \in [n]$, for all $\boldsymbol{q} \in \text{cone}_\mu(\boldsymbol{p}^{mm})$ with $\|\boldsymbol{q}\| = \|\boldsymbol{p}^{mm}\|$, $\boldsymbol{a}' = \boldsymbol{K}\boldsymbol{q}$ and $\boldsymbol{s} = \mathbb{S}(\boldsymbol{K}\boldsymbol{p})$, we have found

$$\left| \boldsymbol{a}_i'^\top \text{diag}(\boldsymbol{s}_i)\boldsymbol{\gamma} - \boldsymbol{a}_i'^\top \boldsymbol{s}_i \boldsymbol{s}_i^\top \boldsymbol{\gamma}_i - \sum_{t \in \mathcal{T}_i}(a_{i1}' - a_{it}')s_{it}(\gamma_{i1} - \gamma_{it}) \right| \leq 2\Gamma A((1 - s_{i1})^2 + Q_i). \tag{45}$$

Plugging in $\boldsymbol{a}_i, \bar{\boldsymbol{a}}_i$ in the bound above and assuming $\pi \leq 1$ (w.l.o.g.), (44) is implied by the following stronger inequality

$$-\frac{1}{n} \sum_{i=1}^n \ell_i' \cdot \left( 6\Gamma A((1 - s_{i1})^2 + Q_i) + \sum_{t \in \mathcal{T}_i}(a_{i1} - a_{it})s_{it}(\gamma_{i1} - \gamma_{it}) \right)$$

$$\leq -\frac{1 + \pi}{n} \sum_{i=1}^n \ell_i' \cdot \sum_{t \in \mathcal{T}_i}(\bar{a}_{i1} - \bar{a}_{it})s_{it}(\gamma_{i1} - \gamma_{it})$$

$$= -\frac{1 + \pi}{n} \sum_{i=1}^n \ell_i' \cdot \sum_{t \in \mathcal{T}_i} s_{it}(\gamma_{i1} - \gamma_{it}).$$

First, we claim that $0.5\pi \sum_{t \in \mathcal{T}_i} s_{it}(\gamma_{i1} - \gamma_{it}) \geq 6\Gamma A((1 - s_{i1})^2 + Q_i)$ for all $i \in [n]$. The proof of this claim directly follows the earlier argument, namely, following (37), (38) and (39) which leads to the choice

$$R \geq \frac{\max(2, \delta^{-1})}{\Theta} \log\left( \frac{C_0 \cdot T\Gamma A}{\pi \gamma_{\min}^{gap}} \right), \tag{46}$$

for some constant $C_0 > 0$. Here, we choose sufficiently large $C_0 \geq 64\pi$ to ensure $R = R_\pi \geq \bar{R}_\mu$.

Following this control over the perturbation term $6\Gamma A((1 - s_{i1})^2 + Q_i)$, to conclude with the result, what remains is proving the comparison

$$-\frac{1}{n} \sum_{i=1}^{n} \ell_i' \cdot \sum_{t \in \mathcal{T}_i} (a_{i1} - a_{it}) s_{it} (\gamma_{i1} - \gamma_{it}) \leq -\frac{1 + 0.5\pi}{n} \sum_{i=1}^{n} \ell_i' \cdot \sum_{t \in \mathcal{T}_i} s_{it} (\gamma_{i1} - \gamma_{it}). \tag{47}$$

To proceed, we split the problem into two scenarios.

**Scenario 1:** $\|\bar{p} - p^{mm}\| \leq \epsilon = \frac{\pi}{4A\Theta}$ for some $\epsilon > 0$. In this scenario, for any token, we find that

$$|a_{it} - \bar{a}_t| = |k_{it}^\top (\bar{p} - p^{mm})| \leq A\Theta\epsilon = \pi/4.$$

Consequently, we obtain

$$a_{i1} - a_{it} \leq \bar{a}_{i1} - \bar{a}_{it} + 2A\Theta\epsilon = 1 + 0.5\pi.$$

Similarly, $a_{i1} - a_{it} \geq 1 - 0.5\pi \geq 0.5$. Since all terms $a_{i1} - a_{it}, s_{it}, \gamma_{i1} - \gamma_{it}$ in (47) are nonnegative and $(a_{i1} - a_{it}) s_{it} (\gamma_{i1} - \gamma_{it}) \leq (1 + 0.5\pi) s_{it} (\gamma_{i1} - \gamma_{it})$, above implies the desired result (47).

**Scenario 2:** $\|\bar{p} - p^{mm}\| \geq \epsilon = \frac{\pi}{4A\Theta}$. Since $\bar{p}$ is not (locally) max-margin, in this scenario, for some $i \in [n]$, $\nu = \nu(\epsilon) > 0$, and $\tau \in \mathcal{T}_i$, we have that

$$\bar{p}^\top (k_{i1} - k_{i\tau}) = a_{i1} - a_{i\tau} \leq 1 - 2\nu.$$

Here $\tau = \arg\max_{t \in \mathcal{T}_i} \bar{p}^\top k_{it}$ denotes the nearest point to $k_{i1}$ (along the $\bar{p}$ direction). Note that a non-neighbor $t \in \bar{\mathcal{T}}_i$ cannot be nearest because $\bar{p} \in \text{cone}_\mu(p^{mm})$ and (33) holds. Recall that $s_i = \mathbb{S}(\bar{R} a_i)$ where $\bar{R} = \|p\|\Theta \geq R\Theta$. To proceed, let $\underline{a}_i := \min_{t \in \mathcal{T}_i} a_{i1} - a_{it}$,

$$\mathcal{I} := \left\{ i \in [n] : \underline{a}_i \leq 1 - 2\nu \right\}, \qquad [n] - \mathcal{I} := \left\{ i \in [n] : 1 - 2\nu < \underline{a}_i \right\}.$$

For all $i \in [n] - \mathcal{I}$,

$$\begin{aligned}
\sum_{t \in \mathcal{T}_i} (a_{i1} - a_{it}) s_{it} (\gamma_{i1} - \gamma_{it}) &- (1 + 0.5\pi) \sum_{t \in \mathcal{T}_i} s_{it} (\gamma_{i1} - \gamma_{it}) \\
&\leq (2A - (1 + 0.5\pi)) \Gamma \sum_{t \in \mathcal{T}_i, \, a_{i1} - a_{it} \geq 1 + \frac{\pi}{2}} s_{it} \\
&\leq (2A - (1 + 0.5\pi)) \Gamma T e^{-\bar{R}(1 + \frac{\pi}{2})} \\
&\leq 2A\Gamma T e^{-\bar{R}(1 + \frac{\pi}{2})}.
\end{aligned} \tag{48}$$

For all $i \in \mathcal{I}$, split the tokens into two groups: Let $\mathcal{N}_i$ be the group of tokens obeying $a_{i1} - a_{it} \leq 1 - \nu$ and $\mathcal{T}_i - \mathcal{N}_i$ be the rest of the neighbors. Observe that

$$\frac{\sum_{t \in \mathcal{T}_i - \mathcal{N}_i} s_{it}}{\sum_{t \in \mathcal{T}_i} s_{it}} \leq T \frac{e^{\nu\bar{R}}}{e^{2\nu\bar{R}}} = T e^{-\bar{R}\nu}.$$

Thus, using $|a_{i1} - a_{it}| \leq 2A$ and recalling the definition of $\gamma_{\min}^{gap}$, observe that

$$\sum_{t \in \mathcal{T}_i - \mathcal{N}_i} (a_{i1} - a_{it}) s_{it} (\gamma_{i1} - \gamma_{it}) \leq \frac{2\Gamma A T e^{-\bar{R}\nu}}{\gamma_{\min}^{gap}} \sum_{t \in \mathcal{T}_i} s_{it} (\gamma_{i1} - \gamma_{it}).$$

Thus,

$$\begin{aligned}
\sum_{t \in \mathcal{T}_i} (a_{i1} - a_{it}) s_{it} (\gamma_{i1} - \gamma_{it}) &= \sum_{t \in \mathcal{N}_i} (a_{i1} - a_{it}) s_{it} (\gamma_{i1} - \gamma_{it}) + \sum_{t \in \mathcal{T}_i - \mathcal{N}_i} (a_{i1} - a_{it}) s_{it} (\gamma_{i1} - \gamma_{it}) \\
&\leq \sum_{t \in \mathcal{N}_i} (1 - \nu) s_{it} (\gamma_{i1} - \gamma_{it}) + \frac{2\Gamma A T e^{-\bar{R}\nu}}{\gamma_{\min}^{gap}} \sum_{t \in \mathcal{T}_i} s_{it} (\gamma_{i1} - \gamma_{it}) \\
&\leq \left( 1 - \nu + \frac{2\Gamma A T e^{-\bar{R}\nu}}{\gamma_{\min}^{gap}} \right) \sum_{t \in \mathcal{T}_i} s_{it} (\gamma_{i1} - \gamma_{it}) \\
&\leq \left( 1 + \frac{2\Gamma A T e^{-\bar{R}\nu}}{\gamma_{\min}^{gap}} \right) \sum_{t \in \mathcal{T}_i} s_{it} (\gamma_{i1} - \gamma_{it}).
\end{aligned}$$

Hence, choosing

$$R \geq \frac{1}{\nu\Theta} \log\left(\frac{8\Gamma AT}{\gamma_{\min}^{gap}\pi}\right) \tag{49}$$

results in that

$$
\begin{aligned}
&\sum_{t\in\mathcal{T}_i}(\boldsymbol{a}_{i1} - \boldsymbol{a}_{it})\boldsymbol{s}_{it}(\boldsymbol{\gamma}_{i1} - \boldsymbol{\gamma}_{it}) - (1 + \frac{\pi}{2})\sum_{t\in\mathcal{T}_i}\boldsymbol{s}_{it}(\boldsymbol{\gamma}_{i1} - \boldsymbol{\gamma}_{it}) \\
&\leq \left(\frac{2\Gamma AT e^{-\bar{R}\nu}}{\gamma_{\min}^{gap}} - \frac{\pi}{2}\right)\sum_{t\in\mathcal{T}_i}\boldsymbol{s}_{it}(\boldsymbol{\gamma}_{i1} - \boldsymbol{\gamma}_{it}) \\
&\leq -\frac{\pi}{4}\sum_{t\in\mathcal{T}_i}\boldsymbol{s}_{it}(\boldsymbol{\gamma}_{i1} - \boldsymbol{\gamma}_{it}) \\
&\leq -\frac{\pi}{4T}\gamma_{\min}^{gap}e^{-\bar{R}(1-2\nu)}.
\end{aligned} \tag{50}
$$

Here, the last inequality follows from the fact that $\sum_{t\in\mathcal{T}_i}\boldsymbol{s}_{it} \geq \max_{t\in\mathcal{T}_i}\boldsymbol{s}_{it} \geq \frac{e^{-\bar{R}(1-2\nu)}}{\sum_{t=1}^{T}e^{-\bar{R}(a_{i1}-a_{it})}} \geq e^{-\bar{R}(1-2\nu)}/T$.

From Assumption A, we have $c_{\min} \leq -\ell' \leq c_{\max}$ for some positive constants $c_{\min}$ and $c_{\max}$. It follows from (48) and (50) that

$$
\begin{aligned}
&-\frac{1}{n}\sum_{i}^{n}\ell_i'\cdot\left(\sum_{t\in\mathcal{T}_i}(\boldsymbol{a}_{i1} - \boldsymbol{a}_{it})\boldsymbol{s}_{it}(\boldsymbol{\gamma}_{i1} - \boldsymbol{\gamma}_{it}) - \sum_{t\in\mathcal{T}_i}(1 + 0.5\pi)\boldsymbol{s}_{it}(\boldsymbol{\gamma}_{i1} - \boldsymbol{\gamma}_{it})\right) \\
&\leq c_{\max}2A\Gamma TT e^{-\bar{R}(1+\frac{\pi}{2})} - \frac{c_{\min}}{nT}\cdot\frac{\pi\gamma_{\min}^{gap}}{4}e^{-\bar{R}(1-2\nu)} \\
&\leq 0.
\end{aligned}
$$

Combing with (49), this is guaranteed by choosing

$$R \geq \max\left\{\frac{1}{\nu\Theta}\log\left(\frac{8\Gamma AT}{\gamma_{\min}^{gap}\pi}\right), \frac{1}{(2\nu + \pi/2)\Theta}\log\left(\frac{8n\Gamma AT^2 c_{\max}}{c_{\min}\gamma_{\min}^{gap}\pi}\right)\right\},$$

where $\nu = \nu(\frac{\pi}{4A\Theta})$ depends only on $\pi$ and global problem variables.

Combining this with the prior $R$ choice (46) (by taking maximum), we conclude with the statement. ∎

## B.2 Proof of Theorem 1

**Proof.** This proof is a direct corollary of Lemma 14 which itself is a special case of the nonlinear head Theorem 8. Let us verify that $f(\boldsymbol{X}) = \boldsymbol{v}^\top\boldsymbol{X}^\top\mathbb{S}(\boldsymbol{X}\boldsymbol{p})$ satisfies the assumptions of Lemma 14 where we replace the nonlinear head with linear $\boldsymbol{v}$. To see this, set the optimal sets to be the singletons $O_i = \{\mathsf{opt}_i\}$. Given $(\boldsymbol{X}_i, Y_i)$, let $\boldsymbol{s}_i = \mathbb{S}(\boldsymbol{K}_i\boldsymbol{p})$ and $q_i = q_i^p = \sum_{t\neq\mathsf{opt}_i}\boldsymbol{s}_{it}$. Recalling score definition $\boldsymbol{\gamma}_i = Y_i \cdot \boldsymbol{X}_i\boldsymbol{v}$ and setting $v_i := \boldsymbol{\gamma}_{i\mathsf{opt}_i}$ and $Z_i := \sum_{t\neq\mathsf{opt}_i}\boldsymbol{\gamma}_{it}\boldsymbol{s}_{it}$, a particular prediction can be written as

$$
\begin{aligned}
Y_i \cdot \boldsymbol{v}^\top\boldsymbol{X}_i^\top\mathbb{S}(\boldsymbol{X}_i\boldsymbol{p}) = \boldsymbol{\gamma}_i^\top\boldsymbol{s}_i &= \boldsymbol{\gamma}_{i\mathsf{opt}_i}(1 - q_i) + \sum_{t\neq\mathsf{opt}_i}\boldsymbol{\gamma}_{it}\boldsymbol{s}_{it} \\
&= v_i(1 - q_i) + Z_i.
\end{aligned}
$$

To proceed, we demonstrate the choices for $C, \epsilon > 0$. Let $C := -\min_{i\in[n],t\in[T]}\boldsymbol{\gamma}_{it} \wedge 0$ and $q_{\max} = \max_{i\in[n]}q_i$. Note that $Z_i \geq \sum_{t\neq\mathsf{opt}_i}\boldsymbol{\gamma}_{it}\boldsymbol{s}_{it} \geq q_i\gamma_{\min} \geq -Cq_{\max}$. Now, using strict score optimality of $\mathsf{opt}_i$'s for all $i \in [n]$, we set

$$\epsilon := 1 - \sup_{i\in[n]}\frac{\sum_{t\neq\mathsf{opt}_i}\boldsymbol{\gamma}_{it}\boldsymbol{s}_{it}}{v_i q_i} \geq 1 - \sup_{i\in[n]}\frac{\sup_{t\neq\mathsf{opt}_i}\boldsymbol{\gamma}_{it}}{\boldsymbol{\gamma}_{i\mathsf{opt}_i}} > 0.$$

We conclude by observing $Z_i \leq v_i q_i \frac{\sum_{t\neq\mathsf{opt}_i}\boldsymbol{\gamma}_{it}\boldsymbol{s}_{it}}{v_i q_i} \leq v_i q_i\epsilon$ as desired. ∎

## B.3 Proof of Theorem 2

**Proof.** We first show that $\lim_{t\to\infty}\|\boldsymbol{p}(t)\| = \infty$. From Lemma 4, we have

$$\left\langle \nabla\mathcal{L}(\boldsymbol{p}), \boldsymbol{p}^{mm\star} \right\rangle = \frac{1}{n}\sum_{i=1}^{n} \ell'(Y_i \cdot \boldsymbol{v}^\top \boldsymbol{X}_i^\top \mathbb{S}(\boldsymbol{K}_i\boldsymbol{p})) \cdot \left\langle \boldsymbol{K}_i\boldsymbol{p}^{mm\star}, \mathbb{S}'(\boldsymbol{a}_i)\boldsymbol{\gamma}_i \right\rangle,$$

where $\boldsymbol{\gamma}_i = Y_i \cdot \boldsymbol{X}_i\boldsymbol{v}$ and $\boldsymbol{a}_i = \boldsymbol{K}_i\boldsymbol{p}$.

It follows from Lemma 4 that $\left\langle \nabla\mathcal{L}(\boldsymbol{p}), \boldsymbol{p}^{mm\star} \right\rangle < 0$ for all $\boldsymbol{p} \in \mathbb{R}^d$. Hence, for any finite $\boldsymbol{p}$, $\left\langle \nabla\mathcal{L}(\boldsymbol{p}), \boldsymbol{p}^{mm\star} \right\rangle$ cannot be equal to zero, as a sum of negative terms. Therefore, there are no finite critical points $\boldsymbol{p}$, for which $\nabla\mathcal{L}(\boldsymbol{p}) = 0$. On the other hand, Lemma 6 states $\nabla\mathcal{L}(\boldsymbol{p}(t)) \to 0$ which implies that $\|\boldsymbol{p}(t)\| \to \infty$.

Next, we provide the directional convergence for the setting $n = 1$. Let us consider an arbitrary value of $\epsilon \in (0, 1)$ and set $\pi = \epsilon/(1 - \epsilon)$. As $\lim_{t\to\infty}\|\boldsymbol{p}(t)\| = \infty$, we can select a specific $t_\epsilon$ such that for all $t \geq t_\epsilon$, it holds that $\|\boldsymbol{p}(t)\| \geq R_\epsilon \vee 1/2$ for any choice of $R_\epsilon$. To proceed, we choose $R_\epsilon$ based on Lemma 5 so that for any $t \geq t_\epsilon$, we have that

$$\left\langle -\nabla\mathcal{L}(\boldsymbol{p}(t)), \frac{\boldsymbol{p}^{mm\star}}{\|\boldsymbol{p}^{mm\star}\|} \right\rangle \geq (1 - \epsilon)\left\langle -\nabla\mathcal{L}(\boldsymbol{p}(t)), \frac{\boldsymbol{p}(t)}{\|\boldsymbol{p}(t)\|} \right\rangle.$$

Multiplying both sides by the stepsize $\eta$ and using the gradient descent update, we get

$$
\begin{aligned}
\left\langle \boldsymbol{p}(t+1) - \boldsymbol{p}(t), \frac{\boldsymbol{p}^{mm\star}}{\|\boldsymbol{p}^{mm\star}\|} \right\rangle &\geq (1 - \epsilon)\left\langle \boldsymbol{p}(t+1) - \boldsymbol{p}(t), \frac{\boldsymbol{p}(t)}{\|\boldsymbol{p}(t)\|} \right\rangle \\
&= \frac{(1 - \epsilon)}{2\|\boldsymbol{p}(t)\|}\left( \|\boldsymbol{p}(t+1)\|^2 - \|\boldsymbol{p}(t)\|^2 - \|\boldsymbol{p}(t+1) - \boldsymbol{p}(t)\|^2 \right) \\
&\geq (1 - \epsilon)\left( \frac{1}{2\|\boldsymbol{p}(t)\|}\left( \|\boldsymbol{p}(t+1)\|^2 - \|\boldsymbol{p}(t)\|^2 \right) - \|\boldsymbol{p}(t+1) - \boldsymbol{p}(t)\|^2 \right) \\
&\geq (1 - \epsilon)\left( \|\boldsymbol{p}(t+1)\| - \|\boldsymbol{p}(t)\| - \|\boldsymbol{p}(t+1) - \boldsymbol{p}(t)\|^2 \right) \\
&\geq (1 - \epsilon)\left( \|\boldsymbol{p}(t+1)\| - \|\boldsymbol{p}(t)\| - 2\eta\left(\mathcal{L}(\boldsymbol{p}(t)) - \mathcal{L}(\boldsymbol{p}(t+1))\right) \right).
\end{aligned}
\tag{51}
$$

Here, the second inequality is obtained from $\|\boldsymbol{p}(t)\| \geq 1/2$; the third inequality follows since for any $a, b > 0$, we have $(a^2 - b^2)/(2b) - (a - b) \geq 0$; and the last inequality uses Lemma 6.

Summing the above inequality over $t \geq t_\epsilon$ gives

$$\left\langle \frac{\boldsymbol{p}(t)}{\|\boldsymbol{p}(t)\|}, \frac{\boldsymbol{p}^{mm\star}}{\|\boldsymbol{p}^{mm\star}\|} \right\rangle \geq 1 - \epsilon + \frac{C(\epsilon, \eta)}{\|\boldsymbol{p}(t)\|},$$

for some finite constant $C(\epsilon, \eta)$ defined as

$$C(\epsilon, \eta) := \left\langle \boldsymbol{p}(t_\epsilon), \frac{\boldsymbol{p}^{mm\star}}{\|\boldsymbol{p}^{mm\star}\|} \right\rangle - (1 - \epsilon)\|\boldsymbol{p}(t_\epsilon)\| - 2\eta(1 - \epsilon)\left(\mathcal{L}(\boldsymbol{p}(t_\epsilon)) - \mathcal{L}^*\right), \tag{52}$$

where $\mathcal{L}^*$ denotes the minimum objective.

Since $\|\boldsymbol{p}(t)\| \to \infty$, we get

$$\liminf_{t\to\infty}\left\langle \frac{\boldsymbol{p}(t)}{\|\boldsymbol{p}(t)\|}, \frac{\boldsymbol{p}^{mm\star}}{\|\boldsymbol{p}^{mm\star}\|} \right\rangle \geq 1 - \epsilon.$$

Given that we can choose any value of $\epsilon \in (0, 1)$, we have $\boldsymbol{p}(t)/\|\boldsymbol{p}(t)\| \to \boldsymbol{p}^{mm\star}/\|\boldsymbol{p}^{mm\star}\|$. ∎

## B.4 Proof of Theorem 3

**Proof.** Following the proof of Lemma 7, let $(\mathcal{T}_i)_{i=1}^n$ denote the sets of SVM-neighbors as defined in Definition 2. We define $\bar{\mathcal{T}}_i = [T] - \mathcal{T}_i - \{\alpha_i\}$ as the tokens that are non-SVM neighbors. Additionally, let $\mu$ be defined as in (32). Let us denote the initialization lower bound as $R_\mu^0 := R$, where $R$ is given in the Theorem 3's statement. Consider an arbitrary value of $\epsilon \in (0, \mu/2)$ and let $1/(1 + \pi) = 1 - \epsilon$.

We additionally denote $R_\epsilon \leftarrow R_\pi \vee 1/2$ where $R_\pi$ was defined in Lemma 7(**L3.**). At initialization $\boldsymbol{p}(0)$, we set $\epsilon = \mu/2$ to obtain $R_\mu^0 = R_{\mu/2}$ and provide the proof in four steps:

**Step 1: There are no stationary points within** $C_{\mu,R_\mu^0}(\boldsymbol{p^{mm}})$**.** We begin by proving that there are no stationary points within $C_{\mu,R_\mu^0}(\boldsymbol{p^{mm}})$. Then, since $R_\mu^0 \geq \bar{R}_\mu$ per Lemma 7, we can apply (**L2.**) to find that: For all $\boldsymbol{q}, \boldsymbol{p} \in \text{cone}_\mu(\boldsymbol{p^{mm}})$ with $\boldsymbol{q} \neq 0$ and $\|\boldsymbol{p}\| \geq R_\mu^0$, we have that $-\boldsymbol{q}^\top \nabla \mathcal{L}(\boldsymbol{p})$ is strictly positive.

**Step 2:** It follows from Lemma 7(**L3.**) that, for all $\epsilon \in (0, \mu/2)$, all $\boldsymbol{p} \in C_{\mu,R_\epsilon}(\boldsymbol{p^{mm}})$ satisfy

$$\left\langle -\nabla \mathcal{L}(\boldsymbol{p}), \frac{\boldsymbol{p^{mm}}}{\|\boldsymbol{p^{mm}}\|} \right\rangle \geq (1 - \epsilon) \left\langle -\nabla \mathcal{L}(\boldsymbol{p}), \frac{\boldsymbol{p}}{\|\boldsymbol{p}\|} \right\rangle. \tag{53}$$

The argument above applies to a general $\epsilon \in (0, \mu/2)$. However, at initialization $\boldsymbol{p}(0)$, we set $\epsilon = \mu/2$ to obtain our earlier $R_\mu^0$ choice. To proceed, for any $\epsilon \in (0, \mu/2)$, we will show that after gradient descent enters the conic set $C_{\mu,R_\epsilon}(\boldsymbol{p^{mm}})$ for the first time, it will never leave the set. Let $t_\epsilon$ be the first time gradient descent enters $C_{\mu,R_\epsilon}(\boldsymbol{p^{mm}})$. In **Step 4**, we will prove that such $t_\epsilon$ is guaranteed to exist. Additionally, for $\epsilon \leftarrow \mu/2$, note that $t_\epsilon = 0$ i.e. the point of initialization.

**Step 3: Updates remain inside the cone** $C_{\mu,R_\epsilon}(\boldsymbol{p^{mm}})$**.** By leveraging the results from **Step 1** and **Step 2**, we demonstrate that the gradient iterates, with an appropriate constant step size, starting from $\boldsymbol{p}(t_\epsilon) \in C_{\mu,R_\epsilon}(\boldsymbol{p^{mm}})$, remain within this cone.

We proceed by induction. Suppose that the claim holds up to iteration $t \geq t_\epsilon$. This implies that $\boldsymbol{p}(t) \in C_{\mu,R_\epsilon}(\boldsymbol{p^{mm}})$. Hence, recalling cone definition, for $\mu > 0$ and $R_\epsilon$, we have $\left\langle \frac{\boldsymbol{p}(t)}{\|\boldsymbol{p}(t)\|}, \frac{\boldsymbol{p^{mm}}}{\|\boldsymbol{p^{mm}}\|} \right\rangle \geq 1 - \mu$ and $\|\boldsymbol{p}(t)\| \geq R_\epsilon$. Let

$$\rho(t) := -\frac{1}{1 - \epsilon} \left\langle \nabla \mathcal{L}(\boldsymbol{p}(t)), \frac{\boldsymbol{p^{mm}}}{\|\boldsymbol{p^{mm}}\|} \right\rangle.$$

Note that $\rho(t) > 0$ due to **Step 1**. This together with the gradient descent update rule gives

$$\begin{aligned}
\left\langle \frac{\boldsymbol{p}(t+1)}{\|\boldsymbol{p}(t)\|}, \frac{\boldsymbol{p^{mm}}}{\|\boldsymbol{p^{mm}}\|} \right\rangle &= \left\langle \frac{\boldsymbol{p}(t)}{\|\boldsymbol{p}(t)\|} - \frac{\eta}{\|\boldsymbol{p}(t)\|} \nabla \mathcal{L}(\boldsymbol{p}(t)), \frac{\boldsymbol{p^{mm}}}{\|\boldsymbol{p^{mm}}\|} \right\rangle \\
&\geq 1 - \mu - \frac{\eta}{\|\boldsymbol{p}(t)\|} \left\langle \nabla \mathcal{L}(\boldsymbol{p}(t)), \frac{\boldsymbol{p^{mm}}}{\|\boldsymbol{p^{mm}}\|} \right\rangle \\
&= 1 - \mu + \frac{\eta \rho(t)(1 - \epsilon)}{\|\boldsymbol{p}(t)\|}.
\end{aligned} \tag{54a}$$

Note that from Lemma 7, we have $\langle \nabla \mathcal{L}(\boldsymbol{p}(t)), \boldsymbol{p}(t) \rangle < 0$ which implies that $\|\boldsymbol{p}(t+1)\| \geq \|\boldsymbol{p}(t)\|$. This together with $R_\epsilon$ definition and $\|\boldsymbol{p}(t)\| \geq 1/2$ implies that

$$\begin{aligned}
\|\boldsymbol{p}(t+1)\| &\leq \frac{1}{2\|\boldsymbol{p}(t)\|} \left( \|\boldsymbol{p}(t+1)\|^2 + \|\boldsymbol{p}(t)\|^2 \right) \\
&= \frac{1}{2\|\boldsymbol{p}(t)\|} \left( 2\|\boldsymbol{p}(t)\|^2 - 2\eta \langle \nabla \mathcal{L}(\boldsymbol{p}(t)), \boldsymbol{p}(t) \rangle + \eta^2 \|\nabla \mathcal{L}(\boldsymbol{p}(t))\|^2 \right) \\
&\leq \|\boldsymbol{p}(t)\| - \frac{\eta}{\|\boldsymbol{p}(t)\|} \langle \nabla \mathcal{L}(\boldsymbol{p}(t)), \boldsymbol{p}(t) \rangle + \eta^2 \|\nabla \mathcal{L}(\boldsymbol{p}(t))\|^2.
\end{aligned} \tag{54b}$$

Hence, using (53)

$$\begin{aligned}
\frac{\|\boldsymbol{p}(t+1)\|}{\|\boldsymbol{p}(t)\|} &\leq 1 - \frac{\eta}{\|\boldsymbol{p}(t)\|} \left\langle \nabla \mathcal{L}(\boldsymbol{p}(t)), \frac{\boldsymbol{p}(t)}{\|\boldsymbol{p}(t)\|} \right\rangle + \eta^2 \frac{\|\nabla \mathcal{L}(\boldsymbol{p}(t))\|^2}{\|\boldsymbol{p}(t)\|} \\
&\leq 1 - \frac{\eta}{(1 - \epsilon)\|\boldsymbol{p}(t)\|} \left\langle \nabla \mathcal{L}(\boldsymbol{p}(t)), \frac{\boldsymbol{p^{mm}}}{\|\boldsymbol{p^{mm}}\|} \right\rangle + \eta^2 \frac{\|\nabla \mathcal{L}(\boldsymbol{p}(t))\|^2}{\|\boldsymbol{p}(t)\|} \\
&= 1 + \frac{\eta \rho(t)}{\|\boldsymbol{p}(t)\|} + \frac{\eta^2 \|\nabla \mathcal{L}(\boldsymbol{p}(t))\|^2}{\|\boldsymbol{p}(t)\|} =: C_1(\rho(t), \eta).
\end{aligned} \tag{54c}$$

Here, the second inequality follows from (53).

Now, it follows from (54a) and (54c) that

$$
\begin{aligned}
\left\langle \frac{\boldsymbol{p}(t+1)}{\|\boldsymbol{p}(t+1)\|}, \frac{\boldsymbol{p}^{mm}}{\|\boldsymbol{p}^{mm}\|} \right\rangle &\geq \frac{1}{C_1(\rho(t),\eta)}\left(1 - \mu + \frac{\eta\rho(t)(1-\epsilon)}{\|\boldsymbol{p}(t)\|}\right) \\
&= 1 - \mu + \frac{1}{C_1(\rho(t),\eta)}\left((1-\mu)(1 - C_1(\rho(t),\eta)) + \frac{\eta\rho(t)(1-\epsilon)}{\|\boldsymbol{p}(t)\|}\right) \\
&= 1 - \mu + \frac{\eta}{C_1(\rho(t),\eta)}\left((\mu-1)(\frac{\rho(t)}{\|\boldsymbol{p}(t)\|} + \frac{\eta\|\nabla\mathcal{L}(\boldsymbol{p}(t))\|^2}{\|\boldsymbol{p}(t)\|}) + \frac{\rho(t)(1-\epsilon)}{\|\boldsymbol{p}(t)\|}\right) \quad (55)\\
&= 1 - \mu + \frac{\eta}{C_1(\rho(t),\eta)}\left(\frac{\rho(t)(\mu-\epsilon)}{\|\boldsymbol{p}(t)\|} - \eta(1-\mu)\frac{\|\nabla\mathcal{L}(\boldsymbol{p}(t))\|^2}{\|\boldsymbol{p}(t)\|}\right) \\
&\geq 1 - \mu,
\end{aligned}
$$

where the last inequality uses our choice of stepsize $\eta \leq 1/L_p$ in Theorem 3's statement. Specifically, we need $\eta$ to be small to ensure the last inequality. We will guarantee this by choosing a proper $R_\epsilon$ in Lemma 7. Specifically, Lemma 7 leaves the choice of $C_0$ in $R_\epsilon$ lower bound of (46) open (it can always be chosen larger). Here, by choosing $C_0 \gtrsim 1/L_p$ will ensure $\eta \leq 1/L_p$ works well.

To proceed, we have that

$$
\begin{aligned}
\frac{(\mu-\epsilon)}{1-\mu}\frac{\rho(t)}{\|\nabla\mathcal{L}(\boldsymbol{p}(t))\|^2} &\geq \frac{\mu-\epsilon}{1-\mu}\frac{1}{1-\epsilon}\frac{c}{C}\frac{\Theta}{\bar{A}}\frac{1}{\bar{A}CT}e^{R_\mu^0\Theta/2} \\
&\geq \frac{\mu}{2(1-\mu)(1-\frac{\mu}{2})}\frac{c}{C}\frac{\Theta}{\bar{A}}\frac{1}{\bar{A}CT}e^{R_\mu^0\Theta/2} \geq \eta.
\end{aligned} \quad (56)
$$

Here, the second inequality uses our choice of $\epsilon \in (0, \mu/2)$ (see **Step 2**), and the first inequality is obtained from Lemma 7 since

$$
\begin{aligned}
\frac{\rho(t)}{\|\nabla\mathcal{L}(\boldsymbol{p}(t))\|} &= -\frac{1}{1-\epsilon}\left\langle \frac{\nabla\mathcal{L}(\boldsymbol{p}(t))}{\|\nabla\mathcal{L}(\boldsymbol{p}(t))\|}, \frac{\boldsymbol{p}^{mm}}{\|\boldsymbol{p}^{mm}\|} \right\rangle \geq \frac{1}{1-\epsilon}\frac{c}{C}\frac{\Theta}{\bar{A}}, \\
\frac{1}{\|\nabla\mathcal{L}(\boldsymbol{p}(t))\|} &\geq \frac{1}{\bar{A}C\frac{1}{n}\sum_{i=1}^n(1-s_{i\alpha_i})} \geq \frac{1}{\bar{A}CTe^{-R_\mu^0\Theta/2}}
\end{aligned}
$$

for some data dependent constants $c$ and $C$, $\bar{A} = \max_{i\in[n],t,\tau\in[T]}\|\boldsymbol{k}_{it} - \boldsymbol{k}_{i\tau}\|$, and $\Theta = 1/\|\boldsymbol{p}^{mm}\|$.

Next, we will demonstrate that the choice of $\eta$ in (56) does indeed meet our step size condition as stated in the theorem, i.e., $\eta \leq 1/L_p$. Recall that $1/(1+\pi) = 1 - \epsilon$, which implies that $\pi = \epsilon/(1-\epsilon)$. Combining this with (46), we obtain:

$$
R_\pi \geq \frac{\max(2,\delta^{-1})}{\Theta}\log\left(\frac{C_0T\Gamma A}{\pi\gamma_{\min}^{gap}}\right), \quad \text{where} \quad C_0 \geq 64\pi,
$$

$$
\Rightarrow R_\epsilon \geq \frac{\max(2,\delta^{-1})}{\Theta}\log\left(\frac{(1-\epsilon)C_0T\Gamma A}{\epsilon\gamma_{\min}^{gap}}\right), \quad \text{where} \quad C_0 \geq 64\frac{\epsilon}{1-\epsilon}.
$$

On the other hand, at the initialization, we have $\epsilon = \mu/2$ which implies that

$$
R_\mu^0 \geq \frac{\max(2,\delta^{-1})}{\Theta}\log\left(\frac{(2-\mu)C_0T\Gamma A}{\mu\gamma_{\min}^{gap}}\right), \quad \text{where} \quad C_0 \geq 64\frac{\mu}{(2-\mu)}. \quad (57)
$$

In the following, we will determine a lower bound on $C_0$ such that our step size condition in Theorem 3's statement, i.e., $\eta \leq 1/L_p$, is satisfied. Note that for the choice of $\eta$ in (56) to meet the condition $\eta \leq 1/L_p$, the following condition must hold:

$$
\frac{1}{L_p} \leq \frac{\mu}{(2-\mu)}\frac{1}{C_2T}e^{R_\mu^0\Theta/2} \Rightarrow R_\mu^0 \geq \frac{2}{\Theta}\log\left(\frac{1}{L_p}\frac{(2-\mu)}{\mu}C_2T\right), \quad (58)
$$

where $C_2 = (1-\mu)\frac{\bar{A}^2C^2}{\Theta c}$.

This together with (57) implies that for sufficiently large

$$
R_\mu^0 \geq \frac{\max(2,\delta^{-1})}{\Theta}\log\left(\frac{(2-\mu)C_3T}{\mu}\right), \quad \text{where} \quad C_3 = \frac{C_0\Gamma A}{\gamma_{\min}^{gap}} \vee \frac{C_2}{L_p},
$$

the step size bound in (56) ensures that $\eta \le 1/L_p$ guarantees (55). Hence, $\boldsymbol{p}(t + 1)$ remains within the cone, i.e., $\boldsymbol{p}(t + 1) \in C_{\mu,R_\epsilon}(\boldsymbol{p}^{mm})$.

**Step 4: The correlation of $\boldsymbol{p}(t)$ and $\boldsymbol{p}^{mm}$ increases over $t$.** The remainder is similar to the proof of Theorem 2. From Step 3, we have that all iterates remain within the initial conic set i.e. $\boldsymbol{p}(t) \in C_{\mu,R_\mu^0}(\boldsymbol{p}^{mm})$ for all $t \ge 0$. Note that it follows from Lemma 7 that $\langle \nabla \mathcal{L}(\boldsymbol{p}), \boldsymbol{p}^{mm}/\|\boldsymbol{p}^{mm}\| \rangle < 0$, for any finite $\boldsymbol{p} \in C_{\mu,R_\mu^0}(\boldsymbol{p}^{mm})$. Hence, there are no finite critical points $\boldsymbol{p} \in C_{\mu,R_\mu^0}(\boldsymbol{p}^{mm})$, for which $\nabla \mathcal{L}(\boldsymbol{p}) = 0$. Now, based on Lemma 6, which guarantees that $\nabla \mathcal{L}(\boldsymbol{p}(t)) \to 0$, this implies that $\|\boldsymbol{p}(t)\| \to \infty$. Consequently, for any choice of $\epsilon \in (0, \mu/2)$ there is a time $t_\epsilon$ such that, for all $t \ge t_\epsilon$, $\boldsymbol{p}(t) \in C_{\mu,R_\epsilon}(\boldsymbol{p}^{mm})$. Once within $C_{\mu,R_\epsilon}(\boldsymbol{p}^{mm})$, following similar steps in (51) and (52), for any $t \ge t_\epsilon$,

$$\left\langle \frac{\boldsymbol{p}(t)}{\|\boldsymbol{p}(t)\|}, \frac{\boldsymbol{p}^{mm}}{\|\boldsymbol{p}^{mm}\|} \right\rangle \ge 1 - \epsilon + \frac{C_2(\epsilon, \eta)}{\|\boldsymbol{p}(t)\|}, \quad \boldsymbol{p}(t) \in C_{\mu,R_\epsilon}(\boldsymbol{p}^{mm}),$$

for some finite constant $C_2(\epsilon, \eta)$. Consequently,

$$\liminf_{t \to \infty} \left\langle \frac{\boldsymbol{p}(t)}{\|\boldsymbol{p}(t)\|}, \frac{\boldsymbol{p}^{mm}}{\|\boldsymbol{p}^{mm}\|} \right\rangle \ge 1 - \epsilon, \quad \text{where} \quad \boldsymbol{p}(t) \in C_{\mu,R_\epsilon}(\boldsymbol{p}^{mm}).$$

Since the choice of $\epsilon \in (0, \mu/2)$ is arbitrary, we obtain $\boldsymbol{p}(t)/\|\boldsymbol{p}(t)\| \to \boldsymbol{p}^{mm}/\|\boldsymbol{p}^{mm}\|$. ∎

## B.5 Proof of Theorem 4

### B.5.1 Supporting Lemma

We present a lemma that will aid in simplifying our analysis. We begin with a definition.

**Definition 3 (Selected-tokens, Neighbors, Margins, and Neighbor-optimality of a direction)**
*Let $\boldsymbol{q} \in \mathbb{R}^d - \{\boldsymbol{0}\}$ and $(Y_i, \boldsymbol{K}_i, \boldsymbol{X}_i)_{i=1}^n$ be our dataset. We define the (possibly non-unique) selected-tokens of $\boldsymbol{q}$ as follows:*[3]

$$\alpha_i \in \arg\max_{t \in [T]} \boldsymbol{k}_{it}^\top \boldsymbol{q}. \tag{59}$$

*Next, we define the margin and directional-neighbors for $\boldsymbol{q}$ as the minimum margin tokens to the selected-tokens, i.e.,*

$$\Gamma_{\boldsymbol{q}} = \min_{i \in [n], t \ne \alpha_i} (\boldsymbol{k}_{i\alpha_i} - \boldsymbol{k}_{it})^\top \boldsymbol{q}, \tag{60}$$

$$\mathcal{M}_{\boldsymbol{q}} = \left\{ (i, t) \mid (\boldsymbol{k}_{i\alpha_i} - \boldsymbol{k}_{it})^\top \boldsymbol{q} = \Gamma_{\boldsymbol{q}} \right\}. \tag{61}$$

*Finally, we say that $\boldsymbol{q}$ is neighbor-optimal if the scores of its directional-neighbors are strictly less than the corresponding selected-token. Concretely, for all $(i, t) \in \mathcal{M}_{\boldsymbol{q}}$, we require that*

$$\gamma_{it} = Y_i \cdot \boldsymbol{x}_{it}^\top \boldsymbol{v} < \gamma_{i\alpha_i} = Y_i \cdot \boldsymbol{x}_{i\alpha_i}^\top \boldsymbol{v}.$$

**Lemma 8 (When does one direction dominate another?)** *Suppose $\boldsymbol{q}, \boldsymbol{p} \in \mathbb{R}^d$ be two unit Euclidean norm vectors with identical selected tokens. Specifically, for each $i \in [n]$, there exists unique $\alpha_i \in [T]$ such that $\alpha_i = \arg\max_{t \in [T]} \boldsymbol{k}_{it}^\top \boldsymbol{q} = \arg\max_{t \in [T]} \boldsymbol{k}_{it}^\top \boldsymbol{p}$. Suppose directional margins obey $\Gamma_{\boldsymbol{q}} < \Gamma_{\boldsymbol{p}}$ and set $\delta_\Gamma = \Gamma_{\boldsymbol{p}} - \Gamma_{\boldsymbol{q}}$.*

- *Suppose $\boldsymbol{q}$ and $\boldsymbol{p}$ are both neighbor-optimal. Then, for some $R(\delta_\Gamma)$ and all $R > R(\delta_\Gamma)$, we have that $\mathcal{L}(R \cdot \boldsymbol{p}) < \mathcal{L}(R \cdot \boldsymbol{q})$.*

- *Suppose $\boldsymbol{q}$ has a unique directional-neighbor and is not neighbor-optimal (i.e. this neighbor has higher score). Let $\delta_{\boldsymbol{q}}$ be the margin difference between unique directional-neighbor and the second-most minimum-margin neighbor (i.e. the one after the unique one, see (65)) of $\boldsymbol{q}$. Then, for some $R(\delta_\Gamma \wedge \delta_{\boldsymbol{q}})$ and all $R > R(\delta_\Gamma \wedge \delta_{\boldsymbol{q}})$, we have that $\mathcal{L}(R \cdot \boldsymbol{q}) < \mathcal{L}(R \cdot \boldsymbol{p})$.*

**Proof.** We prove these two statements in order. First define the directional risk baseline induced by letting $R \to \infty$ and purely selecting the tokens $\boldsymbol{\alpha} = (\alpha_i)_{i=1}^n$. This is given by

$$\mathcal{L}_\star := \frac{1}{n} \sum_{i=1}^n \ell \left( Y_i \cdot \boldsymbol{v}^\top \boldsymbol{x}_{i\alpha_i} \right).$$

---

[3]If $\alpha_i$ is unique for all $i \in [n]$, let us call it, unique selected tokens.

We evaluate $q, p$ with respect to $\mathcal{L}_\star$. To proceed, let $s_i = \mathbb{S}(RK_iq)$. Define $\Gamma_q^{it} = k_{i\alpha_i}^\top q - k_{it}^\top q$. Note that, the smallest value for $t \neq \alpha_i$ is achieved for $\Gamma_q$. For sufficiently large $R \gtrsim O(\log(T)/\Gamma_q)$, observe that, for $t \neq \alpha_i$

$$e^{-R\Gamma_q^{it}} \geq s_{it} = \frac{e^{Rk_{it}^\top q}}{\sum_{t\in[T]} e^{Rk_{it}^\top q}} \geq 0.5 e^{-R\Gamma_q^{it}}. \tag{62}$$

Recalling the score definition and let $M_+, M_-$ be the upper and lower bounds on $-\ell'$ over its bounded domain that scores fall on, respectively. Note that, for some intermediate $M_+ \geq M_i \geq M_-$ values, we have

$$\mathcal{L}(Rq) - \mathcal{L}_\star = \frac{1}{n} \sum_{i=1}^n \ell\Big(\sum_{t\in[T]} s_{it}\gamma_{it}\Big) - \ell(\gamma_{i\alpha_i})$$

$$= \frac{1}{n} \sum_{i=1}^n M_i \sum_{t\neq\alpha_i} s_{it}(\gamma_{i\alpha_i} - \gamma_{it}).$$

Now, using (62) for a refreshed $M_+ \geq M_{it} \geq 0.5M_-$ values, we can write

$$\mathcal{L}(Rq) - \mathcal{L}_\star = \frac{1}{n} \sum_{i=1}^n \sum_{t\neq\alpha_i} M_{it} e^{-R\Gamma_q^{it}}(\gamma_{i\alpha_i} - \gamma_{it}). \tag{63}$$

The same bound also applies to $p$ with some $M'_{it}$, multipliers

$$\mathcal{L}(Rp) - \mathcal{L}_\star = \frac{1}{n} \sum_{i=1}^n \sum_{t\neq\alpha_i} M'_{it} e^{-R\Gamma_p^{it}}(\gamma_{i\alpha_i} - \gamma_{it}). \tag{64}$$

We can now proceed with the proof.

**Case 1: $q$ and $p$ are both neighbor-optimal.** This means that $\gamma_{i\alpha_i} - \gamma_{it} > 0$ for all $i \in [n], t \neq \alpha_i$. Let $K_+ > K_- > 0$ be upper and lower bounds on $\gamma_{i\alpha_i} - \gamma_{it}$ values. We can now upper bound the right hand side of (63) via

$$M_+ K_+ T e^{-R\Gamma_q} \geq \mathcal{L}(Rq) - \mathcal{L}_\star \geq \frac{1}{2n} M_- K_- e^{-R\Gamma_q}.$$

Consequently, $\mathcal{L}(Rq) > \mathcal{L}(Rp)$ as soon as $\frac{1}{2n} M_- K_- e^{-R\Gamma_q} > M_+ K_+ T e^{-R\Gamma_p}$. Since $M_+, K_+, n, T$ are global constants, this happens under the stated condition on the margin gap $\Gamma_p - \Gamma_q$.

**Case 2: $q$ has a unique directional-neighbor and is not neighbor-optimal.** In this scenario, $\mathcal{L}(Rq) - \mathcal{L}_\star$ is actually negative for large $R$. To proceed, define the maximum score difference $K_+ = \sup_{i,t\neq\alpha_i} |\gamma_{i\alpha_i} - \gamma_{it}|$. Also let $(j,\beta)$ be the unique directional neighbor achieving the minimum margin $\Gamma_q$. Then, $\delta_q$ – the margin difference between unique directional-neighbor and the second minimum-margin neighbor (i.e. the one after the unique one) of $q$ – is defined as

$$\delta_q = \min_{i\in[n],\ t\neq\alpha_i,\ (i,t)\neq(j,\beta)} \Gamma_q^{it} - \Gamma_q. \tag{65}$$

To proceed, we can write

$$\mathcal{L}(Rp) - \mathcal{L}_\star \geq -M_+ K_+ T e^{-R\Gamma_p}.$$

On the other hand, setting $\kappa = \gamma_{j\beta} - \gamma_{j\alpha_j} > 0$, we can bound

$$\mathcal{L}(Rq) - \mathcal{L}_\star = -\frac{1}{n} M_{j\beta} e^{-R\Gamma_q} \kappa + \frac{1}{n} \sum_{i\in[n],\ t\neq\alpha_i,\ (i,t)\neq(j,\beta)} M_{it} e^{-R\Gamma_q^{it}}(\gamma_{i\alpha_i} - \gamma_{it})$$

$$\leq -\frac{1}{n} M_- e^{-R\Gamma_q} \kappa + M_+ K_+ T e^{-R(\Gamma_q + \delta_q)}.$$

Consequently, we have found that $\mathcal{L}(Rp) > \mathcal{L}(Rq)$ as soon as

$$\frac{1}{n} M_- e^{-R\Gamma_q} \kappa \geq M_+ K_+ T (e^{-R(\Gamma_q + \delta_q)} + e^{-R\Gamma_p})$$

This happens when $R \gtrsim \frac{1}{\delta_q \wedge (\Gamma_p - \Gamma_q)}$ (up to logarithmic terms) establishing the desired statement. ∎

### B.5.2 Proof of Theorem 4

Define the locally-optimal unit directions

$$\mathcal{P}^{mm} = \left\{ \frac{\boldsymbol{p}^{mm}(\alpha)}{\|\boldsymbol{p}^{mm}(\alpha)\|} \,\Big|\, \alpha \text{ is a locally-optimal set of indices} \right\}.$$

The theorem below shows that cone-restricted regularization paths can only directionally converge to an element of this set.

**Theorem 7 (Non-LOMM Regularization Paths Fail)** *Fix a unit Euclidean norm vector $\boldsymbol{q} \in \mathbb{R}^d$ such that $\boldsymbol{q} \notin \mathcal{P}^{mm}$. Assume that the token scores are distinct (i.e., $\gamma_{it} \neq \gamma_{i\tau}$ for $t \neq \tau$) and the key embeddings $\boldsymbol{k}_{it}$ are in general position. Specifically, we require the following conditions to hold [4]:*

- *When $m = d$, all matrices $\bar{\boldsymbol{K}} \in \mathbb{R}^{m \times d}$ where each row of $\bar{\boldsymbol{K}}$ has the form $\boldsymbol{k}_{it} - \boldsymbol{k}_{i\alpha_i}$ for a unique $(i, \alpha_i, t \neq \alpha_i)$ tuple, are full-rank.*

- *When $m = d + 1$, the vector of all ones is not in the range space of any such $\bar{\boldsymbol{K}}$ matrix.*

*Fix arbitrary $\epsilon > 0, R_0 > 0$. Define the local regularization path of $\boldsymbol{q}$ as its $(\epsilon, R_0)$-conic neighborhood:*

$$\bar{\boldsymbol{p}}(R) = \underset{\boldsymbol{p} \in C_{\epsilon, R_0}(\boldsymbol{q}), \|\boldsymbol{p}\| \leq R}{\arg\min} \mathcal{L}(\boldsymbol{p}), \quad \text{where} \quad C_{\epsilon, R_0}(\boldsymbol{q}) = \mathsf{cone}_\epsilon(\boldsymbol{q}) \cap \left\{ \boldsymbol{p} \in \mathbb{R}^d \,\big|\, \|\boldsymbol{p}\| \geq R_0 \right\}. \tag{66}$$

*Then, either $\lim_{R \to \infty} \|\bar{\boldsymbol{p}}(R)\| < \infty$ or $\lim_{R \to \infty} \bar{\boldsymbol{p}}(R)/\|\bar{\boldsymbol{p}}(R)\| \neq \boldsymbol{q}$. In both scenarios $\lim_{R \to \infty} \bar{\boldsymbol{p}}(R)/R \neq \boldsymbol{q}$.*

**Proof.** We will prove the result by dividing the problem into distinct cases. In each case, we will construct an alternative direction that achieves a strictly better objective than some $\delta = \delta(\epsilon) > 0$ neighborhood of $\boldsymbol{q}$, thereby demonstrating the suboptimality of the $\boldsymbol{q}$ direction. Let's define the $\delta$ neighborhood as follows:

$$\mathcal{N}_\delta = \left\{ \boldsymbol{p} \,\Big|\, \left\| \frac{\boldsymbol{p}}{\|\boldsymbol{p}\|} - \boldsymbol{q} \right\| \leq \delta \quad \text{and} \quad \|\boldsymbol{p}\| \geq R_0 \right\}. \tag{67}$$

Now, let's recall a few more definitions based on Definition 3. First, the tokens selected by $\boldsymbol{q}$ are given by (59). To proceed, let's initially consider the scenario where $\alpha_i$ is unique for all $i \in [n]$, meaning that as we let $c \to \infty$, $c \cdot \boldsymbol{q}$ will choose a single token per input. Later, we will revisit the setting when arg max is not a singleton, and $\boldsymbol{q}$ is allowed to select multiple tokens.

Additionally, it's important to note that $\|\bar{\boldsymbol{p}}(R)\|$ is non-decreasing by definition. Suppose it has a finite upper bound $\|\bar{\boldsymbol{p}}(R)\| \leq M$ for all $R < \infty$. In that scenario, we have $\lim_{R \to \infty} \frac{\bar{\boldsymbol{p}}(R)}{R} = 0 \neq \boldsymbol{q}$.

- **(A) $\boldsymbol{q}$ selects a single token per input:** Given that the indices $\alpha = (\alpha_i)_{i=1}^n$ defined in (59) are uniquely determined, we can conclude that the $\boldsymbol{q}$ direction eventually selects tokens $\boldsymbol{k}_{i\alpha_i}$. Recall the definition of the margin $\Gamma_{\boldsymbol{q}}$ from (60) and the set of directional neighbors, which is defined as the indices that achieve $\Gamma_{\boldsymbol{q}}$, as shown in (61). Let us refer to $\boldsymbol{q}$ as *neighbor-optimal* if $\gamma_{it} < \gamma_{i\alpha_i}$ for all $(i, t) \in \mathcal{M}_{\boldsymbol{q}}$.

We will consider two cases for this scenario: when $\boldsymbol{q}$ is neighbor-optimal and when $\boldsymbol{q}$ is not neighbor-optimal.

◇ **(A1) $\boldsymbol{q}$ is neighbor-optimal.** In this case, we will argue that max-margin direction $\bar{\boldsymbol{p}}^{mm} := \boldsymbol{p}^{mm}(\alpha)/\|\boldsymbol{p}^{mm}(\alpha)\|$ can be used to construct a strictly better objective than $\boldsymbol{q}$. Note that $\boldsymbol{p}^{mm}(\alpha)$ exists because $\boldsymbol{q}$ is already a viable separating direction for tokens $\alpha$. Specifically, consider the direction $\boldsymbol{q}' = \frac{\boldsymbol{q} + \epsilon \bar{\boldsymbol{p}}^{mm}}{\|\boldsymbol{q} + \epsilon \bar{\boldsymbol{p}}^{mm}\|}$. Observe that, $\boldsymbol{q}'$ lies within $\mathsf{cone}_{2\epsilon}(\boldsymbol{q})$,[5]

$$\boldsymbol{q}^\top \boldsymbol{q}' \geq \frac{1 - \epsilon}{1 + \epsilon} \geq 1 - 2\epsilon.$$

We now argue that, there exists $\delta = \delta_\epsilon > 0$ such that for all $R > R_\epsilon$

$$\min_{R_\epsilon \leq r \leq R} \mathcal{L}(r \cdot \boldsymbol{q}') < \min_{\boldsymbol{p} \in \mathcal{N}_\delta, R_\epsilon \leq \|\boldsymbol{p}\| \leq R} \mathcal{L}(\boldsymbol{p}). \tag{68}$$

---

[4]This requirement holds for general data because it is guaranteed by adding arbitrarily small independent gaussian perturbations to keys $\boldsymbol{k}_{it}$.

[5]As a result, let us prove the result for $\epsilon \leftarrow 2\epsilon$ without losing generality.

To prove this, we study the margin $\Gamma_{q'}$ induced by $q'$ and the maximum margin $\Gamma_\delta$ induced within $p \in \mathcal{N}_\delta$. Concretely, we will show that $\Gamma_{q'} > \Gamma_\delta$ and directly apply the first statement of Lemma 8 to conclude with (68).

Let $\Gamma = 1/\|p^{mm}(\alpha)\|$ be the margin induced by $\bar{p}^{mm}$. Note that $\Gamma > \Gamma_q$ by the optimality of $p^{mm}(\alpha)$ and the fact that $q \neq p^{mm}(\alpha)$. Consequently, we can lower and upper bound the margins via

$$\Gamma_{q'} = \min_{i\in[n]} \min_{t\neq\alpha_i}(k_{i\alpha_i} - k_{it})^\top q' \geq \frac{\Gamma_q + \epsilon\Gamma}{1 + \epsilon} \geq \Gamma_q + \frac{\epsilon}{2}(\Gamma - \Gamma_q),$$

$$\Gamma_\delta = \max_{p\in\mathcal{N}_\delta} \min_{i\in[n]} \min_{t\neq\alpha_i}(k_{i\alpha_i} - k_{it})^\top p/\|p\|$$

$$\leq \max_{\|r\|\leq 1} \min_{i\in[n]} \min_{t\neq\alpha_i}(k_{i\alpha_i} - k_{it})^\top (q + \delta r)$$

$$\leq \Gamma_q + M\delta,$$

where $M = \max_{i,t,\tau} \|k_{it} - k_{i\tau}\|$.

Consequently, setting $\delta = \frac{\epsilon}{4M}(\Gamma - \Gamma_q)$, we find that

$$\Gamma_{q'} \geq \Gamma_\delta + \frac{\epsilon}{4}(\Gamma - \Gamma_q).$$

Equipped with this inequality, we apply the first statement of Lemma 8 which concludes that[6] for some $R_\epsilon = R(\frac{\epsilon}{4}(\Gamma - \Gamma_q))$ and all $R > R_\epsilon$, (68) holds. This in turn implies that, within $C_\epsilon$, the optimal solution is

- either upper bounded by $R_\epsilon$ in $\ell_2$ norm (i.e. $\lim_{R\to\infty} \|\bar{p}(R)\| < \infty$) or
- at least $\delta = \delta(\epsilon) > 0$ away from $q$ after $\ell_2$-normalization i.e. $\|\frac{\bar{p}(R)}{\|\bar{p}(R)\|} - q\| \geq \delta$.

In either scenario, we have proven that $\lim_{R\to\infty} \frac{\bar{p}(R)}{R} \neq q$.

$\diamond$ **(A2) $q$ is not neighbor-optimal.** In this scenario, we will prove that $\bar{p}(R)$ is finite to obtain $\lim_{R\to\infty} \bar{p}(R)/R = 0 \neq q$. To start, assume that conic neighborhood $\epsilon$ of $q$ is small enough so that selected-tokens $\alpha$ remain unchanged within $C_\epsilon$. This is without generality because if directional convergence fails in a small neighborhood of $q$, it will fail in the larger neighborhood as well. Secondly, if $\lim_{R\to\infty} \|\bar{p}(R)\| \to \infty$ and $\bar{p}(R) \in C_\epsilon$, since softmax will eventually perfectly select $\alpha$ (i.e. assigning probability 1 on token indices $(i, \alpha_i)$), we would have

$$\lim_{R\to\infty} \mathcal{L}(\bar{p}(R)) = \mathcal{L}_\star = \frac{1}{n}\sum_{i=1}^{n} \ell(\gamma_{i\alpha_i}).$$

Note that, this is simply by selection of $\alpha$ and regardless of $\bar{p}(R)$ directionally converges to $q$. This means that, if there exists a finite $p \in C_\epsilon$ such that $\mathcal{L}(p) < \mathcal{L}_\star$ (i.e. outperforming the training loss of $\|\bar{p}(R)\| \to \infty$), then $\|\bar{p}(R)\| < \infty$. This would conclude the proof.

Thus, we will simply find such a $p$ obeying $\mathcal{L}(p) < \mathcal{L}_\star$. To this aim, we first prove the following lemma.

**Lemma 9** *Given a fixed unit Euclidean norm vector $p$, if all directional neighbors of $p$ consistently have higher scores for their associated selected tokens, i.e., $\gamma_{i\alpha_i} < \gamma_{i\beta}$ for all $(i, \alpha_i)$ and directional neighbor $(i, \beta)$, then there exists $\bar{R}$ such that for all $R > \bar{R}$,*

$$\mathcal{L}(R \cdot p) < \mathcal{L}_\star = \lim_{R\to\infty} \mathcal{L}(R \cdot p). \tag{69}$$

**Proof.** Define the maximum score difference $K_+ = \sup_{i\in[n],t\neq\alpha_i} |\gamma_{i\alpha_i} - \gamma_{it}|$. Also let $\mathcal{M}_p$ be the set of directional neighbors achieving the minimum margin $\Gamma_p$; see (61). Define $\Gamma^{it} = k_{i\alpha_i}^\top p - k_{it}^\top p$. Define $\delta_p$ to be the margin difference between the directional-neighbors and the second-most minimum-margin neighbors defined as

$$\delta_p = \min_{i\in[n],t\neq\alpha_i,(i,t)\notin\mathcal{M}_p} \Gamma_p^{it} - \Gamma_p. \tag{70}$$

---

[6]Here, we apply this lemma to compare $q'$ against all $p \in \mathcal{N}_\delta$. We can do this uniform comparison because the $R$ requirement in Lemma 8 only depends on the margin difference and global problem variables and not the particular choice of $p \in \mathcal{N}_\delta$.

To proceed, setting $\kappa = \min_{(j,\beta) \in \mathcal{M}_p} \gamma_{j\beta} - \gamma_{j\alpha_j} > 0$ and using (63), we can bound

$$\mathcal{L}(Rp) - \mathcal{L}_\star \leq -\frac{1}{n} \sum_{(j,\beta) \in \mathcal{M}_p} M_{j\beta}\, e^{-R\Gamma_p \kappa} + \frac{1}{n} \sum_{i \in [n], t \neq \alpha_i, (i,t) \notin \mathcal{M}_p} M_{it}\, e^{-R\Gamma_p^{it}}(\gamma_{i\alpha_i} - \gamma_{it})$$

$$\leq -\frac{1}{n} M_- e^{-R\Gamma_p \kappa} + M_+ K_+ T e^{-R(\Gamma_p + \delta_p)}.$$

Consequently, we have found that $\mathcal{L}(Rp) < \mathcal{L}_\star$ as soon as

$$\frac{1}{n} M_- e^{-R\Gamma_q \kappa} \geq M_+ K_+ T e^{-R(\Gamma_q + \delta_q)}.$$

This happens when $R \gtrsim 1/\delta_q$ (up to logarithmic terms) establishing the desired statement. ∎

Based on this lemma, what we need is constructing a perturbation to modify $q$'s directional neighbors and make sure all of them have strictly better scores than their associated selected-tokens. Note that, once we construct a new candidate (say $q_0 = q +$ perturbation), all sufficiently large scalings of $q_0$ will achieve $\mathcal{L}(R \cdot q_0) < \mathcal{L}_\star$. Thus, we can find a strictly better solution than $\mathcal{L}_\star$ for any norm lower bound $R_0$ – which is enforced within the definition of $C_\epsilon$.

**Lemma 10** *There are at most $d$ directional neighbors i.e. $|\mathcal{M}_q| \leq d$.*

**Proof.** Directional neighbors are indices $(i,t)$ obeying the inequality

$$(k_{i\alpha_i} - k_{it})^\top q = \Gamma_q.$$

Declare $D$ to be the matrix with rows obtained by these key differences $k_{it} - k_{i\alpha_i}$. $D \in \mathbb{R}^{M \times d}$ where $M = |\mathcal{M}_q|$. We then obtain $Dq = -\Gamma_q \mathbf{1}_M$ where $\mathbf{1}_M$ is the all ones vector. If $M > d$, the equality $Dq = -\Gamma_q \mathbf{1}_M$ cannot be satisfied because $\mathbf{1}_M$ is not in the range space of $D$ by our assumption of general key embedding positions. ∎

To proceed, $|\mathcal{M}_q| \leq d$ and let $D$ be as defined in the lemma above. $D$ is also full-rank by our assumption of general key positions. We use $D$ to construct a perturbation as follows. Let $\mathcal{M}_q^+ \subset \mathcal{M}_q$ be the set of directional neighbor with strictly higher scores than their associated selected-tokens. In other words, all $(j,\beta) \in \mathcal{M}_q^+$ obeys

$$\gamma_{j\beta} > \gamma_{j\alpha_j}.$$

Define the score difference $\kappa = \min_{(j,\beta) \in \mathcal{M}_q^+} \gamma_{j\beta} - \gamma_{j\alpha_j} > 0$. We know $\kappa > 0$ because $\alpha$ is not neighbor-optimal. Finally, define the indicator vector of $\mathbf{1}_+$ with same dimension as cardinality $|\mathcal{M}_q|$. $\mathbf{1}_+$ is 1 for the rows of $D$ corresponding to $\mathcal{M}_q^+$ and is 0 otherwise. Finally, set the perturbation as

$$q^\perp = D^\dagger \mathbf{1}_+.$$

where we used the full-rankness of $D$ during pseudo-inversion. To proceed, for a small $\epsilon_0 > 0$, consider the candidate direction $q_0 = q + \epsilon_0 q^\perp$. We pick $\epsilon_0 = O(\epsilon)$ sufficiently small to ensure $q_0 \in C_\epsilon$. To finalize, let us consider the margins of the tokens within $q_0$. Similar to Lemma 9, set $\delta_q = \min_{i \in [n], t \neq \alpha_i, (i,t) \notin \mathcal{M}_q} \Gamma_q^{it} - \Gamma_q > 0$. Let $\bar{\epsilon}_0 = \|q_0\| - 1 = \|q + \epsilon_0 q^\perp\| - 1$. Using definition of $q^\perp$, we have that

- For $(i,t) \in \mathcal{M}_q^+$, we achieve a margin of

$$(k_{i\alpha_i} - k_{it})^\top (q + \epsilon_0 q^\perp)/(1 + \bar{\epsilon}_0) = \frac{\Gamma_q - \epsilon_0}{1 + \bar{\epsilon}_0}.$$

- For $(i,t) \in \mathcal{M}_q^+$, we achieve a margin of $\frac{\Gamma_q}{1 + \bar{\epsilon}_0}$.

- For $(i,t) \notin \mathcal{M}_q$, setting $K = \|q^\perp\| \cdot \sup_{i,t,\tau} \|k_{i\alpha_i} - k_{it}\|$, we achieve a margin of at most

$$(k_{i\alpha_i} - k_{it})^\top (q + \epsilon_0 q^\perp)/(1 + \bar{\epsilon}_0) \geq \frac{\Gamma_q + \delta_q - \epsilon_0 K}{1 + \bar{\epsilon}_0}.$$

In short, since $\bar{\epsilon}_0 = O(\epsilon_0)$, setting $\epsilon_0$ sufficiently small guarantees that $\mathcal{M}_q^+$ is the set of directional neighbors of $q_0$. Since $\mathcal{M}_q^+$ has strictly higher scores than their associated selected-tokens, applying Lemma 9 on $q_0$ shows that, $\mathcal{L}(R \cdot q_0) < \mathcal{L}_\star$ for sufficiently large $R$ implying $\|\bar{p}(R)\| < \infty$.

• **(B) $q$ selects multiple tokens for some inputs $i \in [n]$:** In this setting, we will again construct a perturbation to create a scenario where $q_0 = q + \epsilon q^\perp$ selects a single token for each input $i \in [n]$. We will then employ margin analysis (first statement of Lemma 8) to conclude that $q_0$ outperforms a $\delta \ll \epsilon$ neighborhood of $q$.

Let $\mathcal{I} \subset [n]$ be the set of inputs for which $q$ selects multiple tokens. Specifically, for each $i \in \mathcal{I}$, there is $\mathcal{T}_i \subset [T]$ such that $|\mathcal{T}_i| \geq 2$ and for any $i \in \mathcal{I}$ and $\theta \in \mathcal{T}_i$,

$$k_{i\theta}^\top q = \arg\max_{t \in [T]} k_{it}^\top q.$$

From these multiply-selected token indices let us select the highest score one, namely, $\beta_i = \arg\max_{\theta \in \mathcal{T}_i} \gamma_{i\theta}$ for $i \in \mathcal{I}$. Now, define the unique optimal tokens for each input as $\alpha \in \mathbb{R}^n$ where $\alpha_i := \beta_i$ for $i \in \mathcal{I}$ and $\alpha_i = \arg\max_{t \in [T]} k_{it}^\top q$ for $i \notin \mathcal{I}$. Define $\mathcal{L}_\star = \frac{1}{n} \sum_{i=1}^n \ell(\gamma_{i\alpha_i})$ as earlier.

Secondly, we construct a perturbation $q^\perp$ to show that $q_0 = q + \epsilon_0 q^\perp$ can select tokens $\alpha$ asymptotically. To see this, define the matrix $D$ where each (unique) row is given by $k_{i\alpha_i} - k_{i\theta}$ where $\theta \in \mathcal{T}_i, \theta \neq \alpha_i$, $i \in \mathcal{I}$. Now note that, $(k_{i\alpha_i} - k_{i\theta})^\top q = 0$ for all $\theta \in \mathcal{T}_i, \theta \neq \alpha_i, i \in \mathcal{I}$. Since keys are in general positions, this implies that $D$ has at most $d - 1$ rows and, thus, its rows are linearly independent. Consequently, choose $q^\perp = D^\dagger \mathbf{1}$ where $\dagger$ denotes pseudo-inverse and $\mathbf{1}$ is the all ones vector. Also let $\Gamma_q$ be the margin of directional margin of $q$ that is

$$\Gamma_q = \min_{i \in [n], t \notin \mathcal{T}_i} (k_{i\alpha_i} - k_{it})^\top q.$$

With this choice and setting $K = \sup_{i,t,\tau} \|k_{i\alpha_i} - k_{it}\|$, we have that

- For $\theta \in \mathcal{T}_i, \theta \neq \alpha_i$: $(k_{i\alpha_i} - k_{i\theta})^\top q_0 = (k_{i\alpha_i} - k_{i\theta})^\top q^\perp = \epsilon_0$.
- For all other $(i, t)$ with $t \neq \alpha_i$: $(k_{i\alpha_i} - k_{it})^\top q_0 \geq \Gamma_q - K\|q^\perp\|\epsilon_0$.

Choosing $\epsilon_0 < \Gamma_q/(1 + K\|q^\perp\|)$, together, these imply that,

- $\alpha = (\alpha_i)_{i=1}^n$ is the selected-tokens of $q_0$,
- $q_0$ achieves a directional margin of

$$\Gamma_{q_0} = \frac{\epsilon_0}{\|q_0\|} \geq \frac{\epsilon_0}{1 + \epsilon_0\|q^\perp\|},$$

- $q_0$ is neighbor-optimal because directional neighbors are $\theta \in \mathcal{T}_i, \theta \neq \alpha_i$ and $\gamma_{i\theta} < \gamma_{i\alpha_i}$.

Note that, these conditions lay the groundwork for applying the first statement of Lemma 8 with $p \leftarrow q_0$. We next explore the optimal directions within $\mathcal{N}_\delta$ and show that $q_0$ strictly outperform them in terms of training loss.

To proceed, given small $\delta \ll \epsilon_0$, let us study $\mathcal{L}_R = \min_{p \in \mathcal{N}_\delta, R \leq \|p\| < \infty} \mathcal{L}(p)$. Here, recall that $p$ has a $\delta$-small directional perturbation around $q$ which can modify the token selections by breaking the ties between the multiply-selected token indices by $q$. However, thanks to the distinct token score assumption, for large $R \leq \|p\|$, the optimal $p \in \mathcal{N}_\delta$ is guaranteed to (uniquely) select indices $\alpha_i \in \mathcal{T}_i$. Because all other $p$ directions – which, asymptotically, either select other tokens $\theta \in \mathcal{T}_i, \theta \neq \alpha_i$ or split the probabilities equally across a subset of $\mathcal{T}_i$ – achieve a larger loss. For instance, $q$ will split the probabilities equally across $\mathcal{T}_i$ to achieve an asymptotic loss of

$$\mathcal{L}_\star^q := \lim_{R \to \infty} \mathcal{L}(R \cdot q) = \frac{1}{n} \sum_{i \notin \mathcal{I}} \ell(\gamma_{i\alpha_i}) + \frac{1}{n} \sum_{i \in \mathcal{I}} \ell\left(\frac{1}{|\mathcal{T}_i|} \sum_{\theta \in \mathcal{T}_i} \gamma_{i\theta}\right)$$

Thus, $\mathcal{L}_\star^q > \mathcal{L}_\star$ because $\ell\left(\frac{1}{|\mathcal{T}_i|} \sum_{\theta \in \mathcal{T}_i} \gamma_{i\theta}\right) > \ell(\gamma_{i\alpha_i}) = \ell(\gamma_{i\alpha_i})$ where $\alpha_i$ has the highest score i.e. $\gamma_{i\alpha_i} > \frac{1}{|\mathcal{T}_i|} \sum_{\theta \in \mathcal{T}_i} \gamma_{i\theta}$. Set $\tilde{p}(R) = \arg\min_{p \in \mathcal{N}_\delta, \|p\| \leq R} \mathcal{L}(p)$. Consequently, there are two scenarios are:

- $\lim_{R \to \infty} \|\tilde{p}(R)\|$ is finite. This already proves the statement of the theorem as $\tilde{p}(R)/R \to 0$ within $\delta < \epsilon$ neighborhood of $q$.

- For sufficiently large $R$, the selected-tokens of $\tilde{p}(R)$ are $\alpha = (\alpha_i)_{i=1}^n$.

Proceeding with the second (remaining scenario), we study the directional margin of $\tilde{p}(R)$. More broadly, for any $p \in \mathcal{N}_\delta$ and $\bar{p} = p/\|p\|$ with selected-tokens $\alpha$, using the fact that $\|\bar{p} - q\| \leq \delta \ll \epsilon_0$, we can bound the directional margin as

- For $\theta \in \mathcal{T}_i, \theta \neq \alpha_i$: $(k_{i\alpha_i} - k_{i\theta})^\top \bar{p} = (k_{i\alpha_i} - k_{i\theta})^\top (\bar{p} - q) \leq K\delta$.
- For all other $(i, t)$ with $t \neq \alpha_i$: $(k_{i\alpha_i} - k_{it})^\top \bar{p} \geq \Gamma_q - K\delta$.

This means that, any such $\bar{p} \in \mathcal{N}_\delta$ achieves a directional margin of at most

$$\Gamma_{\bar{p}} \leq K\delta.$$

Applying Lemma 8 and setting $\delta = O(\epsilon_0)$, this implies that for

$$R \gtrsim \frac{1}{\Gamma_{q_0} - \Gamma_{\bar{p}}} = \frac{1}{\frac{\epsilon_0}{1+\epsilon_0\|q^\perp\|} - K\delta} = O(\frac{1}{\epsilon_0}),$$

we have that $\mathcal{L}(R \cdot q_0) < \min_{\|p\|=R, p\in\mathcal{N}_\delta} \mathcal{L}(p)$. Since this holds for all $R$, (68) holds (similar to Case **(A1)**) and we conclude that whenever $\|\bar{p}(R)\| \to \infty$, it doesn't directionally converge within $\mathcal{N}_\delta$ (i.e. $\delta > 0$ neighborhood of $q$) proving the advertised result. ∎

## B.6 Proof of Lemma 2

We prove a slightly general restatement where we require $v \in \text{range}(W^\top)$ – instead of full-rank $W$.

**Lemma 11** *Suppose for all $i \in [n]$ and $t \neq \text{opt}_i$, $Y_i = 1$ and $\gamma_{it} < \gamma_{i\text{opt}_i}$. Also suppose $v \in \text{range}(W^\top)$. Then, $p^{mm\star}$ exists – i.e. (ATT-SVM) is feasible for optimal indices $\alpha_i \leftarrow \text{opt}_i$.*

**Proof.** To establish the existence of $p^{mm\star}$, we only need to find a direction that demonstrates the feasibility of (ATT-SVM), i.e. we need to find $p$ that satisfies the margin constraints. To begin, let's define the minimum score difference:

$$\underline{\gamma} = \min_{i\in[n],t\neq\text{opt}_i} \gamma_{i\text{opt}_i} - \gamma_{it}.$$

We then set $p = \underline{\gamma}^{-1}(W^\top)^\dagger v$ where $\dagger$ denotes pseudo-inverse. By assumption $W^\top p = \underline{\gamma}^{-1}v$. To conclude, observe that $p$ is a feasible solution since $k_{it} = Wx_{it}$ and for all $i \in [n]$ and $t \neq \text{opt}_i$, we have that

$$(k_{i\text{opt}_i} - k_{it})^\top p = (x_{i\text{opt}_i} - x_{it})^\top W^\top p = \underline{\gamma}^{-1}(x_{i\text{opt}_i} - x_{it})^\top W^\top(W^\top)^\dagger v$$
$$= \underline{\gamma}^{-1}(x_{i\text{opt}_i} - x_{it})^\top v \geq 1,$$

which together with the constraints in (ATT-SVM) completes the proof. ∎

# C  Addendum to Section 3

## C.1 Proof of Theorem 5

**Proof.** Suppose the claim is incorrect and either $p_R/R$ or $v_r/r$ fails to converge as $R, r$ grows. Set $\Xi = 1/\|p^{mm}\|$, $\Gamma = 1/\|v^{mm}\|$, $\tilde{p}^{mm} = R\Xi p^{mm}$ and $\tilde{v}^{mm} = r\Gamma v^{mm}$. The proof strategy is obtaining a contradiction by proving that $(\tilde{v}^{mm}, \tilde{p}^{mm})$ is a strictly better solution compared to $(v_r, p_R)$ for large $R, r$. Without losing generality, we will set $\alpha_i = 1$ for all $i \in [n]$ as the problem is invariant to tokens' permutation. Define $q_i^p = 1 - s_{i1}^p$ to be the amount of non-optimality (cumulative probability of non-first tokens) where $s_i^p = \mathbb{S}(K_i p)$ is the softmax probabilities.

• **Case 1: $p_R/R$ does not converge.** Under this scenario there exists $\delta, \gamma = \gamma(\delta) > 0$ such that we can find arbitrarily large $R$ with $\|p_R/R - \tilde{p}^{mm}/R\| \geq \delta$ and margin induced by $p_R/R$ is at most $\Xi(1 - \gamma)$ (from strong convexity of (ATT-SVM)). Following $q_i^p$ definition above, set $\hat{q}_{\max} = \sup_{i\in[n]} q_i^{p_R}$ to be

worst non-optimality in $\boldsymbol{p}_R$ and $q_{\max}^{\star} = \sup_{i\in[n]} q_i^{\tilde{\boldsymbol{p}}^{mm}}$ to be the same for $\tilde{\boldsymbol{p}}^{mm}$. Repeating the identical argument in Theorem 8 (specifically (84)), we can bound the non-optimality amount $q_i^{\tilde{\boldsymbol{p}}^{mm}}$ of $\tilde{\boldsymbol{p}}^{mm}$ as

$$q_i^{\tilde{\boldsymbol{p}}^{mm}} = \frac{\sum_{t\neq\alpha_i} \exp(\boldsymbol{k}_{it}^{\top}\tilde{\boldsymbol{p}}^{mm})}{\sum_{t\in[T]} \exp(\boldsymbol{k}_{it}^{\top}\tilde{\boldsymbol{p}}^{mm})} \leq \frac{\sum_{t\neq\alpha_i} \exp(\boldsymbol{k}_{it}^{\top}\tilde{\boldsymbol{p}}^{mm})}{\exp(\boldsymbol{k}_{i\alpha_i}^{\top}\tilde{\boldsymbol{p}}^{mm})} \leq T\exp(-R\Xi). \tag{71}$$

Thus, $q_{\max}^{\star} = \max_{i\in[n]} q_i^{\tilde{\boldsymbol{p}}^{mm}} \leq T\exp(-R\Xi)$. Next without losing generality, assume first margin constraint is $\gamma$-violated by $\boldsymbol{p}_R$ and $\min_{t\neq\alpha_1}(\boldsymbol{k}_{1\alpha_1} - \boldsymbol{k}_{1t})^{\top}\boldsymbol{p}_R \leq \Xi R(1-\gamma)$. Denoting the amount of non-optimality of the first input as $q_1^{\boldsymbol{p}_R}$, we find

$$q_1^{\boldsymbol{p}_R} = \frac{\sum_{t\neq\alpha_1} \exp(\boldsymbol{k}_{1t}^{\top}\boldsymbol{p}_R)}{\sum_{t\in[T]} \exp(\boldsymbol{k}_{1t}^{\top}\boldsymbol{p}_R)} \geq \frac{1}{T} \frac{\sum_{t\neq\alpha_1} \exp(\boldsymbol{k}_{1t}^{\top}\boldsymbol{p}_R)}{\exp(\boldsymbol{k}_{1\alpha_1}^{\top}\boldsymbol{p}_R)} \geq T^{-1}\exp(-(1-\gamma)R\Xi). \tag{72}$$

We similarly have $q_{\max}^{\star} \geq T^{-1}\exp(-R\Xi)$ to find that

$$\log(\hat{q}_{\max}) \geq -(1-\gamma)\Xi R - \log T,$$
$$-\Xi R - \log T \leq \log(q_{\max}^{\star}) \leq -\Xi R + \log T. \tag{73}$$

In words, $\tilde{\boldsymbol{p}}^{mm}$ contains exponentially less non-optimality compared to $\boldsymbol{p}_R$ as $R$ grows. The remainder of the proof differs from Theorem 8 as we need to upper/lower bound the logistic loss of $(\tilde{\boldsymbol{v}}^{mm}, \tilde{\boldsymbol{p}}^{mm})$ and $(\boldsymbol{v}_r, \boldsymbol{p}_R)$ respectively to conclude with the contradiction.

First, let us upper bound the logistic loss of $(\tilde{\boldsymbol{v}}^{mm}, \tilde{\boldsymbol{p}}^{mm})$. Set $\boldsymbol{r}_i = \boldsymbol{X}_i^{\top}\mathbb{S}(\boldsymbol{K}_i\tilde{\boldsymbol{p}}^{mm})$. Observe that if $\|\boldsymbol{r}_i - \boldsymbol{x}_{i1}\| \leq \epsilon_i$, we have that $\boldsymbol{v}^{mm}$ satisfies the SVM constraints on $\boldsymbol{r}_i$ with $Y_i \cdot \boldsymbol{r}_i^{\top}\boldsymbol{v}^{mm} \geq 1 - \epsilon_i/\Gamma$. Consequently, setting $\epsilon_{\max} = \sup_{i\in[n]} \epsilon_i$, $\boldsymbol{v}^{mm}$ achieves a label-margin of $\Gamma - \epsilon_{\max}$ on the dataset $(Y_i, \boldsymbol{r}_i)_{i\in[n]}$. With this, we upper bound the logistic loss of $(\tilde{\boldsymbol{v}}^{mm}, \tilde{\boldsymbol{p}}^{mm})$ as follows. Let $M = \sup_{i\in[n],t,\tau\in[T]} \|\boldsymbol{x}_{it} - \boldsymbol{x}_{i\tau}\|$. Let us recall the fact (73) that worst-case perturbation is

$$\epsilon_{\max} \leq M\exp(-\Xi R + \log T) = MT\exp(-\Xi R).$$

This implies that

$$\begin{aligned}
\mathcal{L}(\tilde{\boldsymbol{v}}^{mm}, \tilde{\boldsymbol{p}}^{mm}) &\leq \max_{i\in[n]} \log(1 + \exp(-Y_i\boldsymbol{r}_i^{\top}\tilde{\boldsymbol{v}}^{mm})). \\
&\leq \max_{i\in[n]} \exp(-Y_i\boldsymbol{r}_i^{\top}\tilde{\boldsymbol{v}}^{mm}) \\
&\leq \exp(-r\Gamma + r\epsilon_{\max}) \\
&\leq e^{rMT\exp(-\Xi R)}e^{-r\Gamma}.
\end{aligned} \tag{74}$$

Conversely, we obtain a lower bound for $(\boldsymbol{v}_r, \boldsymbol{p}_R)$. Set $\boldsymbol{r}_i = \boldsymbol{X}_i^{\top}\mathbb{S}(\boldsymbol{K}_i\boldsymbol{p}_R)$. Using Assumption C, we find that solving (SVM) on $(Y_i, \boldsymbol{r}_i)_{i\in[n]}$ achieves at most $\Gamma - \nu e^{-(1-\gamma)\Xi R}/T$ margin. Consequently, we have

$$\begin{aligned}
\mathcal{L}(\boldsymbol{v}_r, \boldsymbol{p}_R) &\geq \frac{1}{n} \max_{i\in[n]} \log(1 + \exp(-Y_i\boldsymbol{r}_i^{\top}\boldsymbol{v}_r)) \\
&\geq \frac{1}{2n} \max_{i\in[n]} \exp(-Y_i\boldsymbol{r}_i^{\top}\boldsymbol{v}_r) \wedge \log 2 \\
&\geq \frac{1}{2n} \exp(-r(\Gamma - \nu e^{-(1-\gamma)\Xi R}/T)) \wedge \log 2 \\
&\geq \frac{1}{2n} e^{r(\nu/T)\exp(-(1-\gamma)\Xi R)}e^{-r\Gamma} \wedge \log 2.
\end{aligned} \tag{75}$$

Observe that, this lower bound dominates the previous upper bound when $R$ is large, namely, when (ignoring the multiplier $1/2n$ for brevity)

$$(\nu/T)e^{-(1-\gamma)\Xi R} \geq MTe^{-\Xi R} \iff R \geq R_0 := \frac{1}{\gamma\Xi} \log\left(\frac{MT^2}{\nu}\right).$$

Thus, we indeed obtain the desired contradiction since such large $R$ is guaranteed to exist when $\boldsymbol{p}_R/R \nrightarrow \boldsymbol{p}^{mm}$.

• **Case 2: $\boldsymbol{v}_r/r$ does not converge.** This is the simpler scenario: There exists $\delta > 0$ such that we can find arbitrarily large $r$ obeying $\|\boldsymbol{v}_r/r - \boldsymbol{v}^{mm}/\|\boldsymbol{v}^{mm}\|\| \geq \delta$. If $\|\boldsymbol{p}_R/R - \Xi\boldsymbol{p}^{mm}\| \nrightarrow 0$, then "Case

1" applies. Otherwise, we have $\|\boldsymbol{p}_R/R - \Xi\boldsymbol{p}^{mm}\| \to 0$, thus we can assume $\|\boldsymbol{p}_R/R - \Xi\boldsymbol{p}^{mm}\| \le \epsilon$ for arbitrary choice of $\epsilon > 0$.

On the other hand, due to the strong convexity of (SVM), for some $\gamma := \gamma(\delta) > 0$, $\boldsymbol{v}_r$ achieves a margin of at most $(1 - \gamma)\Gamma r$ on the dataset $(Y_i, \boldsymbol{x}_{i1})_{i\in[n]}$. Additionally, since $\|\boldsymbol{p}_R/R - \Xi\boldsymbol{p}^{mm}\| \le \epsilon$, $\boldsymbol{p}_R$ strictly separates all optimal tokens (for small enough $\epsilon > 0$) and $\hat{q}_{\max} := f(\epsilon) \to 0$ as $R \to \infty$. Consequently, setting $\boldsymbol{r}_i = X_i^\top \mathbb{S}(K_i\boldsymbol{p}_R)$, for sufficiently large $R > 0$ setting $M = \sup_{i\in[n],t\in[T]} \|\boldsymbol{x}_{it}\|$, we have that

$$
\begin{aligned}
\min_{i\in[n]} Y_i \boldsymbol{v}_r^\top \boldsymbol{r}_i &\le \min_{i\in[n]} Y_i \boldsymbol{v}_r^\top \boldsymbol{x}_{i1} + \sup_{i\in[n]} |\boldsymbol{v}_r^\top (\boldsymbol{r}_i - \boldsymbol{x}_{i1})| \\
&\le (1 - \gamma)\Gamma r + M f(\epsilon) r \\
&\le (1 - \gamma/2)\Gamma r.
\end{aligned}
\tag{76}
$$

This in turn implies that logistic loss is lower bounded by (following (75)),

$$
\mathcal{L}(\boldsymbol{v}_r, \boldsymbol{p}_R) \ge \frac{1}{2n} e^{\gamma\Gamma r/2} e^{-\Gamma r} \wedge \log 2.
$$

Going back to (74), this exponentially dominates the upper bound of $(\tilde{\boldsymbol{p}}^{mm}, \tilde{\boldsymbol{v}}^{mm})$ whenever $rMT \exp(-\Xi R) < r\gamma\Gamma/2$, (that is, whenever $R, r$ are sufficiently large), again concluding the proof. ∎

## C.2 Proof of Theorem 6

We will prove this result in two steps. Our first claim restricts the optimization to the particular quadrant induced by $\min_{t\neq\alpha_i}(\boldsymbol{k}_{i\alpha_i} - \boldsymbol{k}_{it})^\top \boldsymbol{p}_R \ge 0$ under the theorem's condition $\mathbb{S}(K_i\boldsymbol{p}_R)_{\alpha_i} \to 1$.

**Lemma 12** *Suppose $\mathbb{S}(K_i\boldsymbol{p}_R)_{\alpha_i} \to 1$. Then, there exists $R_0$ such that for all $R \ge R_0$, we have that,*

$$
\min_{t\neq\alpha_i} (\boldsymbol{k}_{i\alpha_i} - \boldsymbol{k}_{it})^\top \boldsymbol{p}_R \ge 0, \quad \text{for all} \quad i \in [n].
\tag{77}
$$

**Proof.** Suppose the claim does not hold. Set $s_i^R = \mathbb{S}(K_i\boldsymbol{p}_R)$. Fix $R_0$ such that $s_{i\alpha_i}^R \ge 0.9$ for all $R \ge R_0$. On the other hand, there exists arbitrarily large $R$ for which $(\boldsymbol{k}_{i\alpha_i} - \boldsymbol{k}_{it})^\top \boldsymbol{p}_R < 0$ for some $t \neq \alpha_i \in [T]$ and $i \in [n]$. At this $(R, i, t)$ choices, we have that $s_{it}^R \ge s_{i\alpha_i}^R$. Since $s_{it}^R + s_{i\alpha_i}^R \le 1$, we find $s_{i\alpha_i}^R < 0.5$ which contradicts with $s_{i\alpha_i}^R \ge 0.9$. ∎

Let $Q$ be the set of $\boldsymbol{p}$ satisfying the quadrant constraint (77) – i.e. indices $(\alpha_i)_{i=1}^n$ are selected. Let $\boldsymbol{h}_R$ be the solution of regularization path of $(\boldsymbol{v}, \boldsymbol{p})$ subject to the constraint $\boldsymbol{p} \in Q$. From Lemma 12, we know that, for some $R_0$ and all $R \ge R_0$, $\boldsymbol{h}_R = \boldsymbol{p}_R$. Thus, if the limit exists, we have that $\lim_{R\to\infty} \boldsymbol{h}_R/R = \lim_{R\to\infty} \boldsymbol{p}_R/R$.

To proceed, we will prove that $\lim_{R\to\infty} \boldsymbol{h}_R/R$ exists and is equal to $\boldsymbol{p}^{relax}/\|\boldsymbol{p}^{relax}\|$ and simultaneously establish $\boldsymbol{v}_r/r \to \boldsymbol{v}^{mm}/\|\boldsymbol{v}^{mm}\|$.

**Lemma 13** $\lim_{R\to\infty} \boldsymbol{h}_R/R = \boldsymbol{p}^{relax}/\|\boldsymbol{p}^{relax}\|$ and $\lim_{r\to\infty} \boldsymbol{v}_r/r = \boldsymbol{v}^{mm}/\|\boldsymbol{v}^{mm}\|$.

**Proof.** The proof will be similar to that of Theorem 5. As usual, we aim to show that SVM-solutions constitute the most competitive direction. Set $\Xi = 1/\|\boldsymbol{p}^{relax}\|$.

• **Case 1: $\boldsymbol{h}_R/R$ does not converge.** Under this scenario there exists $\delta, \gamma = \gamma(\delta) > 0$ such that we can find arbitrarily large $R$ with $\|\boldsymbol{h}_R/R - \Xi\boldsymbol{p}^{relax}\| \ge \delta$. This implies that margin induced by $\boldsymbol{h}_R/R$ is at most $\Xi(1 - \gamma)$ over the support vectors $\mathcal{S}$ (from strong convexity of (10)). The reason is that, $\boldsymbol{h}_R$ satisfies $\boldsymbol{h}_R^\top(\boldsymbol{k}_{i\alpha_i} - \boldsymbol{k}_{it}) \ge 0$ for all $t \neq \alpha_i$ by construction as $\boldsymbol{h}_R \in Q$. Thus, a constraint over the support vectors have to be violated (when normalized to the same $\ell_2$ norm as $\|\boldsymbol{p}^{relax}\| = 1/\Xi$).

As usual, we will construct a solution strictly superior to $\boldsymbol{h}_R$ and contradicts with its optimality.

**Construction of competitor:** Rather than using $\boldsymbol{p}^{relax}$ direction, we will choose a slightly deviating direction that ensures the selection of the correct tokens over non-supports $\bar{\mathcal{S}}$. Specifically, consider the solution of (10) where we tighten the non-support constraints by arbitrarily small $\epsilon > 0$.

$$
\boldsymbol{p}^{\epsilon\text{-rlx}} = \arg\min_{\boldsymbol{p}} \|\boldsymbol{p}\| \quad \text{such that} \quad \boldsymbol{p}^\top(\boldsymbol{k}_{i\alpha_i} - \boldsymbol{k}_{it}) \ge \begin{cases} 1 & \text{for all} \quad t \neq \alpha_i, \ i \in \mathcal{S} \\ \epsilon & \text{for all} \quad t \neq \alpha_i, \ i \in \bar{\mathcal{S}} \end{cases}.
\tag{78}
$$

Let $p^{mm}$ be the solution of (ATT-SVM) with $\alpha = (\alpha_i)_{i=1}^n$ (which was assumed to be separable). Observe that $p_\epsilon^{mm} = \epsilon p^{mm} + (1-\epsilon)p^{relax}$ satisfies the constraints of (78). Additionally, $p_\epsilon^{mm}$ would achieve a margin of $\frac{1}{(1-\epsilon)/\Xi+\epsilon/\Delta} = \frac{\Delta\Xi}{\Delta+\epsilon(\Xi-\Delta)}$ where $\Delta = 1/\|p^{mm}\|$. Using optimality of $p^{\epsilon\text{-}rlx}$, this implies that the reduced margin $\Xi_\epsilon = 1/\|p^{\epsilon\text{-}rlx}\|$ (by enforcing $\epsilon$ over non-support) over the support vectors is a Lipschitz function of $\epsilon$. That is $\Xi_\epsilon \geq \Xi - \epsilon M$ for some $M \geq 0$. To proceed, choose an $\epsilon > 0$ such that, it is strictly superior to margin induced by $h_R$, that is,

$$\Xi_\epsilon \geq \Xi\left(1 - \frac{\gamma}{2}\right).$$

To proceed, set $\tilde{p}^{\epsilon\text{-}rlx} = R\Xi_\epsilon p^{\epsilon\text{-}rlx}$. Let us recall the following notation from the proof of Theorem 5: $s_i^p = \mathbb{S}(K_i p)$ and $q_i^p = 1 - s_{i\alpha_i}$. Set $\hat{q}_{\max} = \max_{i \in \mathcal{S}} q_i^{h_R}$ to be worst non-optimality of $h_R$ over **support set**. Similarly, define $q_{\max}^\star = \max_{i \in \mathcal{S}} q_i^{\tilde{p}^{\epsilon\text{-}rlx}}$ to be the same for $\tilde{p}^{\epsilon\text{-}rlx}$. Repeating the identical arguments to (71), (72), (73), and using the fact that $p^{\epsilon\text{-}rlx}$ achieves a margin $\Xi(1 - \frac{\gamma}{2}) \leq \Xi_\epsilon \leq \Xi$, we end up with the lines

$$\log(\hat{q}_{\max}) \geq -(1-\gamma)\Xi R - \log T, \tag{79a}$$
$$-\Xi R - \log T \leq \log(q_{\max}^\star) \leq -\Xi(1 - 0.5\gamma)R + \log T. \tag{79b}$$

In what follows, we will prove that $\tilde{p}^{\epsilon\text{-}rlx}$ achieves a strictly smaller logistic loss contradicting with the optimality of $p_R$ (whenever $\|h_R/R - \Xi p^{relax}\| \geq \delta$).

**Upper bounding logistic loss.** Let us now upper bound the logistic loss of $(\tilde{v}^{mm}, \tilde{p}^{\epsilon\text{-}rlx})$ where $\tilde{v}^{mm} = r\Gamma v^{mm}$ with $v^{mm}$ being the solution of (SVM) with $r_i \leftarrow x_{i\alpha_i}$ and $\Gamma = 1/\|v^{mm}\|$. Set $r_i = X_i^\top \mathbb{S}(K_i \tilde{p}^{\epsilon\text{-}rlx})$. Set $v = \min_{i \in \bar{\mathcal{S}}} Y_i \cdot x_{i\alpha_i}^\top v^{mm} - 1$ to be the additional margin buffer that non-support vectors have access to. Also set $M = \sup_{i \in [n], t, \tau \in [T]} \|x_{it} - x_{i\tau}\|$. Observe that we can write

$$x_{i\alpha_i} - r_i = \sum_{t \neq \alpha_i} s_{it}(x_{i\alpha_i} - x_{it}) \implies \|x_{i\alpha_i} - r_i\| \leq q_i M.$$

Non-supports achieve strong label-margin: Using above and (78) for all $i \in \bar{\mathcal{S}}$ and $t \neq \alpha_i$, we have that $s_{it} \leq e^{-\epsilon\Xi_\epsilon R} s_{i\alpha_i} \leq e^{-\epsilon\Xi(1-\gamma/2)R} s_{i\alpha_i}$. Consequently, whenever $R \geq \bar{R}_0 := (\epsilon\Xi(1 - \gamma/2))^{-1} \log(\frac{TM}{\Gamma v})$,

$$q_i^{\tilde{p}^{\epsilon\text{-}rlx}} \leq \frac{\sum_{t \neq \alpha_i} s_{it}}{s_{i\alpha_i}} \leq T e^{-\epsilon\Xi(1-\gamma/2)R} \leq \frac{\Gamma v}{M}.$$

This implies that, on $i \in \bar{\mathcal{S}}$

$$Y_i \cdot r_i^\top v^{mm} \geq 1 + v + Y_i \cdot (r_i - x_{i\alpha_i})^\top v^{mm} \geq 1 + v - q_i M \|v^{mm}\| \geq 1. \tag{80}$$

*In words:* Above a fixed $\bar{R}_0$ that only depends on $\gamma = \gamma(\delta)$, features $r_i$ induced by all non-support indices $i \in \bar{\mathcal{S}}$ achieve margin at least 1. What remains is analyzing the margin shrinkage over the support vectors as in Theorem 5.

Controlling support margin and combining bounds: Over $\mathcal{S}$, suppose $v^{mm}$ satisfies the SVM constraints on $r_i$ with $Y_i \cdot r_i^\top v^{mm} \geq 1 - \epsilon_i/\Gamma$. Consequently, setting $\epsilon_{\max} = \sup_{i \in [n]} \epsilon_i$, $v^{mm}$ achieves a label-margin of $\Gamma - \epsilon_{\max}$ on the dataset $(Y_i, r_i)_{i \in [n]}$. Next, we recall the fact (79b) that worst-case perturbation is $\epsilon_{\max} \leq M \exp(-\Xi(1 - 0.5\gamma)R + \log T) = MT \exp(-\Xi(1 - 0.5\gamma)R)$. With this and (80), we upper bound the logistic loss of $(\tilde{v}^{mm}, \tilde{p}^{\epsilon\text{-}rlx})$ as follows.

$$\mathcal{L}(\tilde{v}^{mm}, \tilde{p}^{mm}) \leq \max_{i \in [n]} \log(1 + \exp(-Y_i r_i^\top \tilde{v}^{mm})).$$
$$\leq \max_{i \in [n]} \exp(-Y_i r_i^\top \tilde{v}^{mm})$$
$$\leq \exp(-r\Gamma + r\epsilon_{\max})$$
$$\leq e^{rMT \exp(-\Xi(1-0.5\gamma)R)} e^{-r\Gamma}. \tag{81}$$

Conversely, we obtain a lower bound for $(v_r, h_R)$. Set $r_i = X_i^\top \mathbb{S}(K_i h_R)$. Recall the lower bound (79a) over the support vector set $\mathcal{S}$. Combining this with our Assumption C over the support vectors

of (SVM) implies that, solving (SVM) on $(Y_i, r_i)_{i \in [n]}$ achieves at most $\Gamma - \nu e^{-(1-\gamma)\Xi R}/T$ margin. Consequently, we have

$$
\begin{aligned}
\mathcal{L}(v_r, h_R) &\geq \frac{1}{n} \max_{i \in [n]} \log(1 + \exp(-Y_i r_i^\top v_r)) \\
&\geq \frac{1}{2n} \max_{i \in [n]} \exp(-Y_i r_i^\top v_r) \wedge \log 2 \\
&\geq \frac{1}{2n} \exp(-r(\Gamma - \nu e^{-(1-\gamma)\Xi R}/T)) \wedge \log 2 \\
&\geq \frac{1}{2n} e^{r(\nu/T)\exp(-(1-\gamma)\Xi R)} e^{-r\Gamma} \wedge \log 2.
\end{aligned} \tag{82}
$$

Observe that, this lower bound dominates the previous upper bound when $R$ is large, namely, when (ignoring the multiplier $1/2n$ for brevity)

$$
(\nu/T)e^{-(1-\gamma)\Xi R} \geq MT e^{-\Xi(1-0.5\gamma)R} \iff R \geq R_0 := \frac{2}{\gamma\Xi} \log\left(\frac{MT^2}{\nu}\right).
$$

Thus, we obtain the desired contradiction since $\tilde{p}^{\epsilon\text{-}r|x}$ is a strictly better solution compared to $p_R = h_R$ (once $R$ is sufficiently large).

• **Case 2: $v_r/r$ does not converge.** This is the simpler scenario: There exists $\delta > 0$ such that we can find arbitrarily large $r$ obeying $\|v_r/r - v^{mm}/\|v^{mm}\|\| \geq \delta$. First, note that, due to the strong convexity of (SVM), for some $\gamma := \gamma(\delta) > 0$, $v_r$ achieves a margin of at most $(\Gamma - \gamma)r$ on the dataset $(Y_i, x_{i1})_{i \in [n]}$. By theorem's condition, we are provided that $\mathbb{S}(K_i p_R)_{\alpha_i} \to 1$. This immediately implies that, for any choice of $\epsilon = \gamma/3 > 0$, above some sufficiently large $(r_0, R_0)$, we have that $\|x_i^{p_R} - r_i\| \leq \epsilon$. Following (81), this implies that, choosing $\tilde{v}^{mm} = rv^{mm}/\|v^{mm}\|$ achieves a logistic loss of at most $e^{r\gamma/3}e^{-r\Gamma}$. Again using $\|x_i^{p_R} - r_i\| \leq \epsilon$, for sufficiently large $(r, R)$ we have that

$$
\begin{aligned}
\min_{i \in [n]} Y_i v_r^\top r_i &\leq \min_{i \in [n]} Y_i v_r^\top x_{i1} + \sup_{i \in [n]} |v_r^\top(r_i - x_{i1})| \\
&\leq (\Gamma - \gamma)r + \epsilon r \\
&\leq (\Gamma - 2\gamma/3)r.
\end{aligned}
$$

This in turn implies that logistic loss is lower bounded by (following (82)),

$$
\mathcal{L}(v_r, p_R) \geq \frac{1}{2n} e^{2\gamma r/3} e^{-r\Gamma} \wedge \log 2.
$$

This dominates the above upper bound $e^{r\gamma/3}e^{-r\Gamma}$ of $\tilde{v}^{mm}$ whenever $\frac{1}{2n} e^{\gamma r/3} > 1 \iff r > \frac{3}{\gamma} \log(2n)$, (that is, when $r$ is sufficiently large), again concluding the proof. ∎

## D    Regularization Path of Attention with Nonlinear Head

So far our discussion has focused on the attention model with linear head. However, the conceptual ideas on optimal token selection via margin maximization also extends to a general nonlinear model under mild assumptions. The aim of this section is showcasing this generalization. Specifically, we consider the prediction model $f(X) = \psi(X^\top \mathbb{S}(Kp))$ where $\psi(\cdot) : \mathbb{R}^d \to \mathbb{R}$ generalizes the linear head $v$ of our attention model. For instance, following exposition in Section 1.1, $\psi(\cdot)$ can represent a multilayer transformer with $p$ being a tunable prompt at the input layer. Recall that $(X_i, K_i, Y_i)_{i=1}^n$ is the dataset of the input-key-label tuples. We consider the training risk

$$
\mathcal{L}(p) = \frac{1}{n} \sum_{i=1}^n \ell(Y_i, \psi(X_i^\top s_i^p)), \quad \text{where} \quad s_i^p = \mathbb{S}(K_i p) \in \mathbb{R}^T. \tag{83}
$$

The challenge with nonlinear $\psi(\cdot)$ is that, we lack a clear score function (Def. 1) unlike the previous sections. The assumption below introduces a generic condition that splits the tokens of each $X_i$ into an *optimal* set $O_i$ and *non-optimal* set $\bar{O}_i = [T] - O_i$. In words, non-optimal tokens are those that strictly increase the training risk $\mathcal{L}(p)$ if they are not fully suppressed by attention probabilities $s_i^p$.

**Assumption D (Mixing non-optimal tokens hurt)** *There exists sets $(O_i)_{i=1}^n \subset [T]$ as follows. Let $q_i^{\boldsymbol{p}} = \sum_{t \in \bar{O}_i} s_{it}^{\boldsymbol{p}}$ be the sum of softmax similarities over the non-optimal set for $\boldsymbol{p}$. Set $q_{\max}^{\boldsymbol{p}} = \max_{i \in [n]} q_i^{\boldsymbol{p}}$. For any $\Delta > 0$, there exists $\rho < 0$ such that:*

$$\text{For all } \boldsymbol{p}, \boldsymbol{p}' \in \mathbb{R}^d, \text{ if } \log(q_{\max}^{\boldsymbol{p}}) \leq (1 + \Delta) \log(q_{\max}^{\boldsymbol{p}'}) \wedge \rho, \text{ then } \mathcal{L}(\boldsymbol{p}) < \mathcal{L}(\boldsymbol{p}').$$

This assumption is titled *mixing hurts* because the attention output $X_i^\top s_i^{\boldsymbol{p}}$ is mixing the tokens of $X_i$ and our condition is that, to achieve optimal risk, this mixture should not contain any non-optimal tokens. In particular, we require that, a model $\boldsymbol{p}$ that contains *exponentially less non-optimality* (quantified via $\log(q_{\max})$) compared to $\boldsymbol{p}'$ is strictly preferable. As we discuss in the supplementary material, Theorem 1 is in fact a concrete instance (with linear head $\boldsymbol{v}$) satisfying this condition.

Before stating our generic theorem, we need to introduce the max-margin separator towards which regularization path of attention will converge. This is a slightly general version of Section 2's (ATT-SVM) problem where we allow for a set of optimal tokens $O_i$ for each input.

$$\boldsymbol{p}^{mm} = \arg\min_{\boldsymbol{p}} \|\boldsymbol{p}\| \quad \text{subject to} \quad \max_{\alpha \in O_i} \min_{\beta \in \bar{O}_i} \boldsymbol{p}^\top (\boldsymbol{k}_{i\alpha} - \boldsymbol{k}_{i\beta}) \geq 1, \quad \text{for all} \quad i \in [n]. \quad \text{(ATT-SVM')}$$

Unlike (ATT-SVM), this problem is not necessarily convex when the optimal set $O_i$ is not a singleton. To see this, imagine $n = d = 1$ and $T = 3$: Set the two optimal tokens as $\boldsymbol{k}_1 = 1$ and $\boldsymbol{k}_2 = -1$ and the non-optimal token as $\boldsymbol{k}_3 = 0$. The solution set of (ATT-SVM') is $\boldsymbol{p}^{mm} \in \{-1, 1\}$ whereas their convex combination $\boldsymbol{p} = 0$ violates the constraints. To proceed, our final result establishes the convergence of regularization path to the solution set of (ATT-SVM') under Assumption D.

**Theorem 8** *Let $\mathcal{G}^{mm}$ be the set of global minima of (ATT-SVM'). Suppose its margin $\Xi := 1/\|\boldsymbol{p}^{mm}\| > 0$ and Assumption D holds. Let $\mathtt{dist}(\cdot, \cdot)$ denote the $\ell_2$-distance between a vector and a set. Following (83), define $\bar{\boldsymbol{p}}(R) = \arg\min_{\|\boldsymbol{p}\| \leq R} \mathcal{L}(\boldsymbol{p})$. We have that $\lim_{R \to \infty} \mathtt{dist}\left(\frac{\bar{\boldsymbol{p}}(R)}{\Xi R}, \mathcal{G}^{mm}\right) = 0$.*

We note that Theorem 1 is a corollary of this result where $O_i$'s and $\mathcal{G}^{mm}$ are singleton. Based on this result, with multiple optimal tokens, Theorem 1 would gracefully generalize to solve (ATT-SVM').

## D.1  Proof of Theorem 8

**Proof.** The key idea is showing that, thanks to the exponential tail of softmax-attention, (harmful) contribution of the non-optimal token with the minimum margin can dominate the contribution of all other tokens as $R \to \infty$. This high-level approach is similar to earlier works on implicit bias of gradient descent with logistic loss [31, 22].

Pick $\boldsymbol{p}^{mm} \in \mathcal{G}^{mm}$ and set $\boldsymbol{p}_R^\star = R\frac{\boldsymbol{p}^{mm}}{\|\boldsymbol{p}^{mm}\|}$. This will be the baseline model that $\boldsymbol{p}_R$ has to compete against. Also let $\bar{\boldsymbol{p}}_R = \frac{\boldsymbol{p}_R}{\Xi R}$. Now suppose $\mathtt{dist}(\bar{\boldsymbol{p}}_R, \mathcal{G}^{mm}) \not\to 0$ as $R \to \infty$. Then, there exists $\delta > 0$ such that, we can always find arbitrarily large $R$ obeying $\mathtt{dist}(\bar{\boldsymbol{p}}_R, \mathcal{G}^{mm}) \geq \delta$.

Since $\bar{\boldsymbol{p}}_R$ is $\delta > 0$ bounded away from $\mathcal{G}^{mm}$, and $\|\bar{\boldsymbol{p}}_R\| = \|\boldsymbol{p}^{mm}\|$, $\bar{\boldsymbol{p}}_R$ strictly violates at least one of the inequality constraints in (ATT-SVM'). Otherwise, we would have $\bar{\boldsymbol{p}}_R \in \mathcal{G}^{mm}$. Without losing generality, suppose $\bar{\boldsymbol{p}}_R$ violates the first margin constraint, that is, for some $\gamma := \gamma(\delta) > 0$, $\max_{\alpha \in O_1} \min_{\beta \in \bar{O}_1} \bar{\boldsymbol{p}}_R^\top (\boldsymbol{k}_{1\alpha} - \boldsymbol{k}_{1\beta}) \leq 1 - \gamma$. Now, we will argue that this will lead to a contradiction as $R \to \infty$ since we will show that $\mathcal{L}(\boldsymbol{p}_R^\star) < \mathcal{L}(\boldsymbol{p}_R)$ for sufficiently large $R$.

First, let us control $\mathcal{L}(\boldsymbol{p}_R^\star)$. We study $s_i^\star = \mathbb{S}(K_i \boldsymbol{p}_R^\star)$ and let $\alpha_i \in O_i$ be the index $\alpha$ in (ATT-SVM') for which $\text{margin}_i = \max_{\alpha \in O_i} \min_{\beta \in \bar{O}_i} (\boldsymbol{k}_{i\alpha} - \boldsymbol{k}_{i\beta})^\top \boldsymbol{p}^{mm} \geq 1$ is attained. Then, we bound the non-optimality amount $q_i^\star$ of $\boldsymbol{p}_R^\star$ as

$$q_i^\star = \frac{\sum_{t \in \bar{O}_i} \exp(\boldsymbol{k}_{it}^\top \boldsymbol{p}_R^\star)}{\sum_{t \in [T]} \exp(\boldsymbol{k}_{it}^\top \boldsymbol{p}_R^\star)} \leq \frac{\sum_{t \in \bar{O}_i} \exp(\boldsymbol{k}_{it}^\top \boldsymbol{p}_R^\star)}{\exp(\boldsymbol{k}_{i\alpha_i}^\top \boldsymbol{p}_R^\star)} \leq T \exp(-\Xi R).$$

Thus, $q_{\max}^\star = \max_{i \in [n]} q_i^\star \leq T \exp(-\Xi R)$. Secondly, we wish to control $\mathcal{L}(\boldsymbol{p}_R)$ by lower bounding the non-optimality in $\boldsymbol{p}_R$. Focusing on the first margin constraint, let $\alpha \in O_1$ be the index in (ATT-SVM') for which $\text{margin}_1 \leq 1 - \gamma$ is attained. Denoting the amount of non-optimality of the first input as $\hat{q}_1$, we find[7]

$$\hat{q}_1 = \frac{\sum_{t \in \bar{O}_1} \exp(\boldsymbol{k}_{1t}^\top \boldsymbol{p}_R)}{\sum_{t \in [T]} \exp(\boldsymbol{k}_{1t}^\top \boldsymbol{p}_R)} \geq \frac{1}{T} \frac{\sum_{t \in \bar{O}_1} \exp(\boldsymbol{k}_{1t}^\top \boldsymbol{p}_R)}{\exp(\boldsymbol{k}_{1\alpha}^\top \boldsymbol{p}_R)} \geq T^{-1} \exp(-\Xi R(1 - \gamma)).$$

---

[7]Here, we assumed margin is non-negative i.e. $\boldsymbol{k}_{1\alpha}^\top \boldsymbol{p}_R \geq \sup_{t \in \bar{O}_1} \boldsymbol{k}_{1t}^\top \boldsymbol{p}_R$. Otherwise, $\sup_{t \in [T]} \boldsymbol{k}_{1t}^\top \boldsymbol{p}_R$ is attained in $\bar{O}_1$ which implies $\hat{q}_1 \geq T^{-1}$. Thus, we can still use the identical inequality (84) with the choice $\gamma = 1$.

We similarly have $q_{\max}^\star \geq T^{-1}\exp(-\Xi R)$. In conclusion, for $\boldsymbol{p}_R, \boldsymbol{p}_R^\star$, denoting maximum non-optimality by $\hat{q}_{\max} \geq \hat{q}_1$ and $q_{\max}^\star$, we respectively obtained

$$\log(\hat{q}_{\max}) \geq -(1-\gamma)(\Xi R) - \log T,$$
$$-(\Xi R) - \log T \leq \log(q_{\max}^\star) \leq -(\Xi R) + \log T. \tag{84}$$

The above inequalities satisfy Assumption D as follows where $\boldsymbol{p} \leftarrow \boldsymbol{p}_R^\star$ and $\boldsymbol{p}' \leftarrow \boldsymbol{p}_R$: Set $R_0 = 3\gamma^{-1}\Xi^{-1}\log T$ so that $\log T = \frac{\gamma \Xi R_0}{3}$. Secondly, set $\rho_0 = -\Xi R_0 - \log T$. This way, $\rho_0 \geq \log(q_{\max}^\star)$ implies $R \geq R_0$ and $\log T \leq \frac{\gamma \Xi R}{3}$. Using the latter inequality, we bound the $\log T$ terms to obtain

- $\log(\hat{q}_{\max}) \geq -(1-2\gamma/3)(\Xi R)$, and
- $\log(q_{\max}^\star) \leq -(1-\gamma/3)(\Xi R)$.

To proceed, we pick $1 + \Delta = \frac{1-\gamma/3}{1-2\gamma/3}$ implying $\Delta := \frac{\gamma}{3-2\gamma}$. Finally, for this $\Delta$, there exists $\rho(\Delta)$ which we need to ensure $\log(\hat{q}_{\max}) \leq \rho(\Delta)$. This can be guaranteed by picking sufficiently large $R$ that ensures $\log(q_{\max}^\star) \leq -(1-\gamma/3)(\Xi R) \leq \rho(\Delta)$ to satisfy all conditions of Assumption D. Since such large $R$ exists by initial assumption $\mathtt{dist}(\bar{\boldsymbol{p}}_R, \mathcal{G}^{mm}) \nrightarrow 0$, Assumption D in turn implies that $\mathcal{L}(\boldsymbol{p}_R^\star) < \mathcal{L}(\boldsymbol{p}_R)$ contradicting with the optimality of $\boldsymbol{p}_R$ in (83). ■

## D.2  Application to Linearly-mixed Labels

The following example shows that if non-optimal tokens result in reduced score (in terms of the alignment of prediction and label), Assumption D holds. The high-level idea behind this lemma is that, if the optimal risk is achieved by setting $q_{\max}^{\boldsymbol{p}} = 0$, then, Assumption D will hold.

**Lemma 14 (Linear label mixing)** *Recall* $q_i^{\boldsymbol{p}} = \sum_{t \in \bar{O}_i} s_{it}^{\boldsymbol{p}}$ *from Assumption D. Suppose* $Y_i \in \{-1, 1\}$ *and*
$$Y_i \cdot \psi(X_i^\top \boldsymbol{s}_i^{\boldsymbol{p}}) = v_i(1 - q_i^{\boldsymbol{p}}) + Z_i,$$
*for some* $(v_i)_{i=1}^n > 0$. *Here* $Z_i = Z_i(\boldsymbol{p})$ *is the contribution of non-optimal tokens to prediction. For some* $C, \epsilon > 0$ *and for all* $\boldsymbol{p} \in \mathbb{R}^d$, *assume*
$$-Cq_{\max}^{\boldsymbol{p}} \leq Z_i \leq (1-\epsilon)v_i q_i^{\boldsymbol{p}}. \tag{85}$$
*Then, Assumption D holds for* $\mathcal{L}(\boldsymbol{p}) = \frac{1}{n}\sum_{i=1}^n \ell(Y_i \cdot \psi(X_i^\top \boldsymbol{s}_i^{\boldsymbol{p}}))$ *when* $\ell(\cdot)$ *is a strictly decreasing loss function with continuous derivative.*

**Proof.**  Recall the assumption $Y_i \cdot \psi(X_i^\top \boldsymbol{s}_i^{\boldsymbol{p}}) = v_i(1 - q_i^{\boldsymbol{p}}) + Z_i$ with $Z_i$ obeying (85). Let us also write the loss function
$$\mathcal{L}(\boldsymbol{p}) = \frac{1}{n}\sum_{i=1}^n \ell(v_i(1 - s_i^{\boldsymbol{p}}) + Z_i).$$

Define $q_{\max}^{\boldsymbol{p}} = \sup_{i \leq [n]} q_i^{\boldsymbol{p}}$. Let $M$ be the maximum absolute value of score over tokens. Let
$$B = \max_{|x| \leq M} -\ell'(x) \geq A = \min_{|x| \leq M} -\ell'(x) > 0.$$

Through Taylor's Theorem (integral remainder), we have that
$$B(q_i^{\boldsymbol{p}} v_i - Z_i) \geq \ell(v_i(1 - q_i^{\boldsymbol{p}}) + Z_i) - \ell(v_i) \geq A(q_i^{\boldsymbol{p}} v_i - Z_i) \geq \epsilon A v_i q_i^{\boldsymbol{p}}.$$

Set $\mathcal{L}_\star = \frac{1}{n}\sum_{i=1}^n \ell(v_i)$. Set $C_+ = B(C + \max_{i \in [n]} v_i)$ and $C_- = n^{-1}A\epsilon \min_{i \in [n]} v_i$. This also implies

$$C_+ q_{\max}^{\boldsymbol{p}} \geq \frac{1}{n}\sum_{i \in [n]} B(q_i^{\boldsymbol{p}} v_i - Z_i) \geq \mathcal{L}(\boldsymbol{p}) - \mathcal{L}_\star \geq \frac{1}{n}\sum_{i \in [n]} A(q_i^{\boldsymbol{p}} v_i - Z_i)$$
$$\geq \frac{1}{n}\sum_{i \in [n]} \epsilon A v_i q_i^{\boldsymbol{p}} \geq C_- q_{\max}^{\boldsymbol{p}}.$$

Thus, to prove $\mathcal{L}(\boldsymbol{p}') > \mathcal{L}(\boldsymbol{p})$, we simply need to establish the stronger statement $C_- q_{\max}^{\boldsymbol{p}'} > C_+ q_{\max}^{\boldsymbol{p}}$.

Going back to the condition of Assumption D, any $\log(q_{max}^{p}) \leq (1 + \Delta)\log(q_{max}^{p'})$ obeys $q_{max}^{p} \leq (q_{max}^{p'})^{1+\Delta}$ i.e. $q_{max}^{p'} \geq (q_{max}^{p})^{(1+\Delta)^{-1}}$. Following above, we wish to ensure $q_{max}^{p'} > \Theta q_{max}^{p}$ for such $(p, p')$ pairs where $\Theta = C_+/C_- > 1$. This is guaranteed by

$$(q_{max}^{p})^{(1+\Delta)^{-1}-1} > \Theta \iff \frac{\Delta}{1 + \Delta}\log(q_{max}^{p}) < -\log(\Theta).$$

The above is satisfied by choosing a $\rho(\Delta) := -2(1 + \Delta^{-1})\log(\Theta)$ in Assumption D. Thus, all $p, p'$ with $\log(q_{max}^{p}) \leq \rho = \rho(\Delta)$ satisfies the condition of Assumption D finishing the proof. ■

# E  Implementation Details and Additional Experiments

In this section, we provide implementation details and additional experiments.

• We build one attention layer using `PyTorch`. During training, we use SGD optimizer with learning rate 0.1 and train the model for 1000 iterations. To better visualize the convergence path, we normalize the gradient of $p$ (and $v$) at each iteration.

• Next, given the gradient solution $p$, we determine locally-optimal indices to be those with the highest softmax scores. Using these optimal indices, we utilize python package `cvxopt` to build and solve (ATT-SVM), and then get solution $p^{mm}$. After obtaining $p^{mm}$, we also verify that these indices satisfy our local-optimal definition. The examples we use in the paper are all trivial to verify (by construction).

• In Figures 3(a) and 3(b), $v^{mm}$ (blue dashed) is solved using python package `sklearn.svm` via (SVM) based on the given label information, and red dashed line represents $p^{relax}$ direction instead, which is the solution of (10). Note that in both figures, $v^{mm}/\|v^{mm}\| = [0, 1]$. Therefore, in Figure 3(a) all optimal tokens are support vectors and $p^{relax} = p^{mm}$. Whereas in Figure 3(b), yellow ⋆ is not a support vector and only needs to satisfy positive correlation with $p$. Gray dashed line displays the $p^{mm}$ direction.

**Failure of gradient descent's global convergence when** $n \geq 2$ (**refer to Theorem 2**). Figure 8 provides a counter-example demonstrating that the $n = 1$ restriction is indeed necessary and tight to guarantee global convergence of the gradient descent iterates $p(t + 1) = p(t) - \eta\nabla\mathcal{L}(p(t))$ on (ERM).

For this example, we use logistic loss in (ERM). We set $n = T = d = 2$, implying that there is only one non-optimal token, thus Assumption B is satisfied. The red and blue lines represent GMM and LMM solutions, respectively. We note that the green star and teal square indicate the locally-optimal tokens. Specifically, referring to the local optimality definition (Definition 2), for LMM solution ($p^{mm}$) represented by the blue line, the square teal token does not have any SVM-neighbors. The arrows indicate the two trajectories originating from different initializations. This demonstrates that the gradient descent iterates $p(t + 1) = p(t) - \eta\nabla\mathcal{L}(p(t))$ on (ERM) with two different initializations converge to two different SVM solutions (GMM and LMM). Results validate the necessity of $n = 1$ in Theorem 2 to provide the gradient descent convergence to $p^{mm\star}$ from any initialization.

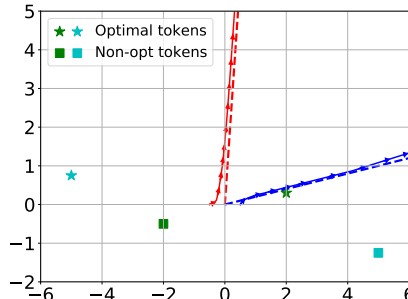

Figure 8: The convergence behavior of the gradient descent on the attention weights $p$ using the logistic loss in (ERM) with $n = T = d = 2$.

**The convergence behavior of gradient descent under over-parameterization.** To illustrate Theorems 3 & 4, we have investigated the convergence behavior of $p(t)$ generated by gradient descent in Figure 9(a), using $n = 4$, $T = 6$, and conducted 1,000 random trials for varying $d \in \{2, 5, 10, 100, 300, 500\}$. These experiments use normalized gradient descent with learning rate 1 for 1000 iterations. Inputs $x_{it}$ and the linear head $v$ are uniformly sampled from the unit sphere, while $Y_i$ is uniformly $\pm 1$, and $W$ is set to $I$.

The bar plot in Figure 9(a) distinguishes between *non-saturated* softmax (red bars) and *saturated* softmax (other bars). Saturation is defined as average softmax probability over tokens selected by gradient descent are at least $1 - 10^{-5}$ and implies that attention selects one token per input. Note that, whenever the norm of gradient descent solution is finite, softmax will be non-saturated. For

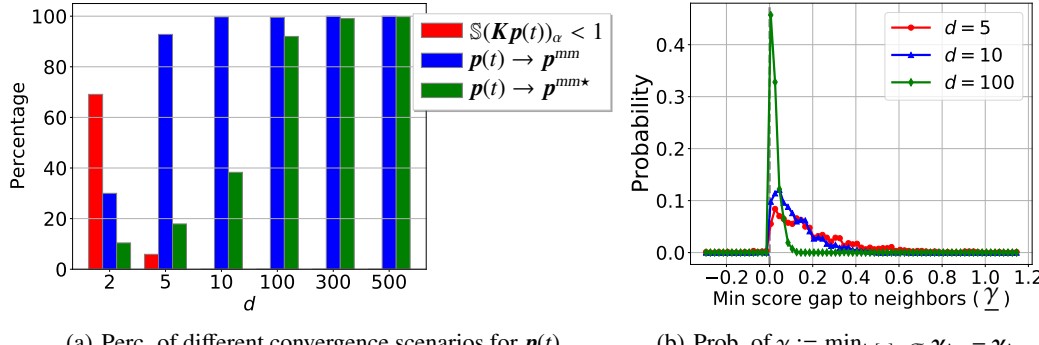

(a) Perc. of different convergence scenarios for $\boldsymbol{p}(t)$

(b) Prob. of $\underline{\gamma} := \min_{i \in [n], t \in \mathcal{T}_i} \boldsymbol{\gamma}_{i\alpha_i} - \boldsymbol{\gamma}_{it}$

Figure 9: Convergence analysis of $\boldsymbol{p}(t)$ trained with random data using gradient descent. **(a)** shows three scenarios: (1) attention failing to select one token per input (i.e. softmax is not saturated); (2) $\boldsymbol{p}$ converging towards $\boldsymbol{p}^{mm}$; and (3) $\boldsymbol{p}^{mm}$ equating to $\boldsymbol{p}^{mm\star}$ with red, blue, and green bars, respectively. Considering saturated softmax instances where $\boldsymbol{p}(t)$ selects one token $\alpha_i$ per-input, **(b)** presents histogram of the minimal score gap between $\alpha_i$ and its corresponding neighbors $\mathcal{T}_i$.

small $d$ (e.g., $d = 2$), problem has small degrees of freedom to separate optimal tokens from the rest (i.e. no SVM solution for LMM directions) – especially due to label randomness. This results in a tall red bar capturing the finite-norm solutions. However, for larger $d$, we observe that softmax saturates (i.e. $\|\boldsymbol{p}(t)\| \to \infty$) and we observe that the selected tokens $\boldsymbol{\alpha}$ almost always converges to an LMM direction (blue bar) – this is in line with Theorems 3 & 4. We also study the convergence to the globally-optimal GMM which is represented by the green bar: GMM is a strict subset of LMM however as $d$ increases, we observe that the probability of GMM convergence increases as well. This behavior is in line with what one would expect from over-parameterized deep learning theory [57, 58, 59, 60] and motivates future research. The average correlation coefficient between $\boldsymbol{p}(t)$ and its associated LMM/GMM direction is 0.997, suggesting that, whenever softmax saturates, gradient descent indeed directionally converges to a LMM solution $\boldsymbol{p} \in \mathcal{P}^{mm}$, confirming Theorem 4.

Furthermore, we found that there exist problem instances, with saturated softmax and $\|\boldsymbol{p}(t)\| \to \infty$, that do not converge to either LMM or GMM. We analyzed this phenomenon using the minimum score gap, $\underline{\gamma} := \min_{i \in [n], t \in \mathcal{T}_i} \boldsymbol{\gamma}_{i\alpha_i} - \boldsymbol{\gamma}_{it}$, where $\mathcal{T}_i, i \in [n]$, represents the sets of SVM-neighbor tokens. Figure 9(b) provides the probability distribution of $\gamma$ (with bins of width < 0.01) and demonstrates the rarity of such cases. Specifically, we found this happens less than 1% of the problems, that is, Prob($\underline{\gamma} < 0$) < 0.01. Figure 9(b) also reveals that, in these scenarios, even if $\underline{\gamma} < 0$, it is typically close to zero i.e. even if there exists a SVM-neighbor with a higher score, it is only slightly so. This is not surprising since when token scores are close, we need a large number of gradient iterations to distinguish them. For all practical purposes, the optimization will treat both tokens equally and rather than solving (ATT-SVM), the more refined formulation (ATT-SVM') developed in Section D will be a better proxy. Confirming this intuition, we have verified that, over the instances $\underline{\gamma} < 0$, gradient descent solution is still > 0.99 correlated with the max-margin solution in average.

• In Figure 9, we again applied normalized gradient descent with a learning rate equal to 1. Each trial involved randomly generated data and training for 1000 iterations as discussed in Theorem 4. The tokens selected by $\boldsymbol{p}$ were denoted as $(\alpha_i)_{i=1}^n$, where $\alpha_i = \arg\max_{t \in [T]} \mathbb{S}(\boldsymbol{X}_i \boldsymbol{p})_t$. The averaged softmax probabilities were calculated as $\bar{s} := \frac{1}{n} \sum_{i=1}^n \mathbb{S}(\boldsymbol{X}_i \boldsymbol{p})_{\alpha_i}$ (same as Figure 3(c)). The red bars in Figure 9(b) represent the values of $\mathbb{P}(\bar{s} \leq 1 - 10^{-5})$ for each choice of $d$. Figure 10 displays the cumulative probability distribution of $\gamma$ from Figure 9(b), with the gray dashed line indicating $\underline{\gamma} = 0$. From this figure, we observe that the minimal score gap exhibit a sharp transition at zero (<1% of the instances have $\underline{\gamma} < 0$), demonstrating that, in most random problem instances with $\|\boldsymbol{p}\| \to \infty$ ($\bar{s} \to 1$), problem directionally

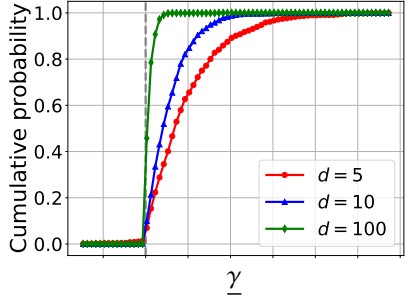

Figure 10: Cumulative prob. of the gap $\underline{\gamma} := \min_{i \in [n], t \in \mathcal{T}_i} \boldsymbol{\gamma}_{i\alpha_i} - \boldsymbol{\gamma}_{it}$ in Figure 9(b).

converges to an LMM i.e. $p(t)/\|p(t)\| \rightarrow p^{mm}/\|p^{mm}\|$. We believe the rare occurrence of a negative score gap is due to small score differences (so that optimal token is not clearly distinguished) and finite number of gradient iterations we run. Interestingly, even in the negative score gap scenarios, gradient descent is aligned with $p^{mm}(\alpha)$ (even if $p^{mm}(\alpha)$ is not LMM) which can be predicted from our Section D which handles the scenario where there are multiple optimal tokens per input.

# F  Addendum to Section 5

We provide an overview of the current literature on implicit regularization and attention mechanism.

## F.1  Related Work on Implicit Regularization

The introduction of Support Vector Machines (SVM), which utilize explicit regularization to choose maximum margin classifiers, represents one of the earliest relevant literature in this field [61]. The concept of maximizing the margin was later connected to generalization performance [62]. From a practical perspective, exponential losses with decaying regularization exhibit asymptotic behavior similar to SVMs, as demonstrated in [22]. While the analysis of the perceptron [63] originally introduced the concept of margins, the method itself does not possess an inherent bias as it terminates with zero classification error. However, establishing a meaningful lower bound for the attained margin is not possible. Initial empirical investigations highlighting the implicit bias of descent methods focused on $\ell_1$-regularization, revealing that coordinate descent, when combined with the exponential loss, exhibits an inherent inclination towards $\ell_1$-regularized solutions [64].

This work draws extensively from the literature on implicit bias and regularization, which has provided valuable techniques and inspiration. A common observation in these studies is the convergence to a specific optimal solution over the training set. This phenomenon has been observed in various approaches, including coordinate descent [65, 66], gradient descent [30, 67, 25, 68, 69, 22, 70], deep linear networks [71, 72], ReLU networks [73, 74, 29, 75, 76], mirror descent [77, 78, 33, 36], and many others. The implicit bias of gradient descent in classification tasks involving separable data has been extensively examined by [22, 25, 26, 27, 28, 29]. The works on classification typically utilize logistic loss or exponentially-tailed losses to establish connections to margin maximization. The results have also been extended to non-separable data by [30, 31, 21]. Additionally, several papers have explored the implicit bias of stochastic gradient descent [37, 38, 41, 42], as well as adaptive and momentum-based methods [43, 44, 45, 46].

While there are some similarities between our optimization approach for $v$ and existing works, the optimization of $p$ presents notable differences. Firstly, our optimization problem is nonconvex and involves a composition of loss and softmax, which introduces new challenges and complexities. The presence of softmax adds a nonlinearity to the problem, requiring specialized techniques for analysis and optimization. Secondly, our analysis introduces the concept of locally-optimal tokens, which refers to tokens that achieve locally optimal solutions in their respective attention cones. This concept is crucial for understanding the behavior of the attention mechanism and its convergence properties. By focusing on the cones surrounding locally-optimal tokens, we provide a tailored analysis that captures the unique characteristics of the attention model. Overall, our work offers novel insights into the optimization of attention-based models and sheds light on the behavior of the attention mechanism during training.

## F.2  Related Work on Attention Mechanism

As the backbone of Transformers [6], the self-attention mechanism [47, 48, 49, 50] plays a crucial role in computing feature representations by globally modeling long-range interactions within the input. Transformers have achieved remarkable empirical success in various domains, including natural language processing [4, 2], recommendation systems [79, 80, 81], and reinforcement learning [82, 83, 84]. With the introduction of Vision Transformer (ViT) [85], Transformer-based models [86, 87] have become a strong alternative to convolutional neural networks (CNN) and become prevalent in vision tasks.

However, the theoretical foundation of Transformers and self-attention mechanisms has remained largely unexplored. Some studies have established important results, including the Lipschitz constant of self-attention [88], properties of the neural tangent kernel [89, 90], and the expressive power and

Turing-completeness of Transformers [91, 92, 93, 51, 23, 94] with statistical guarantees [95, 96]. There is also a growing effort towards a theoretical understanding of emergent abilities of language models – such as in-context learning [97, 98, 99] and chain-of-thought [100, 101, 102] – which are inherently related to the models ability to attend to the relevant information within the input sequence.

Focusing on the self-attention component, Edelman et al. [51] theoretically shows that a single self-attention head can represent a sparse function of the input with a sample complexity for the generalization gap between the training loss and the test loss. However, they did not delve into the algorithmic aspects of training Transformers to achieve desirable loss. Sahiner et al. [52] and Ergen et al. [53] further explored the analysis of convex relaxations for self-attention, investigating potential optimization techniques and properties. The former work applies to self-attention with linear activation (rather than softmax) whereas the latter work attempts to approximate softmax via a linear operation with unit simplex constraints. In contrast, we directly study softmax and characterize its non-convex geometry. In terms of expressive ability, Baldi and Vershynin [54] investigated the capacity of attention layers to capture complex patterns and information, while Dong et al. [23] illustrates the propensity of attention networks to degenerate during the training process, with the result often being an output that is approximately a rank-1 matrix.

Recent works have made progress in characterizing the optimization and generalization dynamics of attention [55, 56, 103, 17, 104]. Jelassi et al. [55] studied gradient-based methods from random initialization and provided a theoretical analysis of the empirical finding that Vision Transformers learn position embeddings that recapitulate the spatial structure of the training data, even though this spatial structure is no longer explicitly represented after the image is split into patches. Li et al. [56] provided theoretical results on training three-layer ViTs for classification tasks. They quantified the importance of self-attention in terms of sample complexity for achieving zero generalization error, as well as the sparsity of attention maps when trained by stochastic gradient descent (SGD). In another related work, Nguyen et al. [104] proposed a primal-dual optimization framework that focuses on deriving attention as the dual expansion of a primal neural network layer. By solving a support vector regression problem, they gained a deeper understanding and explanation of various attention mechanisms. This framework also enables the creation of novel attention mechanisms, offering flexibility and customization in designing attention-based models. In another closely related work, Oymak et al. [17] analyzed the same attention model as ours, denoted by (ERM). Specifically, they jointly optimize $v, p$ for three gradient iterations for a contextual dataset model. This is in contrast to our emphasis on infinite-iteration behavior of $p$-only optimization. However, it is important to note that all of these works make certain assumptions about the data. Specifically, they assume that tokens are tightly clusterable or can be clearly split into relevant and irrelevant sets. Additionally, Li et al. [56] require specific assumptions on the initialization of the model, while Jelassi et al. [55] consider a simplified attention structure where the attention matrix is not directly parameterized with respect to the input.

In contrast, our work offers a comprehensive optimization-theoretic analysis of the attention model, establishing a formal connection to max-margin problems. While comparable works make assumptions on the dataset model, our results apply under minimal assumptions for general data and realistic conditions. Our analysis based on max-margin-equivalence allows us to gain a deeper understanding of the optimization geometry of attention and its behavior during the training process. As articulated in our experiments, our results lead to novel insights even for $n = 1, 2$ samples, $T = 2, 3$ tokens and $d = 2, 3$ dimensions (in contrast to [55, 56, 103, 17]). Notably, our work also presents the first theoretical understanding of the implicit bias exhibited by gradient descent methods in the context of the attention model. We remark that recent work [105] expands the theory presented in this work to 1-layer transformers. By uncovering the underlying optimization principles and thoroughly characterizing the directional convergence of attention, we provide valuable insights into the dynamics and generalization properties of attention-based models opening the path for future research.

