# OpenReview forum: "Max-Margin Token Selection in Attention Mechanism"
_NeurIPS.cc/2023/Conference — NeurIPS 2023 spotlight_

### Official Review · Reviewer_CU4X · 2023-06-29

**Soundness:** 2 fair
**Presentation:** 3 good
**Contribution:** 3 good
**Rating:** 6
**Confidence:** 1

**Summary:**

This paper aims to provide an optimization-theoretic characterization of the softmax attention model $f(X)=v^{\top}X^{\top}{\rm softmax}(XW^{\top}p)$ by linking it to max-margin problems. The authors established the convergence of gradient decent on $p$ for a fixed $v$ choice, and further explored the joint convergence of $(v,p)$ via regularization path analysis. They also showed that the idea on selecting the optimal token via max-margin can be extended to a general nonlinear model. Their results are verified through numerical studies.

**Strengths:**

- Interesting and important result. Attention has played an important role in large language model. However, the theoretical understanding is relatively lack. This paper links the regularized solution and the gradient decent process to the max-margin solution, which may motivate several possible directions for the further research.
- Clear presentation with adequate explanation.


**Weaknesses:**

- Some assumptions are relatively strong. In Assumption B, they assume all non-optimal tokens have equal scores, which may not be true in practice.
- Lack the convergence of the gradient decent when jointly optimizing $(v,p)$.


**Questions:**

- Why Lemma 1 implies line 92-94?

**Limitations:**

See weaknesses.

---

> ### Author Rebuttal · Authors · 2023-08-10
>
> We thank the reviewer for their time and helpful comments.
>
> > **W1:** Some assumptions are relatively strong. In Assumption B, they assume all non-optimal tokens have equal scores, which may not be true in practice.
>
> **R:** Thanks for raising this. We agree that this assumption is fairly strong, however, it is only used within Theorem 1 and nowhere else. Theorem 1 provides a global convergence guarantee for gradient descent and serves as a prelude to the general behavior of the attention’s implicit bias. However, as you can see, our Theorem 3 on the local convergence of gradient descent and all other theorems do not require this assumption. Theorem 3 also clearly establishes that, in general (i.e. when Assumption B does not hold and non-optimal scores are different), the global convergence (to the optimal direction $p^{mm\star}$) can fail due to the existence of locally-optimal directions.
>
> Thus, the only way global convergence (from any initialization) can happen is if $p^{mm\star}$ is the unique locally-optimal direction (per Definition 2). Assumption B is one condition that guarantees this. We believe another condition that guarantees this might be ensuring that “all tokens involved in ATT-SVM are support vectors”. This way, for any choice of non-globally-optimal tokens $\alpha$, there will be some optimal token $opt_i\neq \alpha_i$ that is its SVM-neighbor. Then by Definition 2, $\alpha$ cannot be locally-optimal because $opt_i$ has a higher score than $\alpha_i$. Thus, $p^{mm\star}$ becomes the unique direction satisfying Def 2. While this condition (i.e. all SVM constraints being support vectors) sounds technical, for classical SVM problems, it holds when the embedding dimension $d$ is large [Muthukumar et al. JMLR’21, Hsu et al. AISTATS’21].
>
> Based on this intuition, we provide a new experiment in Figure 4. We solve many random instances of the attention problem for various values of $d$ for 1000 iterations and  investigate the convergence behavior of $p(t)$ generated by gradient descent. Experiments were conducted with various values of $d$ for 1000 iterations. The bar plot in Figure 4 distinguishes between non-local convergence (red bars), local convergence (blue bars), and global convergence (green bars). Global convergence is a strict subset of local convergence. In short, in line with our hypothesis, as $d$ grows, we observe global convergence with probability approaching $1$. While we do not have a proof of this, it certainly makes an interesting discussion for our paper and we hope to incorporate it.
>
>
> > **W2:** Lack the convergence of the gradient descent when jointly optimizing (v,p).
>
> **Response:** While this may appear to be a weakness, we emphasize that the contributions we make are novel and challenging even for $p$-only optimization. For joint optimization, we also provide a regularization path theory that successfully predicts the implicit bias of gradient descent (see Figure 2(b,c)). For joint optimization, there is no remotely similar result in the literature, and we have a surprisingly powerful message (see Sec 3.1): $p$ and $v$ (essentially) converge to their respective max-margin solutions, thus, optimization dynamics of “classification” (v) and “attention” (p) can be decoupled. We genuinely hope that this novel message (and other contributions) will spur interest in the community and invite future research to solve these open problems.
>
> We believe that conducting a separate analysis of attention weights  $p$ can offer a clearer and more comprehensive understanding of the implicit bias ingrained in gradient descent for attention mechanisms. To accomplish this, our approach involves the introduction of concepts like token scores and locally-optimal tokens, each of which demands a more comprehensive and detailed explanation.  Additionally, we undertake an extensive convergence analysis that aims to capture the optimization dynamics through the lens of local SVM geometry and the conic initialization centered around the max-margin solution.
>
> While we acknowledge the potential benefits of a joint implicit bias analysis involving both $(v,p)$, our experiments showcased in Figure 2 of the paper provide evidence for the feasibility of this approach. However, it's important to note that a comprehensive treatment of these intricate technical details might require the incorporation of several novel techniques. Condensing all these details into a single paper could potentially be overwhelming and hinder the clarity of our main findings.
>
>
>
> > **Q1:** Why Lemma 1 implies line 92-94?
>
> **Response:**  Thanks for asking this. Recall that, in our attention model $f(p,W)=v^\top X^\top S(XW^\top p)$ in Line 30 , we can either optimize $W$ or optimize $p$. Throughout the paper, we optimize $p$ because Lemma 1 says that optimizing $W$ can be mapped to optimizing $p$. Lemma 1 creates this mapping as follows: Since we will also use $p$ as a variable, fix a vector $a$ and consider the $W$ optimization with $p\gets a$, $f(W)=v^\top X^\top S(XW^\top a)$, and associated loss function $L(W)$ as defined in Lemma 1.
>
> We map these to the function $f_p(p)=v^\top X^\top S(X p)$ and associated loss function $L_p(p)$ where the idea is viewing the combined $W^\top a$ as a single $p$ variable. We prove that **running gradient descent on $p(t)$ with learning rate $\eta$** is same as **running gradient descent on $W(t)$ with learning rate $\eta||a||^{-2}$** and $W(t)$ iterations precisely track $p(t)$ iterations via $W(t)=||a||^{-2}ap(t)^\top$. Note that, for this to happen, initializations $p(0),W(0)$ should match (and that is all!). In general, for any $W$ initialization, there exists a matching $p$ initialization. Lemma 1 states this for a rank-1 $W$ initialization to avoid verbosity. We are happy to address further questions if anything is unclear.

---

> > ### Comment · Reviewer_CU4X · 2023-08-12
> > **Thanks for the rebuttal**
> >
> > I have read the rebuttal and all the other reviews. To be honest, I am not an expert for this area. After reading the rebuttal, I think this paper has good contributions and the new experiments strengthen the results. Thus I increase my score from 5 to 6.

---

> > > ### Author Response · Authors · 2023-08-12
> > > **Response to Reviewer CU4X**
> > >
> > > We appreciate your time and insights in reviewing the paper. Thank you.

---

### Official Review · Reviewer_iE2T · 2023-07-04

**Soundness:** 2 fair
**Presentation:** 2 fair
**Contribution:** 2 fair
**Rating:** 5
**Confidence:** 3

**Summary:**

The paper is clear and well-written.
Understanding the e optimization dynamics and implicit bias is a significant theoretical issue, especially for morden neural network models.
This paper provides a preliminary theoretical analysis of the margin maximization bias of attention-like models.
Theoretically, the authors provide some global convergence and local convergence results to characterize the implicit bias of gradient descent for attention-like models.

**Strengths:**

The authors present a comprehensive theoretical analysis of the implicit bias associated with margin maximization in attention-like models.
Speficically, they provide theories about global convergence, local convergence, and regularization paths in various training scenarios.
This work extends lots of theoretical results on the implicit bias for linear models to the context of attention-like models.

**Weaknesses:**

In this article, the authors investigate a model that bears resemblance to attention but involves a significant degree of simplification compared to standard attention models.
The model employed combines $W_K$ and $W_Q$ into a single matrix $W$, while substituting one of the $X$'s with $p$.
While attention models are still relatively underexplored in the field of optimization theory, I believe that the level of simplification adopted in this study is overly simplistic and may even be somewhat irrelevant.
With such a simplification by the authors, especially for the optimization about $W$ or $p$, it becomes almost a one-layer neural network optimization problem that is largely irrelevant to attention.

**Questions:**

My primary concern is whether the training dynamics of standard attention models $Atten(X)=W_V X \mathbb{S}(X^\top W_K^\top W_Q X)$ closely resemble the training dynamics of the attention-like model in this study.
Specifically, I find it difficult to discern the training behavior of $W_K$ and $W_Q$, which are crucial components in standard attention, from the results presented by the authors.
If the authors could provide further clarification on this matter, I could change my perspective.

**Limitations:**

As previously mentioned, the main limitation of the article lies in the oversimplification of the attention model, and it remains uncertain whether the training dynamics of the attention model differ fundamentally from the training dynamics of the attention-like model proposed by the authors.

---

> ### Author Rebuttal · Authors · 2023-08-10
>
> We thank the reviewer for their feedback and suggestions. Below, we respond to their concerns point by point. We would be happy to respond to future concerns they may have during the discussion period.
>
> > **W1:* This study is overly simplistic... With such a simplification.. it becomes almost a one-layer neural network problem that is largely irrelevant to attention.
>
> **Response:** We respectfully disagree with this assessment for the following reasons:
>
> 1. Our attention model is practically relevant: In transformers $p$ corresponds to a tunable prompt or [CLS] token [Oymak et al. ICML’23]. When sample size is $n=1$, setting $p=x_1$ (i.e. first token) and optimizing $W$ (via Lemma 1’s equivalence), our theory specializes to establish the implicit bias of a 1-layer self-attention model.
>
> 2. The problem does not become a one-layer neural network optimization. In fact, it is very different for the following reasons:
> Softmax nonlinearity is different from applying $T$ nonlinear activation functions individually because it couples the $T$ nonlinearities to induce a probability distribution. This makes it the standard choice in attention/transformer layers and as a loss function (cross-entropy). Crucially, softmax also induces sparsity which is precisely what happens in real attention maps (e.g. see attached Fig 1) or when attention selects the optimal token within our theory.
> Feedforward neural nets only multiply weights and features. In contrast, our model $f(X)=v^TX^TS(Xp)$ as well as self-attention multiplies features with each other (i.e. the $X$ term appears twice or more).
>
> 3. Finally, we understand the concern that self-attention or transformers may be more practically relevant. On the other hand, we firmly believe our SVM-equivalence framework is fundamental and extensible. To provide a concrete example, we recently discovered that a slight variation of our ATT-SVM seems to predict the implicit bias of self-attention (building on the aforementioned $n=1$ observation). This will be discussed under the next response.
>
> > **Q1:*  My primary concern is whether the training dynamics of standard attention models resemble the training dynamics of the attention-like model in this study. I find it difficult to discern the training behavior of  $W_K$ and $W_Q$, which are crucial components in standard attention. If the authors could provide further clarification on this matter, I could change my perspective.
>
> **Response:** We acknowledge this concern and respond it in two fronts:
> (1) Training behavior of $W_k$ and $W_q$ under our paper’s setting still follows max-margin directions,
> (2) We empirically demonstrate that our Attention<->SVM connection is extensible to self-attention.
>
> **(1) Training behavior of $W_k$ and $W_q$:** Standard self-attention calculates $S(XW_qW_k^\top X^\top)$ where $W_q,W_k$ are size $d\times m$. Clearly, when $W_q,W_k$ are full dimensional ($m=d$), we don’t lose any expressivity by merging them into $W_{prod}=W_qW_k^\top$. On the other hand, we acknowledge that the optimization behavior might be different. Fortunately, for the problem $L(W_k,W_q)=\sum_{i=1}^n v^TX_i^TS(X_iW_qW_k^\top a)$:
> We can prove a version of Lemma 1 that creates a mapping between $W_q,W_k$ iterations and $p$ iterations for any $d\geq m\geq 1$.
> Numerically, we found that $W_{prod}(t)=W_q(t)W_k(t)^\top$ still converges to max-margin direction.
>
> Our experiments are shown in Fig 1. Fig 1(left) is the outcome of our Lemma 1 (iterations on $W_{prod}$ and associated $p$ iterations) whereas Fig 1(right) is the joint $W_k,W_q$ iterations (akin to transformers) and associated $p$ iterations. $W_k,W_q$ still align with the max-margin direction albeit with a slightly different trajectory. To formalize this, we can map the joint gradient updates $W_k(t+1)=W_k(t)-\eta \nabla_{W_k} L(W_k,W_q)$, $W_q(t+1)=W_q(t)-\eta \nabla_{W_q} L(W_k,W_q)$ to the following $p(t)$ iterations on the $L(p)$ objective: Starting with proper $p(0)$ choice and scalar $\nu_0=1$, run
>
> $\nu_{t+1}=\nu_t-\eta\nu_t^{-1}p(t)^\top \nabla L(p(t))$
>
> $p_{t+1}=(\nu_{t+1}/\nu_t)[p(t)-\eta\nu_t^2\nabla L(p(t))]$
>
> This mapping is not as simple as Lemma 1. Regardless, it strongly suggests that our max-margin theory on $p$ should extend to $W_k,W_q$. We emphasize that Lemma 1 is enabled by the vector $a$ being fixed: This way the gradient updates on $W_{prod}$ are rank-1 and stay along $a$ direction. Below, we empirically show that, situation is similar but more intricate for self-attention.
>
>
> **(2) Extensibility** In Fig 3, we study the self-attention objective
>
> $
> \qquad L(W)=\frac{1}{n}\sum_{i=1}^n \ell(Y_i\cdot v^\top X^\top S(X_i W^\top x_{i1}))  \qquad  $ (SA-ERM)
>
> This corresponds to running linear classification on the first token output of a self-attention layer ($x_{i1}$). We consider a slightly modified ATT-SVM to capture the inductive bias of this
>
> $
> \qquad \min_{W} || W||_F \quad  \text{subject to} \quad (x_{i\alpha_i}-x_{it})^\top W x_{i1}\geq 1 \quad  \text{for all }\quad t\neq \alpha_i, i\in[n]   \qquad
> $  (S-ATT-SVM)
>
> The intuition is as follows: In our ATT-SVM (with Lemma 1), $a$ is fixed whereas here $a\gets x_{i1}$ is changing for each training example. Fig 3 shows **self-attention solutions directionally align** with (S-ATT-SVM). Empirically, we found that directly optimizing $W_{prod}$ biases the gradient descent towards (S-ATT-SVM) with the Frobenius norm objective, while optimizing $(W_k, W_q)\in\\mathbb{R}^{d\times d}$ separately biases it towards (S-ATT-SVM) with the nuclear norm objective. In our paper's setting (Lemma 1), both objectives coincide because the solution is rank-1 due to fixed $a$. Finally, if $(W_k,W_q)$ are $d\times m$ with $m<d$, we suspect a low-rank constraint in (S-ATT-SVM) is needed.
>
> To sum up, we agree with the reviewer that $(W_k,W_q)$ or self-attention introduce unique behavior, however, based on empirical evidence, the Attention<->SVM connection introduced by our paper remains valid.

---

> > ### Comment · Reviewer_iE2T · 2023-08-11
> >
> > Thanks for the reviewer's response. Now I acknowledge the insights provided by this work. However, I still have the following concerns:
> >
> > 1. I am still confused about the model. In practice, it is common to use $Atten(X)$ based on the dot-product $\left<W_KX, W_Q X\right>$. The model in this work is a bit strange, and it is unclear why the authors changed the $W_K^\top W_QX$ in it to $W p$. At least, the authors should suggest some motivation for changing $Atten(X)$ in this way
> >
> > 2. In practice, it is hard to train Attention-based Transformer models, i.e., the loss is difficult to converge to $0$. However, the margin-maximization implicit bias is usually observed at the terminal stage of training. Can this implicit bias be observed in the practical setting of training Transformer?
> >
> > 3. Most methods and techniques in this work are extended from the linear model (Ji and Telgrasky), such as the regularized path. Could the authors briefly clarify the difference in proof technique?

---

> > > ### Author Response · Authors · 2023-08-12
> > > **Response to Questions  1 and 2**
> > >
> > > > **Q1-1:**  I am still confused about the model. In practice, it is common to use  $Atten(X)$ based on the dot-product  $<W_KX, W_QX>$. The model in this work is a bit strange, and it is unclear why the authors changed the   in it to $W_K^\top W_QX$ to $Wp$.
> > >
> > > **Response:** As mentioned in our initial response, the main claim that
> > >
> > >     softmax-attention weights, trained with gradient descent, converge to a max-margin solution that effectively separates locally-optimal tokens from non-optimal ones
> > >
> > > unsurprisingly **remains valid when optimization problem is formulated using  $ W_KW_Q^\top$ decomposition** and the gradient descent updates $ W_K,W_Q$ seperatly. Our new experiments (Figure 1 in the attached file) and subsequent analyses, which establish a correlation between $p$ and the matrices $W_K$ and $W_Q$, further shows that their combined product matrix $W_{\text{prod}} = W_KW_Q^\top$ asymptotically approaches the solution seen in our SVM.
> > >
> > > Note that $W_{\text{prod}}$ is what matters for the eventual model; this is why we directly used it in our exposition. However, based on the reviewer's concern
> > >
> > > - We will start our exposition with $(W_K, W_Q)$ and then combine them into $W_{\text{prod}}$.
> > > - We will also discuss the mapping we constructed for $W_K, W_Q$ and explain that their $W_{\text{prod}}$ is also similarly predictable and goes to the SVM solution empirically.
> > >
> > >
> > >
> > >
> > > > **Q1-2:** At least, the authors should suggest some motivation for changing $Atten(X)$   in this way.
> > >
> > > **Response:**  Our attention model is $f(X)=\left<Xv,\mathbb{S}(X W^\top p)\right>$. There are three motivations for this:
> > >
> > > **I. Prompt-Tuning and '[CLS]' Token**: In practice, the model arises from prompt-tuning [Lester et al. EMNLP’21] or the '[CLS]' token [Devlin et al. NAACL’19]. To see this, following [Oymak et al. ICML'23] (see their Sec 2.1), let $p$ be a tunable prompt (attached to input $X$) and consider the cross-attention between prompt-attached input $[X;p]$ and $[X]$. Within this cross-attention layer, this results in the output
> > >
> > >   $\qquad \text{Attn}([X;p],X)=\mathbb{S}([X;p]WX^\top)XV=[\mathbb{S}(XWX^\top)XV; \mathbb{S}(p^TWX^\top)XV]$
> > >
> > >  $\qquad$ Thus, the output associated with the tunable prompt has the form $V^\top X^\top \mathbb{S}(XW^\top p)$ which is exactly our model.
> > >
> > > **II. Unveiling the Self-Attention Mechanism:**  The model recovers self-attention when the sample size is $n=1$. Observe that *first token output* of the self-attention is given by $\text{Attn}(X)=V^\top X^\top \mathbb{S}(X W^\top x_1)$. Setting $a\gets x_1$ in Lemma 1, our results recover this self-attention setting. That is also how we recently discovered implicit bias of self-attention admits an SVM for $n>1$.  Please see initial response and the connection between SA-ERM and S-ATT-SVM.
> > >
> > >
> > >  **III. Connection to Matrix Factorization:** The model is fundamental in nature. If you remove the softmax, it becomes a rank-1 matrix learning model where the goal is learning $(v,p)$ from labels of the form $y=v^\top X^\top Xp=\left<X^\top X,pv^\top \right>$. There is a vast literature on this. We believe extending such a matrix factorization viewpoint to softmax nonlinearity is mathematically fundamental and interesting.
> > >
> > >
> > > > **Q2.** In practice, it is hard to train Attention-based Transformer models, i.e., the loss is difficult to converge to 0. However, the margin-maximization implicit bias is usually observed at the terminal stage of training. Can this implicit bias be observed in the practical setting of training Transformer?
> > >
> > > **Response:**  The loss in Theorems 1-4 **does not converge to zero**. This is because we keep $v$ fixed and only train $p$. Since $v$ remains fixed and attention outputs a convex combination of tokens, the model cannot drive the output to $\infty$ and reduce the loss to zero. We believe that this distinction also represents a **significant difference** from margin maximization in logistic regression applied to separable data [Soudry et al. JMLR’18, Ji and Telgarsky ICLR 2019]. We offer empirical evidence that self-attention's implicit bias can be predicted through a minor adjustment of our ATT-SVM. Additionally, in Section 4, we elaborate on how these theoretical insights extend to nonlinear prediction heads, such as MLP.
> > >
> > > Given these considerations, we maintain that our findings offer valuable insights for transformers. At the very least, our experiments with real data (refer to attached Fig 2) demonstrate that the empirical phenomena in the optimization dynamics of transformers align with our theory. Specifically, softmax/attention maps become sparser over time, while the norm of the attention weights continues to increase over the same period. This observation essentially aligns with our theoretical proposition: softmax saturates on optimal tokens, and the weights tend toward infinity.

---

> > > > ### Author Response · Authors · 2023-08-12
> > > > **Response to Question 3**
> > > >
> > > > > **Q3:** Most methods and techniques in this work are extended from the linear model (Ji and Telgrasky), such as the regularized path. Could the authors briefly clarify the difference in proof technique?
> > > >
> > > > **Response:** The connection between exponentially-tailed functions and margin maximization represents the shared intuition between our work and previous literature on logistic regression, including [Ji et al. COLT20]. Notably, the regularization path arguments have existed for over 20 years [Rosset et al. NeurIPS03] and have motivated our study. However, we introduce novel methods and techniques, elaborated upon below, that extend beyond the scope of the linear models.
> > > >
> > > >
> > > >  **1. [Ji et al. COLT20] utilize convex optimization analysis or not**. Linear models with a convex loss function inherently exhibit convexity, leading gradient descent to converge toward a single global direction. Theorem 1 in [Ji et al. COLT20] demonstrates that when the loss is convex and decreasing to $0$, the regularization path predicts the trajectory of gradient descent, with the authors effectively utilizing convex analysis.
> > > > However, even when considering the simplest scenario we can formulate, our problem remains non-convex. To illustrate this concretely, let's set:
> > > >
> > > > - $\ell(x)=-x$
> > > > - Choose $n=1$ sample and $T=2$ tokens. Ensure that the tokens have unit $\ell_2$ norm and are orthogonal.
> > > > - Set $v=x_1$, $p=cx_1$ with $c\in \mathbb{R}$, resulting in $x_2^\top v=x_2^\top p=0$.
> > > >
> > > > Following this setup, the training objective takes the shape:
> > > >
> > > > $\max_p L(p):= \mathbb{S}([c, 0])_1 = \frac{e^c}{1+e^c}.$
> > > >
> > > > This function corresponds to the standard logistic function and is non-convex. Consequently, the framework presented in [Ji et al. COLT20] substantially differs from our context, where gradient descent can provably converge to distinct locally-optimal directions. These directions (as outlined in Definition 2) possess a nontrivial characterization, yet we effectively describe them and establish the guaranteed convergence of gradient descent.
> > > >
> > > >
> > > >
> > > > **2. Locally Optimal Solutions, Conic Initialization, and Gradient Correlation.** As discussed in response to Reviewer 8XUp, our proof deviates from relying on convexity and instead centers on constructing a cone around $p^{mm}$.  Let's refer to this cone as  $\mathcal{C}_{\mu}(p^{mm})$ for simplicity, as defined in Eq. (5) on page 5. The framework of this constructed cone serves as the cornerstone of our analysis, unfolding as follows:
> > > >
> > > > * **S1**: Our initial focus resides in demonstrating the absence of finite stationary points within $\mathcal{C}_{\mu}(p^{mm})$ unequivocally emphasizing that the single locally optimal solution within the cone is $p^{mm}$.
> > > >
> > > > * **S2**: Within this established cone, we leverage the advantageous gradient correlation, provided in Lemma 5. For example, in Lemma 5, we establish that for any choice of $\pi$, there exists $R_{\pi}$ such that: $ \langle \nabla \cal{L}(p), \frac{p}{|| p||}\rangle \geq (1+\pi) \langle \nabla \mathcal{L}(p), \frac{p^{mm}}{|| p^{mm}||} \rangle.$
> > > >
> > > > * **S3**: It is rigorously proven that gradient descent remains confined within the cone $\mathcal{C}_{\mu}(p^{mm})$ when initiated within it.
> > > >
> > > > * **S4**: By leveraging the well-established descent properties of gradient descent, we illustrate the directional convergence towards $p^{mm}$.
> > > >
> > > > Our novel cone construction and **S1-S3** steps distinguish themselves from the linear models literature. While
> > > >  convexity simplifies gradient correlation in **S2**, presenting this result in a nonconvex setting is both novel and intriguing. Conversely, **S4** aligns with standard gradient descent analysis, paralleling the intuition that  $p(t) \rightarrow \infty$, ultimately converging to the maximum margin solution.
> > > >
> > > >
> > > > **3. ATT-SVM: A Novel Token-Separating SVM Formulation.**  It's important to highlight that our ATT-SVM significantly differs from existing SVM formulations in the linear models' literature. While the logistic<->SVM connection typically revolves around standard classification, our ATT-SVM focuses on token separation within input sequences. Furthermore, our regularization path results and findings also bring novelty. Despite the comparative simplicity of analyzing regularization paths over gradient descent, the theories and findings in both Sections 3 and 4 are far from trivial. A prime example of this complexity lies in the nuanced differentiation between Sections 3.1 and 3.2. This intricate understanding demands a delicate characterization of problem geometry concerning support vectors and their contributions to the ultimate logistic loss. For in-depth insights, refer to the proofs of Theorems 5 and 6 in the appendix.
> > > >
> > > >
> > > >
> > > > In summary, our nonconvex formulation, conic-based analysis of gradient descent, and attention SVM formulation significantly distinguish themselves from existing literature on linear models, including [Ji et al. COLT20].

---

> > > > > ### Comment · Reviewer_iE2T · 2023-08-18
> > > > >
> > > > > Thank you for the detailed responses. Technically, this work is solid. It introduces new techniques for analyzing the maximum margin implicit bias in nonlinear models and offers new insights into attention's dynamics. My current recommendation is split between 4 and 5. I still have some reservations over the "attention-like" model in this work. In practice, I have rarely seen such an "attention-like" model. Nevertheless, this work makes a technically sound contribution, and the theoretical insights from the "attention-like" model align well with experimental results for attention. I raise my recommendation to 5 and encourage the authors to clarify the motivations for studying this "attention-like" model in the final version.

---

> > > > > > ### Author Response · Authors · 2023-08-18
> > > > > > **Response to Reviewer iE2T**
> > > > > >
> > > > > > We appreciate the time and effort you invested in reviewing our paper.  Your valuable suggestions will be incorporated into the final revision.

---

### Official Review · Reviewer_8XUp · 2023-07-07

**Soundness:** 3 good
**Presentation:** 2 fair
**Contribution:** 3 good
**Rating:** 7
**Confidence:** 3

**Summary:**

The paper focuses on the attention mechanism which is commonly used in transformer architectures. In particular, the authors introduce a certain attention model and investigate its optimization dynamics and inductive biases under various assumptions on token's scores. In particular, the setting is a single-head attention mechanism trained by gradient descent and with decreasing losses, such as logistic or linear loss for binary classification. The results mainly hold for attention with linear head and fixed classifier head however some of the results hold without these limitations. The first main contribution of the paper is proving (under the assumption that all non-optimal tokens have the same score value) directional convergence of tunable prompt (denoted by $p$) to a certain max-margin solution which separate one token from the rest of tokens for each input. However, the imposed assumption on score values can be limiting; thus the authors assert that proving local convergence is possible if the initialization is within a cone of the final solution. The paper also studies some extensions such as the joint optimization of the classifier head and trainable parameters and demonstrate that under a specific label margin conditions, the classifier head and trainable parameters converge to their respective max-margin solutions. Some numerical results on synthetic data are designed to validate the claims of the paper.
















































































































**Strengths:**

The paper is the first work on the implicit bias behavior of GD for the attention mechanism. Understanding implicit bias is crucial for several directions such as fairness, optimization behavior and generalization bounds. Some assumptions of this work are idealized, but that is fine given that it one of the first solid works in this direction. The most important aspect of the paper is giving a formalized understanding of the attention mechanism as a token-selection mechanism and providing sufficient conditions for convergence to a solution which favors optimal tokens. By connecting the attention mechanism to the implicit bias literature and the max-margin SVM formulation (which for the attention mechanism takes a new and interesting form), the study establishes a solid foundation for future research. The required work for obtaining the results is non-trivial and sufficiently distinct from previous works in implicit-bias literature. The paper is also well-written in most parts, although some parts of the paper become very technical with little intuition and thus it can be difficult to clearly understand the results such as the results in section 2.3.

**Weaknesses:**

some questions and minor weaknesses and suggestions:

- Regarding theorem 1, the fact that any initialization leads to convergence is rather counter-intuitive. Does the assumptions of Theorem 1 imply that the problem convex or is there any other reason?

- Also related to theorem 1, it is not trivial why parameters norm ($|| p_t ||$) is diverging since $p_t$ is inside the soft-max. can the authors please explicitly specify the behavior of $|| p_t ||$ in the statement of theorem? We know for GD on ReLU neural networks that parameters norm diverges, but that is due to using ReLU non-linearities which implies that the loss prefers large first-layer weights.

- The contributions section can be more specific by providing more details for contributions; for example in line 52 the authors can be more specific in explaining their contributions for the model with non-linear head and specifing the key distinctions with previous parts.

- How are the values of $\mu$ and $R$ determined in theorem 3? This can be insightful as they are the key parameters in specifying the cone.

**Questions:**

please see the section above.

**Limitations:**

The authors adequately discuss the limitations throughout the paper. I do not see any potential negative impact with this work.

---

> ### Author Rebuttal · Authors · 2023-08-10
>
>
> We thank the reviewer for their thorough feedback and helpful suggestions.
>
> >**W1:**  Is the problem in theorem 1 convex, any other reason?
>
> **R:** Thank you for the great question. First, let us clarify that the problem is not convex even under Assumption B. One reason is that Assumption A actually allows for very general nonconvex loss functions. However, even for the simplest setting (we can come up with), the problem is not convex: Concretely, let us set
>
> - $\ell(x)=-x$
> - Pick n=1 samples and $T=2$ tokens. Make tokens unit $\ell_2$ norm and orthogonal.
> - Set $v=x_1$, $p=cx_1$ with $c\in \mathbb{R}$. This way $x_2^\top v=x_2^\top p=0$.
>
> Following this, the training objective takes the form $max_p L(p):= \mathbb{S}([c, 0])_1 = \frac{e^c}{1+e^c}$. This is the standard logistic function and is not convex or concave. We will add this example to the paper.
>
> Rather than convexity, our proof relies on establishing favorable **gradient correlation** presented in Lemmas 3 and 5. For instance, in Lemma 5, we establish that for any choice of $\pi$, there exists $R_{\pi}$ such that:
>
> $ \langle \nabla \cal{L}(p),  \frac{p}{|| p||}\rangle \geq (1+\pi) \langle \nabla \mathcal{L}(p), \frac{p^{mm}}{|| p^{mm}||} \rangle.$
>
> If $\mathcal{L}(p)$ was convex, it would actually help with establishing the above. Instead, our analysis could be perceived as a directional convergence version of the restricted secant inequality [Karimi et al. ECML’16] where the gradient behaves nicely towards the $p^{mm}$ direction. Via gradient correlation, we are also establishing the **weak convexity** of $\mathcal{L}$: As $p$ converges in direction to $p^{mm}$, it can be demonstrated from Lemma 4 that $\nabla \mathcal{L}((1+\pi) ||p||p^{mm}) \rightarrow 0$, which implies the weak monotonicity of the gradient and the weak convexity of $\mathcal{L}$.
>
> > **W2:** It is not trivial why parameters norm ($p(t)$) is diverging since $p(t)$ is inside softmax. Can the authors please explicitly specify the behavior of $||p(t)||$?
>
> **R:** We will provide a better discussion in the final manuscript. The intuition is as follows: Softmax output is a probability vector, thus the attention output $f(X)=\left<Xv, \mathbb{S}(Xp)\right>$ creates a convex combination of token scores $\gamma=Xv\in \mathbb{R}^T$ (here we simply set label $Y=1$). When $v$  is fixed, $\gamma$ is fixed and, since the loss function is decreasing, the smallest training loss the attention model can achieve is by **assigning all softmax probability to the tokens with the highest score**. Otherwise we are strictly worse off when the convex combination contains some non-optimal tokens. Thus, we want probability 1 for optimal tokens and 0 for others. On the other hand, softmax with finite weights cannot accomplish this because softmax output is strictly positive. This is precisely why norms go to $\infty$: Softmax asymptotically sets token probabilities to $1$ and $0$.
>
> To provide a GD-specific intuition, Lemma 4 in the supplementary shows: $\langle \nabla \mathcal{L}(p), p^{mm} \rangle < 0$ for all finite $p \in \mathbb{R}^d$. Consequently, there are no finite critical points $p$ for which $\nabla \mathcal{L}(p) = \mathbf{0}$. This implies that $|| p\left(t\right) ||\rightarrow \infty$. In our Thms 1 and 3, we prove that $||p(t)||$ diverges and aligns with the SVM solution, and attention maps $\mathbb{S}(Xp)$ select optimal tokens as $t$ grows. However, the precise quantification of behavior of $||p(t)||$ remains open.
>
>
> > **W3:** Contributions section can be more specific, specifically for nonlinear head
>
> **R:** Thank you for the suggestion. We will incorporate the following in the main text:
>
> To establish the margin maximizing nature of attention under broader conditions, we study the general model $f(\mathbf{X})=\psi(\mathbf{X}^\top \mathbb{S}(\mathbf{X}\mathbf{W}^\top p))$ where $\psi$ is a nonlinear head. This setting poses challenges as we lack a clear score function, unlike the previous sections. To address this, we introduce a generic condition that splits the tokens of each input into an optimal set and a non-optimal set. Non-optimal tokens are those that strictly increase the training risk if they are not fully suppressed by attention probabilities $\mathbb{S}(X_iW^\top p)$.
>
>
> > **W4:** How are the $\mu$ and $R$ determined in Thm 3?
>
> **R:**  As stated in Step 1 of the proof sketch on page 6, we $\mu$ is a function of the margin of the entire dataset. Specifically:
> $\delta := \frac{1}{2}\min_{i\in[n]}\min_{t\in \cal{T}i,\tau\in \bar{\cal{T}}i}(\mathbf{k}{it}-\mathbf{k}{i\tau})^\top p^{mm}, \quad A := \max_{i\in[n],t\in[T]} ||\mathbf{k}_{it}|| \cdot  || p^{mm}||, \quad  \mu = \frac{1}{8}\left(\frac{\min(0.5,\delta)}{A}\right)^2.$
>
> Furthermore, Lemma 3 reveals that $R$ is inversely dependent on both $\mu$ and the score gap:
>
> $ R \geq \frac{\max(2,\delta^{-1})}{||p^{mm} ||}\log\left(\frac{64T\Gamma A}{\boldsymbol{\gamma}}\right),$
>
> where $\boldsymbol{\gamma} = \min_{i\in[n]}\boldsymbol{\gamma}_i$ represents the worst-case score gap across all inputs.
>
> The definition of $\mu$ offers valuable insights into the gradient descent initialization process. When tokens are more separable, $\mu$ increases, leading to a reduction in correlation ($1-\mu$) and an expansion of the initialization cone. Similarly, $R$ exhibits an inverse relationship with both $\mu$ and the score gap $\boldsymbol{\gamma}$. When the data's score gap and $\mu$ are large, gradient descent can commence with a small norm, allowing for initialization within a wider cone around $p^{mm}$.
>
> We will integrate the aforementioned insights into the discussion of Theorem 3, emphasizing the significant impact of considering the interplay between $\mu$, the score gap, and the properties of tokens on the gradient descent initialization process.

---

> > ### Comment · Reviewer_8XUp · 2023-08-18
> >
> > Thank you for the detailed response. I will maintain my score.

---

> > > ### Author Response · Authors · 2023-08-18
> > > **Response to Reviewer 8XUp**
> > >
> > > Thank you for your time and effort in reviewing our paper.

---

### Official Review · Reviewer_5i57 · 2023-07-07

**Soundness:** 3 good
**Presentation:** 3 good
**Contribution:** 4 excellent
**Rating:** 8
**Confidence:** 3

**Summary:**

This work studies the mechanism for relevant token selection in the attention model by drawing connections with the implicit bias literature and max-margin SVM formulation. The authors consider the prompt attention model $f(X)=v^TX^T\text{softmax}(XW^Tp)$, with tokenized input $X$, value weights $v$, key-query weights $W$, and tunable token/prompt $p$. They show that running gradient descent (GD) on $p$ and $W$ (for fixed $W$ and $p$, respectively) is equivalent, so they consider optimizing only $p$ for fixed $W$.

They consider a data setting where the quality of $t^{th}$ token of input $X$ is determined by its scores $Yv^Tx_t$, with the globally optimal tokens being the ones with the highest scores.

First, they consider optimizing $p$ when $v$ is fixed using a loss that is decreasing and smooth. In this setting, they show the following results:

- Under an assumption on the token scores, $p$ converges (in direction) to the global max-margin solution, which separates the globally optimal tokens from the rest.
- They also develop a regularization path analysis to show global convergence by relaxing the said assumption.
- The main result shows that with an appropriate initialization and a small enough step size, the GD iterates of $p$ converge (in direction) to a local max-margin solution that separates the locally-optimal tokens from the rest. Here, locally optimal tokens are defined as the ones which have higher scores than their SVM neighbors.

Next, they consider the joint optimization of $p$ and $v$ using logistic loss. In this case, for a given $p$, if the resulting features are separable, $v$ has an implicit bias to converge to the max-margin solution as the problem is linear in $v$. Here, optimal tokens as the ones that maximize the downstream label margin. They show that:

- When the attention features (for the max-margin $p$) are all support vectors (for the respective max-margin $v$), both $v$ and $p$ converge to their respective max-margin solutions.
- When this is not the case (i.e. the attention features are not all support vectors), $p$ asymptotically selects one token per input, and it suffices to select tokens with the highest scores while also mixing other tokens. This does not impact the margin of $v$, which still converges to the max-margin solution.

Throughout, the authors illustrate these results through numerical experiments.

**Strengths:**

1. This work gives interesting theoretical insights into the mechanism for relevant token selection in the attention model.

2. It lays the groundwork to analyze attention mechanism using the lens of implicit bias and motivates several interesting directions for future work.

3. Overall, it is an interesting paper, with clear exposition and well-connected ideas. It makes several meaningful contributions (as listed in the summary).

**Weaknesses:**

1. The experimental results seem limited and the paper would benefit from the inclusion of more experiments.

2. Some points of discussion can enhance intuition, and certain aspects regarding figures need some clarification.

**Questions:**

1. Experiments:
- The numerical experiments illustrate the theoretical results well. However, it would be good to include additional experiments on semi-synthetic or real datasets, such as those considered in [1], [2].
- Fig. 1(b) illustrates the local convergence of $p$ in the red or blue direction, depending on the initialization. It would be helpful to see if there are cases when the GD iterates do not converge to either of the two solutions shown in Fig. 1(b). This would give some insight into the gap between the empirical observations and the theoretical result for this case.

2. Discussion/clarification:
- Points of discussion:
    - Generally, margin maximization refers to separating samples/features from two classes, whereas in this work, the main contribution is to show that the tunable prompt converges to a solution that separates the globally/locally optimal tokens from non-optimal ones. I suggest including more discussion on this part in the introduction. Relatedly, it would help to clarify lines 36-37.
    - In Section 2, choice of $v$ determines the scores, and hence the solution learned by $p$. Some discussion on this would be nice.
    - The data setting is interesting, but since there are a lot of cases, it would be helpful to show some connections between (some of) the synthetic data setting(s) and some datasets that are commonly used in practice.
    - In the discussion on convergence to the local max-margin soution, and the description of Fig. 1(b) in Section 2.2, it would be helpful to clarify that depending on the initialization, the GD iterates will converge to either the global max-margin (when initialized in the cone associated with that solution) or the local max-margin.

- Figures:
    - In Fig. 1(a), it is unclear what the role of the blue line (local max-margin solution) is, since both the non-opt tokens take the same value.
    - Fig. 1(c) description needs some clarification. The separating hyperplane in. the figure looks like it's the max-margin solution for the teal datapoint, but not the max-margin across all three colors (margin with green is small).
    - In Figs. 2(b) and (c), it would be helpful to have the legend for red and blue curves associated with $p$ and $v$, respectively.

- Other:
    - There are some typos/inconsistencies in the proof of Lemma 1 that should be corrected.
    - Theorem 2 shows global convergence of $p$ via regularization path analysis. It can be moved above assumption B to improve flow, as it is more general.
    - In Section 2.2, it would be helpful to use some specific notation for the solution that separates the locally optimal tokens from the rest ($p^{mm}(\alpha)$), such as $p^{mm}_l$.
    - Fig. 2(a) comparing the transient dynamics for correlation and logistic loss is interesting. However, it needs a minor clarification in the description. It is stated that when $p$ selects the optimal token, the gradient norm $\propto \gamma_i$ for correlation loss, and $\propto \gamma_ie^{-\gamma_i}$ for logistic loss, and we can compare loss for tokens with different scores. However, if $p$ selects the optimal token, the score would be fixed. Maybe, it can be rephrased to “if $p$ selects the token with score $\gamma_i$”.
    - In Section 3.1, label margin is defined as $\frac{1}{||v_{mm}||}$, but in the example, label margin $\gamma$ is defined as $||v_*||$. It would be helpful to mention what $v_*$ is for clarity.

References:

[1] Samet Oymak, Ankit Singh Rawat, Mahdi Soltanolkotabi, and Christos Thrampoulidis. On the role of attention in prompt-tuning. In ICLR 2023 Workshop on Mathematical and Empirical Understanding of Foundation Models, 2023.

[2] Hongkang Li, Meng Wang, Sijia Liu, and Pin-Yu Chen. A theoretical understanding of shallow vision transformers: Learning, generalization, and sample complexity. arXiv preprint arXiv:2302.06015, 2023.

---

> ### Author Rebuttal · Authors · 2023-08-10
>
> We thank the reviewer for their thorough review and helpful suggestions, they will definitely help improve the paper.
>
> > **Q1:** The numerical experiments such as those considered in [1], [2].
>
> **R:** Following your suggestion, we conducted additional experiments using real-world datasets to further substantiate our hypothesis concerning the optimization dynamics in transformers and attention. Our theory successfully anticipates two crucial empirical phenomena:
>
>    - The attention map (i.e. softmax output) becomes more sparse over time by focusing on the most informative tokens.
>    - This is achieved by the norm of the attention weights ($W$) growing over time and leading to a "saturating" effect on softmax, resulting in a sparse pattern.
>
> Our experiments in Figure 2 verify both of these predictions. We train a vision transformer (ViT-base) model from scratch with the CIFAR-10 dataset for 400 epochs with fixed learning rate $3\times 10^{-3}$.
>
>    - In Figure 2 (left), we present the progressive change in attention weights of the [CLS] token (which corresponds to our $p$ parameter) during training, computed from all attention heads within the model.
>   - In Figure 2 (right), we display the norm of attention weights and the sparsity level of attention maps averaged over all layers. We used $($L1norm/L2norm$)^2$ of the attention maps as a soft-sparsity measure, where a smaller value indicates a sparser vector.
>
> Initially, during the early epochs of training, the attention weights are randomly distributed, leading to a dense pattern. However, as training progresses, the weights grow, causing the attention map to gradually become sparser in line with our Thm 1-3.
>
> > **Q2:** Scenarios where GD doesn’t converge to locally-optimal directions
>
> **R:**  Thank you for the comment.  We've examined the gradient descent-generated behavior of $p(t)$ in Figure 4 (attached) with random initialization. We ran experiments with varying $d$ values for 1000 iterations.  We find that, **(1, red bar)** for small $d\in[2,5]$,  $p(t)$ does not have to saturate softmax i.e. may not converge to a local ($p^{mm}$) or global ($p^{mm\star}$) max-margin direction. We suspect this is because the **ATT-SVM is not feasible for small $d$**. **(2, blue bar)** For larger $d$’s,  $p(t)$  indeed converges to a locally-optimal direction. **(3, green bar)** As $d$ gets even larger, $p(t)$  converges more frequently to the globally-optimal direction; please also see table below.
>
> | d   | Finite p(t) | % of $p(t) \rightarrow p^{mm}$ | % of   $p(t) \rightarrow p^{mm \star}$ |
> |-----|--------|-------|--------|
> | 2   | 69.1   | 30    | 10.4   |
> | 5   | 5.9    | 92.8  | 17.9   |
> | 10  | 0      | 99.7  | 38.3   |
> | 100 | 0      | 99.5  | 92     |
> | 300 | 0      | 100   | 99.1   |
> | 500 | 0      | 100   | 99.8   |
>
> > **Q3:** Clarify the main contribution (nature of ATT-SVM) and lines 36-37
>
> **R:** Thank you for the great suggestion. We will incorporate the following discussion in the main text:
>
> Gradient descent on logistic loss and separable datasets converges to the hard margin SVM solution for linear classification   [Soudry et al. JMLR’18, Rosset et al. NeurIPS’03, Telgarsky ICML’13]. Similarly, the attention layer in neural networks, utilizing softmax nonlinearity, exhibits behavior resembling margin-maximizing solutions. However, attention mechanism operates on input tokens rather than performing direct classification. Thus, it aims to separate tokens within input sequences, favoring SVM-like solutions, represented by (ATT-SVM). Formalizing this intuition is challenging due to the highly nonconvex nature of the optimization landscape caused by the softmax operation.
>
> > **Q4:** Discussion on choice of $v$ and scores
>
> **R:** Agreed, we will elaborate on the score definition and how the choice of $v$ impacts the solution learned by $p$.
>
>
>
> > **Q5:** The data setting is  … used in practice.
>
> **R:** Please refer to our response to your **Q1**. We also emphasize that Def 2 and Thm 3 notably apply to general datasets. We only need ATT-SVM to be feasible.
>
> > **Q6:** In the discussion on convergence,  clarify dependence of GD on initialization
>
> **R:** Thank you for the suggestion. We will highlight that convergence of attention weights depends on the point of initialization and GD can potentially converge to any of the locally-optimal directions per Def 2.
>
> > **Q7:** In Fig. 1(a), the role of the blue line is unclear
>
> **R:** Agreed with the reviewer. In Fig 1(a), the blue line is unnecessary and will be removed. We will also clarify Fig 1(c) by providing additional explanatory notes. In Fig 2(b) and (c), we will include a legend to denote the red curve associated with $p$ and the blue curve associated with $v$, providing clear identification for each curve.
>
> > **Q8:** typos/inconsistencies in the proof of Lemma 1 that should be corrected.
>
> **R:** Thanks for catching this. We also noticed it after submission. We replaced all $q$ terms with $p$ and we are now consistently using $\cal{L}_p$ and $\cal{L}_W$ to distinguish the optimization objectives of $p$ and $W$.
>
> > **Q9:** Theorem 2 is more general.
>
> **R:** Totally agreed. We'll include your suggestion in the final paper version.
>
> > **Q10:**  In Section 2.2, it would be helpful … such as $p^{mm}_l$.
>
> **R:**  Thanks, good point. We may include this in the final paper.
>
> > **Q11:**  Clarify Fig. 2(a) … “if  $p$ selects the token with score $\gamma_i$.
>
> **R:**  We will clarify that $n=2$ and there are two optimal tokens with scores $\gamma_1=1,\gamma_2=C$ (non-opt score is 0). Also the sentence “$p$ *approximately* selects optimal tokens” refers to softmax output assigning high-probability to the optimal tokens. This means $\approx 1$ probability (which indeed occurs as we run GD longer) but not necessarily $=1$.
>
> > **Q12:**  Clarify $v_*$ and its margin.
>
> **R:**  Thank you, we will clarify that $v^{mm}=v^*/||v^*||^2$ resulting in $1/||v^{mm}||=||v^*||$.

---

> > ### Comment · Reviewer_5i57 · 2023-08-11
> >
> > Thank you for the detailed responses. All concerns have been addressed and the experimental results shared by the authors further strengthen the paper. Hence, I have increased the score to 8.

---

> > > ### Author Response · Authors · 2023-08-11
> > > **Response to Reviewer 5i57**
> > >
> > > We thank you for your thorough review and valuable suggestions that significantly improved the quality of our work.

---

### Official Review · Reviewer_SE93 · 2023-07-26

**Soundness:** 4 excellent
**Presentation:** 3 good
**Contribution:** 3 good
**Rating:** 6
**Confidence:** 2

**Summary:**

This paper proposes to give a mathematical explanation and analysis for widely used attention mechanism. They formulate normal attention, self-attention and prompt tuning into one single formulation. And they connected attention to max-margin problems.

**Strengths:**

1. This paper proposes a mathematical analysis for attention mechanims, which deepen our understanding of the operator.
2. The authors formulate several kinds of attentions and prompt-tuning into one single formulate, which is practical and novel.

**Weaknesses:**

1. To be honest, I am not an expert for this area. I appreciate the author's effort on the mathmatical part for deep learning.
    However, I think it would be better to give conclusions and design guidances based on the authors' observations.
    For example, can we link and explain some phenomena or pains during attention-based model training? Can we improve or accelerate training by improving network structure or losses?

2. The authors discuss attention modules based on isolated simple operators. I think it is very helpful.
    However, I wonder if we extend to real large-scale attention based models, do the conclusions remain the same?

**Questions:**

Please address the questions in weakness part.

**Limitations:**

As indicated in weakness part.

---

> ### Author Rebuttal · Authors · 2023-08-10
>
> Thank you for your time and helpful suggestions.
>
> > **W1:**  To be honest, ... improving network structure or losses?
>
> **R:** Thank you for your questions. In response to reviewer’s concern, under **W2**, we provide and discuss real data experiments which demonstrate that our theory successfully predicts important empirical phenomena related to the optimization dynamics of transformers and attention mechanisms. To provide a more general perspective, as highlighted by Reviewer 8XUp, the recognition of implicit bias is essential across various avenues, including but not limited to fairness, behavior optimization, and generalization bounds. We provide further discussions on these aspects as follows:
>
> - **Fairness and Bias Mitigation**: Gradient descent is a fundamental optimization algorithm widely used in the training and fine-tuning of large language models.  Language models trained using gradient descent can inherit biases present in the training data. Understanding the implicit bias of gradient descent in this context allows researchers to identify and mitigate biases, ensuring fairness and ethical use of language models.
>
> - **Generalization Bounds**: The implicit bias of gradient descent is closely tied to the generalization capabilities of trained models. Understanding this relationship helps in establishing theoretical bounds on a model's generalization performance. Specifically,  the implicit bias of gradient descent influences how well language models can apply their learned knowledge to new language tasks. Understanding this relationship helps in determining how effectively language models generalize to various language-related challenges.
>
>  - **Robustness and Regularization**: Implicit bias can influence the regularization properties of gradient descent. By understanding how the algorithm tends to favor certain solutions, we can develop regularization techniques that encourage better model generalization and robustness against noise and overfitting.
>
> - **Algorithmic Choices**: Knowledge of the implicit bias of gradient descent helps in selecting appropriate optimization methods when training language models. Different algorithms exhibit varying biases, and understanding these nuances can guide the choice of optimization approach based on the desired behavior of the language model.
>
> In summary, acknowledging implicit bias in gradient descent is crucial for exploring the fairness, optimization behavior, and generalization bounds of training LLM algorithms. Our theory takes the initial steps in this direction.
>
> > **W2:** The authors ... the conclusions remain the same?
>
> **R:** Thank you for your suggestion. We have carried out additional experiments to demonstrate the expansion of our findings to real large-scale attention-based models. These experiments also serve to illustrate how our theory proficiently anticipates two essential empirical phenomena associated with the optimization dynamics of transformers and attention mechanisms:
>
>    - The attention map (i.e. softmax output) becomes more sparse over time by focusing on the most informative tokens.
>    - This is achieved by the norm of the attention weights ($W$) growing over time and leading to a "saturating" effect on softmax, resulting in a sparse pattern.
>
> Our experiments in Figure 2 verify both of these predictions. We train a vision transformer (ViT-base) model from scratch with the CIFAR-10 dataset for 400 epochs with a fixed learning rate $3\times 10^{-3}$.
>
>    - In Figure 2 (left), we present the progressive change in attention weights of the [CLS] token (which corresponds to our $p$ parameter) during training, computed from all attention heads within the model.
>   - In Figure 2 (right), we display the norm of attention weights and the sparsity level of attention maps averaged over all layers. We used $($L1norm/L2norm$)^2$ of the attention maps as a soft-sparsity measure, where a smaller value indicates a sparser vector.
>
> Initially, during the early epochs of training, the attention weights are randomly distributed, leading to a dense pattern. However, as training progresses, the weights grow, causing the attention map to gradually become sparser. Consequently, the attention map starts to focus on fewer salient patches within the image that possess distinct features that aid in classification.
>
> In light of the concerns raised by the reviewer regarding the framework investigated in this study, we find it crucial to highlight the practical significance of our attention model. Within transformers, the parameter $p$ corresponds to a trainable prompt [Lester et al. EMNLP’21] or the '[CLS]' token [Devlin et al. NAACL’19], as mentioned in [Oymak et al. ICML’23]. To comprehensively address the reviewer's concerns, we also provide new experiments that showcase the extensibility of our work. To this aim, we studied optimization dynamics of self-attention, as illustrated in the attached Figure 3, using the following objective
>
> $
> \quad L(W)=\frac{1}{n}\sum_{i=1}^n \ell(Y_i\cdot v^\top X^\top S(X_i W^\top x_{i1}))  \quad  $ (SA-ERM)
>
> This corresponds to running linear classification on the first token output of a self-attention layer ($x_{i1}$). We recently discovered a slightly modified ATT-SVM can predict the implicit bias of Self-Attention
>
> $\min_{W} \left\Vert W \right\Vert_F \quad     \text{subject to} \quad ( x_{i\alpha_i}-x_{it})^\top W x_{i1} \geq 1 \quad  \text{for all } \quad t\neq \alpha_i, i\in[n] \quad$  (S-ATT-SVM)
>
> Fig 3 shows **self-attention solutions directionally align** with (S-ATT-SVM). Empirically, we observed that optimizing $W_{prod}$ biases gradient descent towards (S-ATT-SVM) with the Frobenius norm objective, while optimizing $(W_k, W_q)$ separately biases it towards (S-ATT-SVM) with the nuclear norm objective. In short, while self-attention introduces different behavior and deserves separate investigation, we believe the attention<->SVM connection introduced by our work is fundamental and remains valid.

---

> > ### Comment · Reviewer_SE93 · 2023-08-22
> >
> > Thank you for the response. I will slightly increase score.

---

> > > ### Author Response · Authors · 2023-08-22
> > >
> > > Your review of our paper is greatly appreciated.

---

### Official Review · Reviewer_sy1C · 2023-07-27

**Soundness:** 3 good
**Presentation:** 3 good
**Contribution:** 3 good
**Rating:** 7
**Confidence:** 1

**Summary:**

The paper focusses on the optimization dynamics of attention mechanism. The authors analyze a softmax-attention model and demonstrate that running gradient descent on its parameters leads to a max margin solution, separating optimal tokens from non-optimal ones. The authors also present a regularization path analysis, demonstrating the convergence of solutions for nonlinear classifier heads. Overall, the paper aims to enhance the understanding of attention mechanisms and their optimization dynamics in large language models.

**Strengths:**

1. Comprehensive Characterization: The paper analyzes the fundamental attention model and its connection to max-margin problems. Analysis of this connection is quite original to the best of my knowledge. It overall advances the understanding of the attention mechanism from another theoretical perspective.

2. Convergence Insights: The paper looks into the convergence characterstics of gradient descent for tuning the token/prompt. This analysis can be used to drive futher improvements in optimization of large language models.

3. Joint Parameter Analysis: Through the analysis of the regularization paths, this work highlights the implicit biases and interactions between parameters (v, p), and describing their joint convergence.

4. Implications for Future Research: The work suggests promising avenues for future studies, such as exploring similar analysis for self-attention layers and multiple tunable tokens.

5. Real-World Relevance: Exhaustively understanding attention mechanisms in large language models is crucial for enhancing their performance in natural language processing tasks.

6. Numerical Validation: The authors provide empirical evidence supporting their theoretical findings through numerical experiments.

**Weaknesses:**

1. Lack of Concrete Examples: The paper could definitely benefit from providing more examples to illustrate the concepts. The findings of the paper are very abstract and make it hard for readers to grasp the implications.

2. Limited support from other works: While the paper thoroughly analyzes the attention mechanism from a max-margin problem perspective, it does not highlight directly if the claims align with other relvant theoretical analysis of attention mechanism.

3. Complexity of Analysis: The optimization-theoretic characterization might be challenging for readers without a strong background in the subject, making it less accessible to a broader audience.

4. Lack of application to Real-World Data: The paper leaves uncertainty about the applicability of the findings in practical attention based model design and applications.

**Questions:**

1. Can you provide more concrete examples illustrating the application of the findings to real-world LLM tasks?

2. How do the revealed implicit biases in the joint parameter analysis affect the interpretability and generalization capabilities of the attention model?

3. Can you elaborate on how the optimization-theoretic characterization and convergence insights presented in your work can be practically leveraged to enhance the training and fine-tuning of large language models?

**Limitations:**

The work is highly theoretical and does not present any potential negative societal impacts to the best of my knowledge.

---

> ### Author Rebuttal · Authors · 2023-08-10
>
>
> Thank you for your positive feedback and helpful suggestions.
>
> > **Q1:** Can you provide more concrete examples/applications on real-world tasks?
>
> **R:** Thank you for this suggestion. We have conducted new experiments using real data, demonstrating how our theory successfully predicts two important empirical phenomena related to the optimization dynamics of transformers and attention mechanisms:
>
>    - The attention map (i.e. softmax output) becomes more sparse over time by focusing on the most informative tokens.
>    - This is achieved by the norm of the attention weights ($W$) growing over time and leading to a "saturating" effect on softmax, resulting in a sparse pattern.
>
> Our experiments in Figure 2 verify both of these predictions in line with our gradient descent convergence in Theorems 1&3. We train a vision transformer (ViT-base) model from scratch with the CIFAR-10 dataset for 400 epochs with a fixed learning rate $3\times 10^{-3}$.
>
>    - In Figure 2 (left), we present the progressive change in attention weights of the [CLS] token (which corresponds to our $p$ parameter) during training, computed from all attention heads within the model.
>   - In Figure 2 (right), we display the norm of attention weights and the sparsity level of attention maps averaged over all layers. We used $($L1norm/L2norm$)^2$ of the attention maps as a soft-sparsity measure, where a smaller value indicates a sparser vector.
>
> Initially, during the early epochs of training, the attention weights are randomly distributed, leading to a dense pattern. However, as training progresses, the weights grow, causing the attention map to gradually become sparser. Consequently, the attention map starts to focus on fewer salient patches within the image that possess distinct features that aid in classification.
>
>
>
> > **Q2:** How do the revealed implicit biases in the joint parameter analysis affect the interpretability and generalization capabilities of the attention model? Can you elaborate on how the optimization-theoretic characterization and convergence insights presented in your work can be practically leveraged to enhance the training and fine-tuning of large language models?
>
>
> **R:**  Our joint analysis in Section 3 has a surprisingly interpretable message: $p$ and $v$ (essentially) converge to their respective max-margin solutions, thus, optimization dynamics of “classification” (v) and “attention” (p) can be decoupled. Second, as also pointed out by Reviewer 8XUp, understanding implicit bias is essential across various domains, including interpretability, fairness, optimizer choice, and generalization bounds, because it connects complex optimization dynamics to amenable problems (like our attention SVM). Future works can study various aspects of transformers (TF) through the SVM lens. Below, we provide detailed discussion in the context of language models and how our optimization-theoretic characterization can enhance the training and fine-tuning of LLMs.
>
> * **Fairness and Bias Mitigation**: Gradient descent is a fundamental optimization algorithm widely used in the training and fine-tuning of large language models.  Language models trained using gradient descent can inherit biases present in the training data [[GPT-4] (https://arxiv.org/pdf/2303.08774.pdf)]. Understanding the implicit bias of gradient descent in this context allows researchers to identify and mitigate biases, ensuring fairness and ethical use of language models.
>
>
> * **Generalization Bounds**: The implicit bias of gradient descent is closely tied to the generalization capabilities of trained models [[Vardi23](https://arxiv.org/abs/2208.12591)]. Understanding this relationship helps in establishing theoretical bounds on a model's generalization performance. Specifically, the implicit bias of gradient descent influences how well language models can apply their learned knowledge to new language tasks. Understanding this relationship helps in determining how effectively language models generalize to various language-related challenges.
>
> * **Robustness and Regularization**: Implicit bias can influence the regularization properties of gradient descent. By understanding how the algorithm tends to favor certain solutions, we can develop regularization techniques that encourage better model generalization and robustness against noise and overfitting.
>
> * **Algorithmic Choices**: Knowledge of the implicit bias of gradient descent helps in selecting appropriate optimization methods when training language models. Different algorithms exhibit varying biases, and understanding these nuances can guide the choice of optimization approach based on the desired behavior of the language model. Finally, it is also possible that one can develop new training algorithms: For instance, can we literally solve an SVM during training to accelerate TF optimization (e.g. after identifying which tokens to separate with SVM)?
>
> In conclusion, characterizing implicit bias is paramount for multiple aspects and, by comprehending the interplay between optimization and these aspects, we can potentially enhance training and fine-tuning processes, leading to more principled, efficient, and trustworthy language models.

---

### Author Rebuttal · Authors · 2023-08-10

We sincerely appreciate the time and efforts of the reviewers. We highlight the **main contributions (C1-C3)** of the paper and present **new experiments (E1-E4)** along with explanations for the corresponding **attached figures (Figs 1-4)**. We would be grateful to respond to any reviewer inquiries during the discussion period.

**C1:**  As Reviewer 8XUp kindly states: "The paper is the **first work** on the implicit bias behavior of GD for the attention mechanism". Implicit regularization is extensively studied for linear models and standard neural net architectures [Soudry et al. JMLR’18; Gunasekar et al. NeurIPS'18; Arora et al. NeurIPS’19; Li, Wang et al. NeurIPS’22; Frei et al. ICLR’23 and more]. However, the attention mechanism remains an important unexplored topic. We investigate attention's optimization landscape and analyze its implicit bias, shedding light on the role of softmax nonlinearity. Soudry et al. and others connect logistic regression to standard SVM, which separates inputs based on their labels. Instead, we show attention is biased towards ATT-SVM which separates and selects optimal tokens within the input sequences.

**C2:** We make innovative and nontrivial theoretical contributions:
  - Defs. 1 and 2 introduce novel concepts of  **token scores** and **locally-optimal tokens**, and Lems 2-5 present innovative theories integrating these concepts with the softmax nonlinearity's special structure.
  - Gradient analysis (specifically Thm 3) requires novel proof ideas that capture the optimization dynamics in terms of the local SVM geometry and conic initialization around the max-margin solution.
  - Our regularization path analysis in Secs. 3 and 4 yield nontrivial findings, remarkably predicting the implicit bias of gradient descent when jointly optimizing $(v,p)$.

**C3.**  Reviewers sy1C, 5i57, and 8XUp acknowledge the potential of **our work to open new research avenues**.  ​​We see the connection between attention and ATT-SVM as fundamental, providing a general framework to comprehend complex architectures and generalization dynamics,  akin to the logistic regression and classical SVM connection in deep learning theory  [Soudry et al. JMLR’18, Rosset et al. NeurIPS’03, Telgarsky ICML’13 and more]. See **E3** below for supporting evidence.

**Supporting Experiments**

**E1:**  Reviewer iE2T remarks that transformers use separate key and query weights $(W_k, W_q)$, while our approach uses the combined matrix $W_{prod}:=W_q W_k^\top$. In Fig 1 (attached), we demonstrate that regardless of whether we optimize $W_k$ and $W_q$ separately or as $W_{prod}$, the resulting **trajectories align in direction with the ATT-SVM solution**. As predicted by Lemma 1, $W_{prod}$ converges to a rank-1 matrix of the form $a {p^{mm}}^\top$, where $p^{mm}$ is the parameter obtained by running gradient descent on the $p$ parameter. We use the objective

$
\qquad   L(W_k,W_q)=\frac{1}{n}\sum_{i=1}^n \ell(Y_i\cdot v^\top X_i^\top \mathbb{S}(X_i W_k W_q^\top a)) \qquad  $  (QK-ERM)

with a fixed vector $a$, in accordance with Lemma 1.

    More in the response to Reviewer iE2T.

**E2:** Our theory predicts crucial empirical phenomena in transformer optimization dynamics:

- The attention map (i.e. softmax output) becomes more sparse over time by focusing on the most informative tokens.
- To do so, the norm of the attention weights $W$ should grow over time and “saturate” softmax towards a sparse pattern.

In support of this, Fig 2 (left) displays the evolving attention map of the [CLS] token during training, aggregated from all attention heads of a vision transformer. Fig 2 (right) shows the average norm of attention weights and sparsity level of attention maps across all layers. Initially dense and random, the attention weights gradually grow, leading to a sparser attention map.

     More in the response to Reviewer sy1C.

**E3:** In Fig 3, we consider the **self-attention** objective

$
\qquad L(W)=\frac{1}{n}\sum_{i=1}^n \ell(Y_i\cdot v^\top X^\top S(X_i W^\top x_{i1}))  \qquad  $ (SA-ERM)

This corresponds to running linear classification on the first token output of a self-attention layer ($x_{i1}$). We recently discovered a slightly modified ATT-SVM can predict the implicit bias of Self-Attention

$\min_{W} \left\Vert W \right\Vert_F \quad     \text{subject to} \quad ( x_{i\alpha_i}-x_{it})^\top W x_{i1} \geq 1 \quad  \text{for all } \quad t\neq \alpha_i, i\in[n]  \qquad $  (S-ATT-SVM)

Fig 3 shows self-attention solutions directionally align with (S-ATT-SVM). Empirically, we found that optimizing $W_{prod}$ biases gradient descent towards (S-ATT-SVM) with the Frobenius norm objective, while optimizing $(W_k, W_q)$ separately biases it towards (S-ATT-SVM) with the nuclear norm objective. In our paper's setting (Lemma 1 and **E1** above), this distinction vanishes as we fix $x_{i1}\gets a$, leading to a rank-1 matrix solution for S-ATT-SVM.

     More in the response to Reviewer iE2T.

**E4:** We study the convergence of attention with random problem instances with $n=4, T=6$ and varying dimension $d = 2,5,10,100,300,500$. We find that,
- **(1, red bar)** for small $d=2,5$, $p(t)$ generated by gradient descent does not have to saturate softmax i.e. may not converge to a local ($p^{mm}$) or global ($p^{mm\star}$) max-margin direction. This is likely because the ATT-SVM is not feasible for small $d$.
-  **(2, blue bar)** For larger $d$’s,  $p(t)$  indeed converges to a $p^{mm}$.
- **(3, green bar)** As $d$ gets even larger, $p(t)$  converges more frequently to the $p^{mm\star}$ (for $d\geq 300$, 99% of the problem instances). We believe over-parameterization (large $d$) enables global convergence because as $d$ grows, the global direction $p^{mm\star}$ becomes the only direction that satisfies our Definition 2 (local optimality). Thus, local and global convergence coincide.

.

    More in the response to Reviewers CU4X & 5i57.

---

### Decision · Program_Chairs · 2023-09-21

**Decision:**

Accept (spotlight)

**Comment:**

The paper studies the optimization dynamics of softmax attention mechanism, which can potentially have a great impact as the softmax attention is widely used in different applications. The authors showed that  softmax-attention weights trained with gradient descent converge to a max-margin solution that separates locally-optimal tokens from non-optimal ones. Reviewers generally liked the idea and appreciated the contribution. This paper would be of broad interest of the community that are working on improving softmax attention models.